# Return of individual research results from genomic research: A systematic review of stakeholder perspectives

Danya F. Vears[1,2,3,4]*, Joel T. Minion[5,6], Stephanie J. Roberts[5], James Cummings[7], Mavis Machirori[5,8], Mwenza Blell[5], Isabelle Budin-Ljøsne[9], Lorraine Cowley[10,11], Stephanie O. M. Dyke[12], Clara Gaff[2,13,14], Robert Green[15,16,17,18], Alison Hall[19], Amber L. Johns[20,21], Bartha M. Knoppers[22], Stephanie Mulrine[23], Christine Patch[24,25], Eva Winkler[26], Madeleine J. Murtagh[27,28]

1 Melbourne Law School, University of Melbourne, Carlton, Australia, 2 Murdoch Children's Research Institute, The Royal Children's Hospital, Parkville, Australia, 3 Center for Biomedical Ethics and Law, Department of Public Health and Primary Care, KU Leuven, Leuven, Belgium, 4 Leuven Institute for Human Genetics and Society, Leuven, Belgium, 5 Policy, Ethics and Life Sciences (PEALS) Research Centre, Newcastle University, Newcastle, United Kingdom, 6 Department of Community Health Sciences, O'Brien Institute for Public Health, University of Calgary, Calgary, Canada, 7 School of Art, Media and American Studies, University of East Anglia, Norwich, United Kingdom, 8 Ada Lovelace Institute, London, United Kingdom, 9 Department of Genetics and Bioinformatics, Norwegian Institute of Public Health, Oslo, Norway, 10 Newcastle upon Tyne NHS Foundation Hospitals Trust, Northern Genetics Service, Centre for Life, Newcastle, United Kingdom, 11 Population Health Sciences Institute, Newcastle University, Newcastle, United Kingdom, 12 McGill Centre for Integrative Neuroscience, Montreal Neurological Institute, Department of Neurology & Neurosurgery, McGill University, Montreal, Canada, 13 Department of Paediatrics, Faculty of Medicine Dentistry & Health Sciences, The University of Melbourne, Parkville, Australia, 14 Walter and Eliza Hall Institute of Medical Research, Parkville, Australia, 15 Harvard Medical School, Boston, Massachusetts, United States of America, 16 Mass General Brigham, Boston, Massachusetts, United States of America, 17 Broad Institute, Boston, Massachusetts, United States of America, 18 Ariadne Labs, Boston, Massachusetts, United States of America, 19 PHG Foundation, University of Cambridge, Cambridge, United Kingdom, 20 Cancer Division, Garvan Institute of Medical Research, Sydney, Australia, 21 International Cancer Genome Consortium, University of Glasgow, Glasgow, United Kingdom, 22 Centre of Genomics and Policy, McGill University, Montreal, Canada, 23 Northumbria University, Newcastle, United Kingdom, 24 Genomics England, Queen Mary University of London, London, United Kingdom, 25 Society and Ethics Research Group, Connecting Science, Wellcome Genome Campus, Cambridge, United Kingdom, 26 National Center for Tumour Diseases (NCT), Section of Translational Medical Ethics, University of Heidelberg, Heidelberg, Germany, 27 University of Glasgow, Glasgow, United Kingdom, 28 Newcastle University, Newcastle, United Kingdom

* danya.vears@mcri.edu.au

**Data Availability Statement:** All relevant data are within the manuscript and its Supporting information files.

## Abstract

Despite the plethora of empirical studies conducted to date, debate continues about whether and to what extent results should be returned to participants of genomic research. We aimed to systematically review the empirical literature exploring stakeholders' perspectives on return of individual research results (IRR) from genomic research. We examined preferences for receiving or willingness to return IRR, and experiences with either receiving or returning them. The systematic searches were conducted across five major databases in August 2018 and repeated in April 2020, and included studies reporting findings from primary research regardless of method (quantitative, qualitative, mixed). Articles that related to the clinical setting were excluded. Our search identified 221 articles that met our search criteria. This included 118 quantitative, 69 qualitative and 34 mixed methods studies. These

**Funding:** This work was supported by the Australian Government through the Medical Research Future Fund, as part of the Genomics Health Futures Mission (Grant number 76749 - DV). We are grateful for funding support from ESRC, MRC and Wellcome Trust (METADAC, Grant agreements MR/N01104X/1, ES/S008349/1, MR/N01104X/2, 213422/Z/18/Z - MJM, MB, JC, JTM, MM, SJR; 206194 - CP) and the EU Horizon 2020 programme (EUCAN-connect, Grant agreement ID: 824989 - MJM, MB, JTM, MM), and the Canada Research Chair in Law and Medicine; Genome Quebec; Genome Canada (BMK); Canada Institute of Health Research (BMK), the NIH (Grants HL143295, TR003201 - RG), the Franca Fund (RG) and Biobank Norway funded by The Research Council of Norway <https://www.forskningsradet.no/en/>, (grant number 296162/F50 – IBL). We acknowledge Can-SHARE Connect (2019-2020): Supporting the Regulatory and Ethics Work Stream of the Global Alliance for Genomics and Health_GA4GH.

**Competing interests:** The authors have declared that no competing interests exist.

articles included a total number of 118,874 stakeholders with research participants (85,270/72%) and members of the general public (40,967/35%) being the largest groups represented. The articles spanned at least 22 different countries with most (144/65%) being from the USA. Most (76%) discussed clinical research projects, rather than biobanks. More than half (58%) gauged views that were hypothetical. We found overwhelming evidence of high interest in return of IRR from potential and actual genomic research participants. There is also a general willingness to provide such results by researchers and health professionals, although they tend to adopt a more cautious stance. While all results are desired to some degree, those that have the potential to change clinical management are generally prioritized by all stakeholders. Professional stakeholders appear more willing to return results that are reliable and clinically relevant than those that are less reliable and lack clinical relevance. The lack of evidence for significant enduring psychological harm and the clear benefits to some research participants suggest that researchers should be returning actionable IRRs to participants.

## Introduction

Although next generation sequencing technologies (NGS) were implemented in the research setting well over a decade ago, debate continues about whether and to what extent results from genomic research should be returned to participants. An individual research result (IRR) broadly refers to any finding that arises from the research endeavour, which can include: 1) study-specific results (i.e., results related to the condition under investigation; SSR), 2) unsolicited findings (i.e., disease-causing variants unrelated to the genetic condition under investigation that are identified inadvertently during the research study; UF), 3) secondary findings (i.e., disease-causing variants unrelated to the genetic condition under investigation that are actively searched for by the research team; SF).

A plethora of empirical studies show that participants have high interest in receiving individual research results (IRR) [1–13]. A recent policy on Clinically Actionable Genomic Research Results from the Global Alliance for Genomics and Health (GA4GH), has advocated for return of results to participants, arguing that an ethical and legal consensus on this point is emerging [14]. Yet, there remains a degree of hesitancy from some research projects and biobanks to return IRR to participants, the reasons for which are numerous, complex, and context-dependent. Examination of the reasons for the hesitation to return IRR will help us understand the challenges researchers and other professional stakeholders either foresee, or are experiencing in the return process, which will then enable the development of systems to support this process.

Although some authors have postulated that drawing a distinct boundary between the research and clinic contexts is inappropriate in translational genomics [15–17], we believe it is important to explore the issue of return of results between the two settings separately. While we acknowledge that there is significant overlap between the clinic and the research setting in some situations [17], including hybrid models such as the 100,000 Genomes project [18], we argue that what differs between the two is the primary goal of the genetic analysis. The primary goal of testing in the clinical setting is to identify any potential underlying genetic contributions to the condition seen in the patient, or to provide genomic risk information in healthy individuals, and is performed under the auspices of clinical care and established guidelines. In

research, the primary goal is to generate knowledge and, in certain circumstances, a secondary goal of using the results to guide clinical care is also present [19]. This distinction arises because the duty of care that a clinician has to their patients is different to the responsibilities a researcher has to a research participant. This distinction applies even to a clinician-researcher, depending on their relationship with the patient-participant. It equally suggests that the degree to which it is appropriate or necessary to return particular types of results, such as unsolicited findings, secondary findings and variants of uncertain significance (i.e., variants that could potentially be the cause of the genetic condition under study but where existing evidence is insufficient to classify the variant as either (likely) benign or (likely) pathogenic; VUS) [20], will be different between the two contexts. It is important to recognize that in the clinical setting, results that are returned to patients can be positive (i.e., a cause has been identified), negative (i.e., a cause has not been identified), or uncertain (i.e., something has been found but its clinical significance cannot be determined). Yet, in the context of research, a result that is returned will almost always be positive; participants are unlikely to be informed that nothing has been found. For these reasons, there is a need to focus specifically on the research setting, where the necessity and appropriateness of returning findings to patients may be less distinct.

Previous literature reviews relating to return of results have not only chosen to include publications that reported on return of SF in both clinical and research settings, they also focused purely on return of SF [21] rather than return of IRR overall. Although SF are an important type of IRR, the majority of results identified through genomic research will be either study-related or identified inadvertently through the course of the research. As such, a comprehensive understanding of stakeholder perspectives on receiving all types of IRR is required.

The legal and regulatory landscape regarding return of results currently comprises a patchwork of often contradictory rules for researchers, especially where research collaborations stretch across countries and continents as many now do in the field of genomics. In their recent review, Thorogood et al identified sufficient discrepancies between policies to prevent reconciliation of rules about which results should or should not be returned in research projects [22]. Moreover, they found that policies, including thresholds for data quality and clinical significance, were evolving in uneven ways, further complicating policy development for return of results. Thorogood et al call for greater clarity in the ethical and policy approach to return of results. We argue that such clarity must be based on the actual wishes and perspectives of those most affected by policy development for return of results.

To address these gaps, we aimed to systematically review the empirical literature exploring stakeholders' (i.e., participants, patients, publics, health professionals, researchers, Institutional Review Boards (IRBs), or mixed professionals; for definitions see Table 1) perspectives and experiences with return of individual results from genomic research to examine the interest in receiving, or willingness to return IRR, and experiences with receiving or returning IRR.

## Methods

In 2018, the four lead authors (DFV, JTM, SJR, MJM) registered a systematic review protocol on Prospero (https://www.crd.york.ac.uk/prospero/display_record.php?RecordID=117551). Two authors (JTM, SJR) formulated a search strategy to identify studies reporting 1) stakeholders' views on whether or not research participants, and potentially their relatives, should receive individual genomics findings, 2) which findings should be returned and the reasons for such opinions and perspectives, and 3) experiences with either receiving IRR or returning them to research participants. This search strategy was reviewed and refined by the other two lead authors (MJM and DFV). Searches were conducted across five major databases in August

**Table 1. Glossary of stakeholder, setting and context definitions.**

| Stakeholders | Definition |
|---|---|
| Participants | Respondents' views on return of research results elicited based on their position as a participant of a research study, regardless of whether or not they were offered results. |
| Patients (and parents of patients) | Respondents' views on return of research results elicited based on their clinical status as a patient. This includes parents of patients in the case of minors. |
| Publics | Respondents' views on return of research results elicited based on their position as a member of the community. |
| Health professionals | Respondents recruited based on their status as a health professional, with or without genetics training. |
| Researchers | Respondents recruited based on their status as a researcher, although some also had medical degrees (i.e., clinical researchers) or were directors of research groups or biobanks. |
| Institutional Review Boards | Respondents were members of institutional review boards, including general members, chairs and coordinators. |
| Mixed professionals | Respondents included individuals from different professional groups, including (but not limited to) health professionals, researchers, medical centre representatives and government representatives. |
| **Setting** | |
| Clinical research | Respondents were asked to consider their views on return of research results within the context of a clinical research setting (i.e., studies undertaken to investigate the genetic basis of disease whether or not they were research participants). |
| Biobanks | Respondents were asked to consider their views on return of research results within the biobank setting, regardless of whether or not they were biobanks participants. |
| **Context** | |
| Hypothetical / Policy / Practice | Respondents were asked to make hypothetical decisions about whether or not they wished to receive results and, in some cases, which results they wished to receive. |
| Actual decisions | Respondents were asked to decide whether or not they wished to receive results and, in some cases, which results they wished to receive. This applied both to those who had and had not received their results at the time of the study. |

2018 and repeated in April 2020 and then again in May 2021 (see Fig 1). Details about search terms used can be found in S1 Table.

The review considered studies reporting findings from primary research regardless of method (quantitative, qualitative, mixed). Specific articles were included if they were published from 2005 onwards; in English; contained empirical data; and related to the return of results from genomic research in a research (not clinical) context, which also included biobanks, as well as both germline and somatic research testing. Articles that incorporated both clinical and research settings were included. Articles were rejected if they were opinion-based or review papers; related only to clinical or diagnostic sequencing; discussed solely the return of raw data (i.e., not annotated/interpreted); related only to the direct-to-consumer context; or related only to carrier screening or neonatal genomic screening, rather than diagnostic testing, even when part of a research protocol.

For the 2018, 2020 and 2021 search results, the lead authors independently screened titles (and abstracts as necessary) for all records identified once duplicates had been removed. Full text articles were consulted where it remained unclear whether the article met the inclusion criteria. Full text screening was then undertaken for all articles thought to meet the inclusion criteria. Each record/article was screened by at least two lead authors, and disagreements were resolved through discussions among at least three of the lead authors.

For articles identified in 2018, data extraction was undertaken by 19 members of the Return of Results Task Team of the Regulatory and Ethics Work Stream of the Global Alliance for Genomics and Health (GA4GH). All were active data and/or health researchers in North

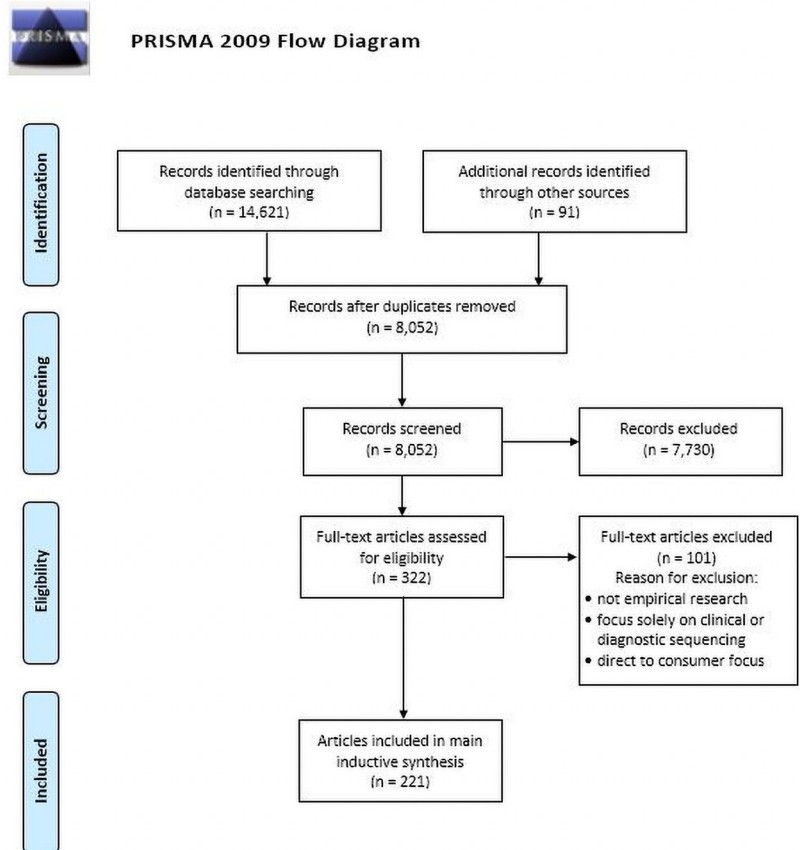

**Fig 1. PRISMA flowchart.** *From*: Moher D, Liberati A, Tetzlaff J, Altman DG, The PRISMA Group (2009). *Preferred Reporting Items for Systematic Reviews and Meta-Analyses: The PRISMA Statement*. PLoS Med 6(7): e1000097. doi:10.1371/journal.pmed1000097 **For more information, visit** www.prisma-statement.org.

America, Europe, Asia or Australia. Each volunteer was randomly assigned six articles and provided with an extraction form and instructions developed by the lead authors. Data from remaining articles was extracted by two authors (JTM, MJM). The extraction form requested study details, participant characteristics, main findings and conclusions, relevant references, and reviewers' assessment of relevance to the research question and of the quality of the article (see S2 Table). Where only some of an article's content was eligible for inclusion, only this data was extracted and included for synthesis. Data extractors were provided with a written overview of the study aims and criteria for article inclusion. They were instructed to complete all fields of the data extraction form for each of their allocated articles and, when entering the 'Key findings' section, to only insert data that was relevant to the research question. Data extractors were encouraged to seek clarity from the lead authors via email where necessary.

Once complete, each form was screened for completeness by JTM before being independently quality checked, including checking the quality assessment of each paper, against the article by one of the other lead authors. During the extraction process, three articles were removed for not meeting the inclusion criteria, while five articles identified from reference lists were deemed eligible for full data extraction. No articles were excluded due to poor quality. This process was repeated twice: in April 2020, the initial search process was repeated to identify new publications since 2018. Full data extraction was conducted by two authors (JTM, MJM) and two volunteers (JC, MM). All articles were quality checked by DFV. In this instance,

ten articles were removed during extraction for not meeting the inclusion criteria, while 17 were identified from reference lists as eligible for full extraction. In May 2021, this process was updated once more, with ten articles again removed and eight identified through reference lists. For this final cycle, data extraction was conducted by two authors (JTM, MJM) and checked by a third (DFV). Across all three searches, every record/article was ultimately assessed in full by at least two reviewers by the end of the screening and extraction process.

Given the number and diversity of studies and methodologies covered by the review, no single or combination of existing disciplinary or methodologically appropriate quality appraisal tools could enable valid or rational comparison between the studies. Instead, we employed a reason-based approach derived from the bioethics literature to evaluate and critique the "literature in a transparent and systematic way in order to provide a comprehensive and unbiased overview of the information sought" [23]. To account for the heterogeneity of methodologies, assessment was based on the transparency of findings, methodological appropriateness and coherence of the findings with the methodological approach. We assessed the clarity and appropriateness of the methodology on the following aspects: study description, methods, methodology, analysis and conclusions [24]. All articles were broadly appraised for quality by the data extractors during the data extraction process. All articles were further reviewed for quality by DFV, JTM and MJM during the checking processes.

We used content analysis to enable systematic analysis of the methodologically diverse articles in this review [25, 26]. Data extracted from the articles were analyzed using inductive content analysis in which content categories were derived from the data, rather than predetermined [27–29]. The data were coded into broad content categories, such as 'preferences and expectations for return of study-specific results', 'preferences for unsolicited or secondary findings' and 'experiences receiving or returning results'. Subcategories were delineated within these broad categories in two stages: 1) based on the stakeholders from which the perspectives were gained, 2) based on study setting (i.e., clinical research versus biobanks). Note was also taken about whether the stakeholder perspectives were gauged in a hypothetical/policy context, whether they were being asked to make decisions about results that would be returned in the future, or where results were returned. Data were coded and interpreted by DV using Word documents; MM analysed subsets of the data to confirm the coding scheme.

## Results

Our search identified 221 articles that met our search criteria. This included 97 (53.4%) quantitative, 54 (31.2%) qualitative and 32 (15.4%) mixed methods studies (Figs 2 and 3). These articles included a total number of 118,874 stakeholders with research participants (85,270/72%) and members of the general public (40,967/35%) being the largest groups represented. The articles spanned at least 17 different countries with most (125/65%) being from the USA (Figs 4 and 5). A high proportion (76%) related to research projects as distinct from the biobank setting. More than half (58%) gauged views that were hypothetical. A complete list of the articles and their characteristics, such as participant numbers and country of origin, can be found in Table 2. A summary of the study demographics can be found in Table 3.

Here we present the data for three data categories from our analysis that correspond with our research question: 1) views on return of study-specific results; 2) views on return of UF and SF findings; 3) experiences with receiving IRR (participants) or returning results (health professionals). Data are presented grouped by stakeholder, by setting, and in relation to study context (i.e., whether the participants were being asked to comment on return of results in a hypothetical/policy setting or whether they were being asked to make real decisions about receiving or returning results).

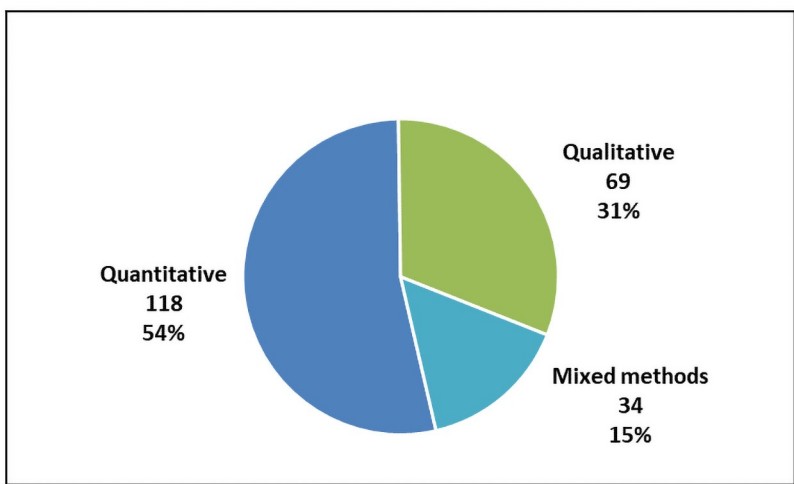

**Fig 2. Number of articles by research method.**

## 1. Views on return of study-specific results

Summaries of interest in receiving SSR for participants, patients (and their parents), and members of the public can be found in Table 4.

### 1.1 Participants' preferences for receiving study-specific results

**Clinical research setting.** Overall, participants, and parents of children participating in genomic research studies, generally have strong preferences for, or expectations to receive SSR. Those wanting results that are related to the research question ranged from 47.6% [10] to 97% [6]. Percentages seemed to depend on the genetic condition being studied, whether the decision to receive results was hypothetical or actual (either with or without results having been returned), and the framing of the question. Receiving SSR was also highlighted by participants as a high motivation for participation [30, 32, 64], based on the hope of a cure for themselves and future generations [65].

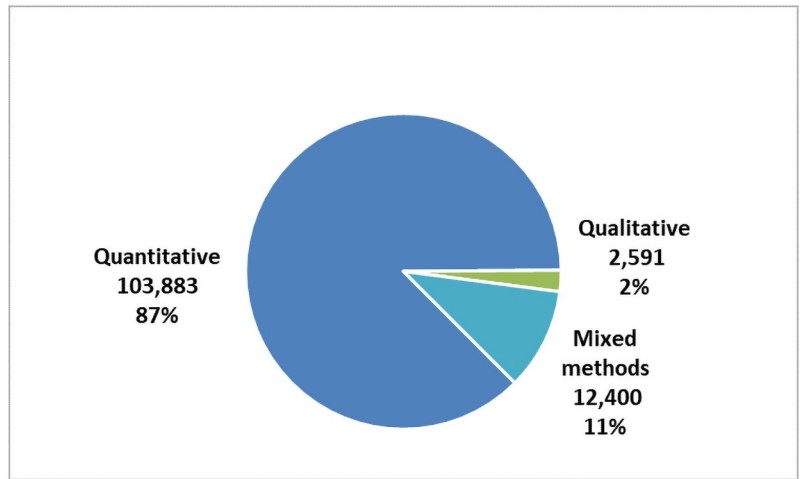

**Fig 3. Number of participants by research method.**

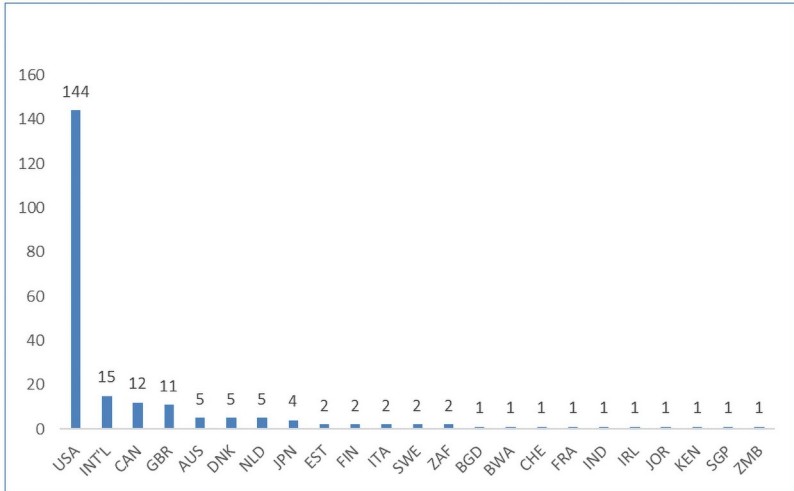

**Fig 4. Number of articles by country.**

Investigating a hypothetical context, a study of 103 racially diverse adult patients with late-stage kidney disease participating in a genomic study showed that 62.13% felt it was very important to receive genetic and health results related to the condition under study, with only a small proportion (11.7%) indicating return of results was not important at all [3]. Likewise, in a study of 241 persons with mental disorders, 95% wanted 'pertinent' findings to be made available [13]. Studies have shown that participants without an existing health condition are highly interested in receiving all categories of results [5] and generally prefer 'knowing' to 'not knowing' [40]. Yet, a US-based study of 311 mostly healthy volunteers, referred to as the Clin-Seq study, indicated that interest was highest for receiving results for treatable/preventable conditions and carrier status [5]. This accords with other studies where respondents have been most interested in receiving information about genes that are life-threatening or may increase their risks for future health problems, where preventable options are available, or that are likely

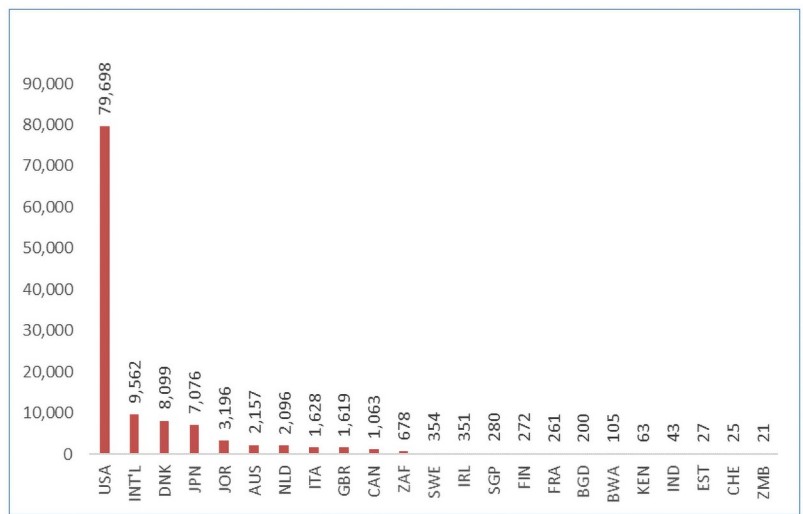

**Fig 5. Number of participants by country.**

**Table 2. Details of all articles included.**

| Full Citation | Data Type | Focusgroups | Interviews | Survey | Other | Country | Results (to be) Received | Results (to be) Given | Biobanks | Research | HCPs | Reivew Boards | Researchers | Research Participants | Publics | Hypothetical/Policy/Practice | Decided | Given/Received |
|---|---|---|---|---|---|---|---|---|---|---|---|---|---|---|---|---|---|---|
| | | | | | | | | | Context | | Stakeholders | | | | | Decision Type | | |
| Abul-Husn NS, Soper ER, Braganza GT, Rodriguez JE, Zeid N, Cullina S, et al. Implementing genomic screening in diverse populations. Genome Medicine, 2021;13(17):1–11. | Quant | | | 7,535 | | USA | Own | | X | | | | | X | | X | X | X |
| Ahram M, Othman A, Shahrouri M, Mustafa E. Factors influencing public participation in biobanking. European Journal of Human Genetics. 2014;22(4):445–51. | Quant | | | 3,196 | | JOR | Own | | X | | | | | | X | X | | X |
| Alahmad G, Alzahrany H and Almutairi AF. Returning results of stored biological samples and biobanks: perspectives of Saudi Arabian biomedical researchers. Biopreservation and Biobanking, 2020;18(5):395–402. | Qual | | 19 | | | INT'L | | Adults | X | | | | X | | | X | | |
| Allen NL, Karlson EW, Malspeis S, Lu B, Seidman CE, Lehmann LS. Biobank participants' preferences for disclosure of genetic research results: perspectives from the OurGenes, OurHealth, OurCommunity project. Mayo Clinic Proceedings. 2014;89(6):738–46. | Quant | | | 555 | | USA | Own | | X | | | | | X | | X | | |
| Amendola LM, Horike-Pyne M, Trinidad SB, Fullerton SM, Evans BJ, Burke W, et al. Patients' choices for return of exome sequencing results to relatives in the event of their death. The Journal of Law, Medicine & Ethics. 2015;43(3):476–85. | Quant | | | | 78 | USA | Relatives' | | | X | | | | X | | | X | |
| Amendola LM, Robinson JO, Hart R, Biswas S, Lee K, Bernhardt BA, et al. Why patients decline genomic sequencing studies: experiences from the CSER consortium. Journal of Genetic Counseling. 2018;27(5):1220–7. | Quant | | | 1,088 | | USA | Own & Child's | | | X | | | | X | | | X | |
| Anderson J, Meyn M, Shuman C, Shaul RZ, Mantella L, Szego M, et al. Parents perspectives on whole genome sequencing for their children: qualified enthusiasm? Journal of Medical Ethics. 2017;43(8):535–9. | Qual | | 23 | | | CAN | Child's | | | X | | | | X | | | X | |
| Anderson RL, Murray K, Chong JX, Ouwenga R, Antillon M, Chen P, et al. Disclosure of genetic research results to members of a founder population. Journal of Genetic Counseling. 2014;23(6):984–91. | Mixed | | | 86 | 448 | USA | Own | | | X | | | | X | | X | | X |
| Appelbaum PS, Fyer A, Klitzman RL, Martinez J, Parens E, Zhang Y, et al. Researchers' views on informed consent for return of secondary results in genomic research. Genetics in Medicine. 2015;17(8):644–50. | Quant | | | 198 | | USA | | Adults & Children | | X | | | X | | | X | | |

(Continued)

**Table 2.** (Continued)

| Full Citation | Data Type | Focusgroups | Interviews | Survey | Other | Country | Results (to be) Received | Results (to be) Given | Biobanks | Research | HCPs | Review Boards | Researchers | Research Participants | Publics | Hypothetical/Policy/Practice | Decided | Given/Received |
|---|---|---|---|---|---|---|---|---|---|---|---|---|---|---|---|---|---|---|
| Appelbaum PS, Waldman CR, Fyer A, Klitzman R, Parens E, Martinez J, et al. Informed consent for return of incidental findings in genomic research. Genetics in Medicine. 2014;16(5):367–73. | Mixed | | 48 | 254 | | USA | Own & Child's | Adults & Children | X | X | X | | X | X | | X | | |
| Arar N, Seo J, Lee S, Abboud HE, Copeland L, Noel P, et al. Preferences regarding genetic research results: comparing veterans and nonveterans responses. Public Health Genomics. 2010;13(7–8):431–9. | Quant | | | 1,522 | | USA | Own & Relatives' | | | X | | | | X | | | X | |
| Ashida S, Koehly LM, Roberts JS, Chen CA, Hiraki S, Green RC. The role of disease perceptions and results sharing in psychological adaptation after genetic susceptibility testing: the REVEAL Study. European Journal of Human Genetics. 2010;18(12):1296–301. | Quant | | | 269 | | USA | Own | | | X | | | | X | | | | X |
| Bacon PL, Harris ED, Ziniel SI, Savage SK, Weitzman ER, Green RC, et al. The development of a preference-setting model for the return of individual genomic research results. Journal of Empirical Research on Human Research Ethics. 2015;10(2):107–20. | Qual | | 25 | | | USA | Child's | | | X | | | | X | | X | | |
| Bak MAR, Veeken R, Blom MT, Tan HL and Willems DL. Health data research on sudden cardiac arrest: perspectives of survivors and their next-of-kin. BMC Medical Ethics, 2021;22(1):1–15. | Qual | | 17 | | | NLD | Own & Relatives' | | | X | | | | X | | | X | |
| Ballard LM, Horton RH, Dheensa S, Fenwick A, Lucassen AM. Exploring broad consent in the context of the 100,000 Genomes Project: a mixed methods study. European Journal of Human Genetics. 2020;28:732–41. | Mixed | | 24 | 1,337 | | GBR | Own | | | X | | | | X | | | X | |
| Barazzetti G, Cavalli S, Benaroyo L, Kaufmann A. "Still rather hazy at present": citizens' and physicians' views on returning results from biobank research using broad consent. Genetic testing and molecular biomarkers. 2017;21(3):159–65. | Qual | 16 | 9 | | | CHE | Own | Adults | X | X | X | | X | X | X | X | | |
| Baret L, Godard B. Opinions and intentions of parents of an autistic child toward genetic research results: two typical profiles. European Journal of Human Genetics. 2011;19(11):1127–32. | Quant | | | 158 | | CAN | Child's | | | X | | | | X | | X | | |
| Basson F, Futter MJ, Greenberg J. Qualitative research methodology in the exploration of patients' perceptions of participating in a genetic research program. Ophthalmic genetics. 2007;28(3):143–9. | Qual | | 4 | | | ZAF | Own | | | X | | | | X | | | | X |
| Beil A, Hornsby W, Uhlmann WR, Aatre R, Arscott P, Wolford B, et al. Disclosure of clinically actionable genetic variants to thoracic aortic dissection biobank participants. BMC Medical Genomics. 2021;14(66):1–12. | Quant | | | 10 | | USA | Own | | X | | | | | X | | | | X |

*(Continued)*

**Table 2.** (Continued)

| Full Citation | Data Type | Focusgroups | Interviews | Survey | Other | Country | Results (to be) Received | Results (to be) Given | Biobanks | Research | HCPs | Review Boards | Researchers | Research Participants | Publics | Hypothetical/Policy/Practice | Decided | Given/Received |
|---|---|---|---|---|---|---|---|---|---|---|---|---|---|---|---|---|---|---|
| Bergner AL, Bollinger J, Raraigh KS, Tichnell C, Murray B, Blout CL, et al. Informed consent for exome sequencing research in families with genetic disease: the emerging issue of incidental findings. American Journal of Medical Genetics Part A. 2014;164 (11):2745–52. | Qual | | 15 | | | USA | Own & Child's | | | X | | | | X | | | X | |
| Berrios C, James CA, Raraigh K, Bollinger J, Murray B, Tichnell C, et al. Enrolling genomics research participants through a clinical setting: the impact of existing clinical relationships on informed consent and expectations for return of research results. Journal of Genetic Counseling. 2018;27(1):263–73. | Qual | | 15 | | | USA | Own & Child's | | | X | | | | X | | | X | |
| Beskow LM and Smolek SJ. Prospective biorepository participants' perspectives on access to research results. Journal of Empirical Research on Human Research Ethics, 2009;4(3):99–111. | Qual | | 40 | | | USA | Own | | X | | | | | X | | X | | |
| Beskow LM, O'Rourke PP. Return of genetic research results to participants and families: IRB perspectives and roles. The Journal of Law, Medicine & Ethics. 2015;43(3):502–13. | Quant | | | 65 | | USA | | Adults & Relatives' | | X | | X | | | | X | | |
| Blazek AD, Kinnamon DD, Jordan E, Ni HY and Hershberger RE. Attitudes of dilated cardiomyopathy patients and investigators toward genomic study enrollment, consent process, and return of genetic results. Clinical and Translational Science, 2021;14(2):550–557. | Quant | | | 34 | | USA | Own | | | X | | | X | X | | X | | |
| Bollinger JM, Bridges JF, Mohamed A, Kaufman D. Public preferences for the return of research results in genetic research: a conjoint analysis. Genetics in Medicine. 2014;16(12):932–9. | Quant | | | 1,515 | | USA | Own | | | X | | | | X | X | X | | |
| Bollinger JM, Scott J, Dvoskin R, Kaufman D. Public preferences regarding the return of individual genetic research results: findings from a qualitative focus group study. Genetics in Medicine. 2012;14 (4):451–7. | Qual | 89 | | | | USA | Own | | | X | | | | | X | X | | |
| Bradbury AR, Patrick-Miller L, Egleston BL, Maxwell KN, DiGiovanni L, Brower J, et al. Returning individual genetic research results to research participants: uptake and outcomes among patients with breast cancer. Precision Oncology. 2018;2:1–24. | Quant | | | 107 | | USA | Own | | | X | | | | X | | X | | X |
| Breitkopf CR, Petersen GM, Wolf SM, Chaffee KG, Robinson ME, Gordon DR, et al. Preferences regarding Rreturn of genomic results to relatives of research rarticipants, including after participant death: empirical results from a cancer biobank. Journal of Law, Medicine & Ethics. 2015;43(3):464–75. | Quant | | | 3,630 | | USA | Own & Relatives' | | X | | | | | X | X | X | | |

*(Continued)*

**Table 2.** (Continued)

| Full Citation | Data Type | Participants by Method | | | | Country | Results (to be) Received | Results (to be) Given | Context | | Stakeholders | | | | | Decision Type | | |
|---|---|---|---|---|---|---|---|---|---|---|---|---|---|---|---|---|---|---|
| | | Focusgroups | Interviews | Survey | Other | | | | Biobanks | Research | HCPs | Review Boards | Researchers | Research Participants | Publics | Hypothetical/Policy/Practice | Decided | Given/Received |
| Brothers KB, East KM, Kelley WV, Wright MF, Westbrook MJ, Rich CA, et al. Eliciting preferences on secondary findings: the Preferences Instrument for Genomic Secondary Results. Genetics in Medicine. 2017;19(3):337–44. | Mixed | 110 | 10 | | | USA | Own & Child's | | | X | | | | X | | | X | |
| Bui ET, Anderson NK, Kassem L, McMahon FJ. Do participants in genome sequencing studies of psychiatric disorders wish to be informed of their results? A survey study. PLoS ONE. 2014;9(7):e101111. | Quant | | | 58 | | USA | Own | | | X | | | | X | | X | | |
| Burnett-Hartman AN, Blum-Barnett E, Carroll NM, Madrid SD, Jonas C, Janes K, et al. Return of research-related genetic test results and genetic discrimination concerns: facilitators and barriers of genetic research participation in diverse groups. Public Health Genomics. 2020;23(1–2):59–68. | Quant | | | 10,369 | | USA | Own | | | X | | | | X | | X | | |
| Byrjalsen A, Stoltze U, Wadt K, Hjalgrim LL, Gerdes AM, Schmiegelow K, et al. Pediatric cancer families' participation in whole-genome sequencing research in Denmark: parent perspectives. European Journal of Cancer Care. 2018;27(6):e12877. | Qual | | 17 | | 15 | DNK | Child's | | | X | | | | X | | | X | |
| Cacioppo CN, Chandler AE, Towne MC, Beggs AH, Holm IA. Expectation versus reality: the impact of utility on emotional outcomes after returning individualized genetic research results in pediatric rare disease research, a qualitative interview study. PLoS ONE. 2016;11(4):e0153597. | Mixed | | 9 | 9 | | USA | Child's | | | X | | | | X | | | | X |
| Cadigan RJ, Michie M, Henderson G, Davis AM, Beskow LM. The meaning of genetic research results: reflections from individuals with and without a known genetic disorder. Journal of Empirical Research on Human Research Ethics. 2011;6(4):30–40. | Qual | | 24 | | | USA | Own | | X | X | | | | X | | | | X |
| Cakici JA, Dimmock DP, Caylor SA, Gaughran M, Clarke C, Triplett C, et al. A prospective study of parental perceptions of rapid whole-genome and -exome sequencing among seriously ill infants. American Journal of Human Genetics. 2020;107(5):953–962. | Quant | | | 161 | | USA | Child's | | | X | | | | X | | | | X |
| Cassidy MR, Roberts JS, Bird TD, Steinbart EJ, Cupples LA, Chen CA, et al. Comparing test-specific distress of susceptibility versus deterministic genetic testing for Alzheimer's disease. Alzheimer's & Dementia. 2008;4(6):406–13. | Quant | | | 123 | | USA | Own | | | X | | | | X | | | | X |

(Continued)

**Table 2.** (Continued)

| Full Citation | Data Type | Focusgroups | Interviews | Survey | Other | Country | Results (to be) Received | Results (to be) Given | Biobanks | Research | HCPs | Reivew Boards | Researchers | Research Participants | Publics | Hypothetical/Policy/Practice | Decided | Given/Received |
|---|---|---|---|---|---|---|---|---|---|---|---|---|---|---|---|---|---|---|
| Chao S, Roberts JS, Marteau TM, Silliman R, Cupples LA, Green RC. Health behavior changes after genetic risk assessment for Alzheimer disease: The REVEAL Study. Alzheimer disease and associated disorders. 2008;22(1):94. | Quant | | | 147 | | USA | Own | | | X | | | | X | | | | X |
| Christensen KD, Karlawish J, Roberts JS, Uhlmann WR, Harkins K, Wood EM, et al. Disclosing genetic risk for Alzheimer's dementia to individuals with mild cognitive impairment. Alzheimer's & Dementia: Translational Research & Clinical Interventions. 2020;6 (1):e12002. | Quant | | | 114 | | USA | Own | | | X | | | | X | | | | X |
| Christensen KD, Roberts JS, Shalowitz DI, Everett JN, Kim SY, Raskin L, et al. Disclosing individual CDKN2A research results to melanoma survivors: interest, impact, and demands on researchers. Cancer Epidemiology and Prevention Biomarkers. 2011;20(3):522–9. | Quant | | | 19 | | USA | Own | | | X | | | | X | | | | X |
| Christensen KD, Roberts JS, Whitehouse PJ, Royal CD, Obisesan TO, Cupples LA, et al. Disclosing pleiotropic effects during genetic risk assessment for Alzheimer disease: a randomized trial. Annals of Internal Medicine. 2016;164(3):155–63. | Quant | | | 257 | | USA | Own | | | X | | | | X | | | | X |
| Christensen KD, Savage SK, Huntington NL, Weitzman ER, Ziniel SI, Bacon PL, et al. Preferences for the return of individual results from research on pediatric biobank samples. Journal of Empirical Research on Human Research Ethics. 2017;12(2):97–106. | Quant | | | 1,027 | | USA | Child's | | X | | | | | X | | X | | |
| Christensen KD, Uhlmann WR, Roberts JS, Linnenbringer E, Whitehouse PJ, Royal CD, et al. A randomized controlled trial of disclosing genetic risk information for Alzheimer disease via telephone. Genetics in Medicine. 2018;20(1):132–41. | Quant | | | 257 | | USA | Own | | | X | | | | X | | | | X |
| Cooke Bailey JN, Crawford DC, Goldenberg A, Slaven A, Pencak J, Schachere M, et al. Willingness to participate in a national precision medicine cohort: attitudes of chronic kidney disease patients at a Cleveland public hospital. Journal of Personalized Medicine. 2018;8 (3):21. | Quant | | | 103 | | USA | Own | | X | X | | | | X | | X | | |
| Coors ME, Raymond KM, McWilliams SK, Hopfer CJ, Mikulich-Gilbertson SK. Adolescent perspectives on the return of individual results in genomic addiction research. Psychiatric Genetics. 2015;25 (3):127–30. | Quant | | | 429 | | USA | Own | | X | X | | | | X | | X | | |

*(Continued)*

**Table 2.** (Continued)

| Full Citation | Data Type | Focusgroups | Interviews | Survey | Other | Country | Results (to be) Received | Results (to be) Given | Biobanks | Research | HCPs | Reivew Boards | Researchers | Research Participants | Publics | Hypothetical/Policy/Practice | Decided | Given/Received |
|---|---|---|---|---|---|---|---|---|---|---|---|---|---|---|---|---|---|---|
| Daack-Hirsch S, Driessnack M, Hanish A, Johnson VA, Shah LL, Simon CM, et al. 'Information is information': a public perspective on incidental findings in clinical and research genome-based testing. Clinical Genetics. 2013;84(1):11–8. | Qual | 54 | 9 | | | USA | Own & Child's | | | X | | | | | X | X | | |
| De S, Tringham M, Hopia A, Tahvonen R, Pietila AM and Vahakangas K. Ethical aspects of genotype disclosure: perceptions of participants in a nutrigenetic study in Finland. Public Health Genomics. 2021;24(1–2):33–43. | Quant | | | 250 | | FIN | Own | | | X | | | | X | | X | | X |
| Dheensa S, Samuel G, Lucassen AM, Farsides B. Towards a national genomics medicine service: the challenges facing clinical-research hybrid practices and the case of the 100,000 genomes project. Journal of Medical Ethics. 2018;44(6):397–403. | Qual | | 20 | | | GBR | | Adults | | X | | | | X | | X | | |
| Dressler LG, Smolek S, Ponsaran R, Markey JM, Starks H, Gerson N, et al. IRB perspectives on the return of individual results from genomic research. Genetics in Medicine. 2012;14(2):215–22. | Qual | | 31 | | | USA | | Adults | | X | | X | | | | X | | |
| Driessnack M, Daack-Hirsch S, Downing N, Hanish A, Shah LL, Alasagheirin M, et al. The disclosure of incidental genomic findings: an "ethically important moment" in pediatric research and practice. Journal of Community Genetics. 2013;4(4):435–44. | Qual | 54 | 112 | | | USA | Child's | Children | | X | X | X | X | | X | X | | |
| Dye DE, Youngs L, McNamara B, Goldblatt J, O'Leary P. The disclosure of genetic information: a human research ethics perspective. Journal of Bioethical Inquiry. 2010;7(1):103–9. | Qual | | | | 29 | AUS | | Adults & Relatives | | X | X | X | X | | X | X | | |
| Edwards K, Goodman D, Johnson C, Wenzel L, Condit C, Bowen D. Controversies among cancer registry participants, genomic researchers, and Institutional Review Boards about returning participants' genomic results. Public Health Genomics. 2018;21(1–2):18–26. | Quant | | | 1,009 | | USA | Own | | | X | X | X | | X | | X | | |
| Edwards K, Lemke A, Trinidad S, Lewis S, Starks H, Griffin MQ, et al. Attitudes toward genetic research review: results from a survey of human genetics researchers. Public Health Genomics. 2011;14(6):337–45. | Quant | | | 351 | | INT'L | | Adults | | X | | X | X | | | X | | |
| Facio FM, Brooks S, Loewenstein J, Green S, Biesecker LG, Biesecker BB. Motivators for participation in a whole-genome sequencing study: implications for translational genomics research. European Journal of Human Genetics. 2011;19(12):1213–7. | Mixed | | | 322 | 322 | USA | Own | | | X | | | | X | | X | | |

*(Continued)*

**Table 2.** (Continued)

| Full Citation | Data Type | Focusgroups | Interviews | Survey | Other | Country | Results (to be) Received | Results (to be) Given | Biobanks | Research | HCPs | Review Boards | Researchers | Research Participants | Publics | Hypothetical/Policy/Practice | Decided | Given/Received |
|---|---|---|---|---|---|---|---|---|---|---|---|---|---|---|---|---|---|---|
| Facio FM, Eidem H, Fisher T, Brooks S, Linn A, Kaphingst KA, et al. Intentions to receive individual results from whole-genome sequencing among participants in the ClinSeq study. European Journal of Human Genetics. 2013;21(3):261–5. | Mixed | | | 311 | | USA | Own | | | X | | | | X | | X | | |
| Fernandez CV, Bouffet E, Malkin D, Jabado N, O'Connell C, Avard D, et al. Attitudes of parents toward the return of targeted and incidental genomic research findings in children. Genetics in Medicine. 2014;16(8):633–40. | Quant | | | 362 | | CAN | Child's & Relatives' | | | X | | | | X | | X | | |
| Fernandez CV, O'Connell C, Ferguson M, Orr AC, Robitaille JM, Knoppers BM, et al. Stability of attitudes to the ethical issues raised by the return of incidental genomic research findings in children: a follow-up study. Public Health Genomics. 2015;18(5):299–308. | Quant | | | 149 | | CAN | Own & Child's | | | | | | | | | | X | X |
| Fernandez CV, O'Rourke PP, Beskow LM. Canadian research ethics board leadership attitudes to the return of genetic research results to individuals and their families. The Journal of Law, Medicine & Ethics. 2015;43(3):514–22. | Quant | | | 22 | | CAN | | Adults | | X | | X | | | | X | | |
| Fernandez CV, Strahlendorf C, Avard D, Knoppers BM, O'Connell C, Bouffet E, et al. Attitudes of Canadian researchers toward the return to participants of incidental and targeted genomic findings obtained in a pediatric research setting. Genetics in Medicine. 2013;15(7):558–64. | Quant | | | 74 | | CAN | | Adults & Children & Relatives | | X | | | X | | | X | | |
| Ferriere M, Van Ness B. Return of individual research results and incidental findings in the clinical trials cooperative group setting. Genetics in Medicine. 2012;14(4):411–6. | Quant | | | 10 | | USA | | Adults | X | X | | | X | | | X | | |
| Fiallos K, Applegate C, Mathews DJ, Bollinger J, Bergner AL, James CA. Choices for return of primary and secondary genomic research results of 790 members of families with Mendelian disease. European Journal of Human Genetics. 2017;25(5):530–7. | Quant | | | | 790 | USA | Own & Child's & Relatives' | | | X | | | | X | | | X | |
| Fleming J, Critchley C, Otlowski M, Stewart C, Kerridge I. Attitudes of the general public towards the disclosure of individual research results and incidental findings from biobank genomic research in AUS. Internal Medicine Journal. 2015;45(12):1274–9. | Quant | | | 800 | | AUS | Own | | X | | | | | | X | X | | |
| Fong M, Braun KL, Chang RM. Native Hawaiian preferences for informed consent and disclosure of results from genetic research. Journal of Cancer Education. 2006;21(suppl 1):S47-S52. | Quant | | | 429 | | USA | Own | | X | | | | | | X | X | | |

(Continued)

Table 2. (Continued)

| Full Citation | Data Type | Participants by Method | | | | Country | Results (to be) Received | Results (to be) Given | Context | | Stakeholders | | | | | Decision Type | | |
|---|---|---|---|---|---|---|---|---|---|---|---|---|---|---|---|---|---|---|
| | | Focusgroups | Interviews | Survey | Other | | | | Biobanks | Research | HCPs | Reivew Boards | Researchers | Research Participants | Publics | Hypothetical/Policy/Practice | Decided | Given/Received |
| Gaieski JB, Patrick-Miller L, Egleston BL, Maxwell KN, Walser S, DiGiovanni L, et al. Research participants' experiences with return of genetic research results and preferences for web-based alternatives. Molecular Genetics and Genomic Medicine. 2019;7(9):e898. | Mixed | | | 88 | | USA | Own | | | X | | | | X | | | | X |
| Gliwa C, Yurkiewicz IR, Lehmann LS, Hull SC, Jones N, Berkman BE. Institutional review board perspectives on obligations to disclose genetic incidental findings to research participants. Genetics in Medicine. 2016;18(7):705–11. | Quant | | | 796 | | USA | | Adults | | X | | X | | | | X | | |
| Goblar J, Roe CM, Selsor NJ, Gabel MJ, Morris JC. Attitudes of research participants and the general public regarding disclosure of Alzheimer disease research results. JAMA Neurology. 2015;72(12):1484–90. | Quant | | | 219 | | USA | Own | | | X | | | | X | X | X | | |
| Goodman D, Johnson CO, Bowen D, Smith M, Wenzel L, Edwards K. De-identified genomic data sharing: the research participant perspective. Journal of community genetics. 2017;8(3):173–81. | Quant | | | 450 | | USA | Own | | | X | | | | X | X | X | | |
| Goodman JL, Amendola LM, Horike-Pyne M, Trinidad SB, Fullerton SM, Burke W, et al. Discordance in selected designee for return of genomic findings in the event of participant death and estate executor. Molecular Genetics & Genomic Medicine. 2017;5(2):172–6. | Quant | | | 61 | | USA | Own | | | X | | | | X | | | X | |
| Gordon DR, Radecki Breitkopf C, Robinson M, Petersen WO, Egginton JS, Chaffee KG, et al. Should Researchers Offer Results to Family Members of Cancer Biobank Participants? A Mixed-Methods Study of Proband and Family Preferences. AJOB Empirical Bioethics. 2019;10(1):1–22. | Mixed | | 51 | 1,903 | | USA | Relatives' | | X | | | | | X | X | X | | |
| Graves K, Sinicrope P, Esplen M, Peterson S, Patten C, Lowery J, et al. Communication of genetic test results to family and health-care providers following disclosure of research results. Genetics in Medicine. 2014;16(4):294–301. | Quant | | | 107 | | USA | Own | | | X | | | | X | | | X | X |
| Green RC, Roberts JS, Cupples LA, Relkin NR, Whitehouse PJ, Brown T, et al. Disclosure of APOE genotype for risk of Alzheimer's disease. New England Journal of Medicine. 2009;361(3):245–54. | Quant | | | 162 | | USA | Own | | | X | | | | X | | | | X |
| Grill JD, Bateman RJ, Buckles V, Oliver A, Morris JC, Masters CL, et al. A survey of attitudes toward clinical trials and genetic disclosure in autosomal dominant Alzheimer's disease. Alzheimer's Research & Therapy. 2015;7(1):50. | Quant | | | 80 | | USA | Own | | | X | | | | X | | X | | |

(Continued)

**Table 2.** (Continued)

| Full Citation | Data Type | Focusgroups | Interviews | Survey | Other | Country | Results (to be) Received | Results (to be) Given | Biobanks | Research | HCPs | Reivew Boards | Researchers | Research Participants | Publics | Hypothetical/Policy/Practice | Decided | Given/Received |
|---|---|---|---|---|---|---|---|---|---|---|---|---|---|---|---|---|---|---|
| | | | | | | | | | Context | | Stakeholders | | | | | Decision Type | | |
| Groisman IJ, Godard B. Impact of next generation sequencing on the organization and funding of returning research results: survey of Canadian Research Ethics Boards members. PLoS ONE. 2016;11(5):e0154965. | Quant | | | 81 | | CAN | | Adults & Children | | X | | X | | | | X | | |
| Guo SH, Goodman M, Kaphingst K. Comparing preferences for return of genome sequencing results assessed with rating and ranking items. Journal of Genetic Counseling. 2020;29(1):131–4. | Quant | | | 1,045 | | USA | Own | | | X | | | | X | | X | | |
| Hallowell N, Alsop K, Gleeson M, Crook A, Plunkett L, Bowtell D, et al. The responses of research participants and their next of kin to receiving feedback of genetic test results following participation in the AUSn Ovarian Cancer Study. Genetics in Medicine. 2013;15(6):458–465. | Qual | | 25 | | | AUS | Own & Relatives' | | | X | | | | X | | | | X |
| Halverson CM, Ross LF. Attitudes of African-American parents about biobank participation and return of results for themselves and their children. Journal of Medical Ethics. 2012;38(9):561–6. | Mixed | | | 45 | 45 | USA | Own & Child's | | X | | | | | | X | X | | |
| Halverson CM, Ross LF. Engaging African-Americans about biobanks and the return of research results. Journal of Community Genetics. 2012;3(4):275–83. | Mixed | | | 45 | 45 | USA | Own & Child's | | X | | | | | | X | X | | |
| Halverson CME, Jones SH, Novak L, Simpson C, Edwards DRV, Zhao SK, et al. What results should be returned from opportunistic screening in translational research? Journal of Personalized Medicine. 2020;10(1):1–13. | Mixed | | 36 | 675 | | USA | Own | Adults | | X | X | | | X | | | | X |
| Harris ED, Ziniel SI, Amatruda JG, Clinton CM, Savage SK, Taylor PL, et al. The beliefs, motivations, and expectations of parents who have enrolled their children in a genetic biorepository. Genetics in Medicine. 2012;14(3):330–7. | Qual | 19 | | | | USA | Child's | | X | | | | | X | | X | | |
| Hart MR, Biesecker BB, Blout CL, Christensen KD, Amendola LM, Bergstrom KL, et al. Secondary findings from clinical genomic sequencing: prevalence, patient perspectives, family history assessment, and health-care costs from a multisite study. Genetics in Medicine. 2019;21(5):1100–10. | Qual | | 18 | | | USA | Own & Child's | | | X | | | | X | | | | X |
| Hartz SM, Olfson E, Culverhouse R, Cavazos-Rehg P, Chen L-S, DuBois J, et al. Return of individual genetic results in a high-risk sample: enthusiasm and positive behavioral change. Genetics in Medicine. 2015;17(5):374–379. | Quant | | | 43 | | USA | Own | | | X | | | | X | | | | X |

(Continued)

Table 2. (Continued)

| Full Citation | Data Type | Focusgroups | Interviews | Survey | Other | Country | Results (to be) Received | Results (to be) Given | Biobanks | Research | HCPs | Review Boards | Researchers | Research Participants | Publics | Hypothetical/Policy/Practice | Decided | Given/Received |
|---|---|---|---|---|---|---|---|---|---|---|---|---|---|---|---|---|---|---|
| Haukkala A, Kujala E, Alha P, Salomaa V, Koskinen S, Swan H, et al. The return of unexpected research results in a biobank study and referral to health care for heritable long QT syndrome. Public Health Genomics. 2013;16(5):241–50. | Mixed | | 5 | 17 | | FIN | Own | | X | | | | | X | | | | X |
| Heaney C, Tindall G, Lucas J, Haga SB. Researcher practices on returning genetic research results. Genetic Testing and Molecular Biomarkers. 2010;14(6):821–7. | Quant | | | 105 | | USA | | Adults & Children | | X | | | X | | | | | X |
| Henrikson NB, Scrol A, Leppig KA, Ralston JD, Larson EB and Jarvik GP. Preferences of biobank participants for receiving actionable genomic test results: results of a recontacting study. Genetics in Medicine, 2021;23(6):1163–1166. | Quant | | | | 123 | USA | Own | | X | | | | | X | | | X | |
| Hiratsuka VY, Beans JA, Blanchard JW, Reedy J, Blacksher E, Lund JR, et al. An Alaska Native community's views on genetic research, testing, and return of results: results from a public deliberation. PLoS ONE. 2020;15(3):e0229540. | Mixed | | 19 | 19 | | USA | Own | | | X | | | | X | | X | | |
| Hoell C, Wynn J, Rasmussen LV, Marsolo K, Aufox SA, Chung WK, et al. Participant choices for return of genomic results in the eMERGE Network. Genetics in Medicine, 2020;22(11):1821–1829. | Quant | | | | 4,664 | USA | Own | | | X | | | | X | | | X | |
| Holm IA, Iles BR, Ziniel SI, Bacon PL, Savage SK, Christensen KD, et al. Participant satisfaction with a preference-setting tool for the return of individual research results in pediatric genomic research. Journal of Empirical Research on Human Research Ethics. 2015;10 (4):414–26. | Quant | | | 2,718 | | USA | Child's | | X | | | | | X | | X | | |
| Holzer K, Culhane-Pera KA, Straka RJ, Wen YF, Lo M, Lee K, et al. Hmong participants' reactions to return of individual and community pharmacogenetic research results: "A positive light for our community". Journal of Community Genetics, 2021;12(1):53–65. | Qual | 24 | | | | USA | Own & Community | | | X | | | | X | | | | X |
| Hyams T, Bowen DJ, Condit C, Grossman J, Fitzmaurice M, Goodman D, et al. Views of cohort study participants about returning research results in the context of precision medicine. Public Health Genomics. 2016;19(5):269–75. | Qual | | 30 | | | USA | Own | | X | | | | | X | | X | | |
| Hylind R, Smith M, Rasmussen-Torvik L, Aufox S. Great expectations: patient perspectives and anticipated utility of non-diagnostic genomic-sequencing results. Journal of Community Genetics. 2018;9(1):19–26. | Qual | | 14 | | | USA | Own | | | X | | | | X | | | X | |

(Continued)

Table 2. (Continued)

| Full Citation | Data Type | Focusgroups | Interviews | Survey | Other | Country | Results (to be) Received | Results (to be) Given | Biobanks | Research | HCPs | Review Boards | Researchers | Research Participants | Publics | Hypothetical/Policy/Practice | Decided | Given/Received |
|---|---|---|---|---|---|---|---|---|---|---|---|---|---|---|---|---|---|---|
| Jelsig AM, Qvist N, Brusgaard K, Ousager LB. Research participants in NGS studies want to know about incidental findings. European Journal of Human Genetics. 2015;23(10):1423–6. | Quant | | | | 127 | DNK | Own | | | X | | | | X | | | X | |
| Joffe S, Sellers DE, Ekunwe L, Antoine-Lavigne D, McGraw S, Levy D, et al. Preferences for Return of Genetic Results among Participants in the Jackson Heart Study and Framingham Heart Study. Circulation: Genomic and Precision Medicine. 2019;12 (12):552–60. | Quant | | | 2,075 | | USA | Own | | | X | | | | X | | X | | |
| Kaphingst K, Janoff J, Harris L, Emmons K. Views of female breast cancer patients who donated biologic samples regarding storage and use of samples for genetic research. Clinical Genetics. 2006;69 (5):393–8. | Qual | 12 | 14 | | | USA | Own | | X | | | | | X | | X | | |
| Kauffman TL, Irving SA, Leo MC, Gilmore MJ, Himes P, McMullen CK, et al. The NextGen Study: patient motivation for participation in genome sequencing for carrier status. Molecular genetics & genomic medicine. 2017;5(5):508–15. | Quant | | | 310 | | USA | Own | | | X | | | | X | | X | X | |
| Kaufman D, Geller G, Leroy L, Murphy J, Scott J, Hudson K. Ethical implications of including children in a large biobank for genetic-epidemiologic research: a qualitative study of public opinion. American Journal of Medical Genetics—Part C. 2008;148 (1):31–9. | Qual | 141 | | | | USA | Child's | | X | | | | | | X | X | | |
| Kaufman D, Murphy J, Scott J, Hudson K. Subjects matter: a survey of public opinions about a large genetic cohort study. Genetics in Medicine. 2008;10(11):831–9. | Quant | | | 4,659 | | USA | Own | | X | | | | | | X | X | | |
| Kaufman DJ, Baker R, Milner LC, Devaney S, Hudson KL. A survey of US adults' opinions about conduct of a nationwide Precision Medicine Initiative® cohort study of genes and environment. PLoS ONE. 2016;11(8):e0160461. | Quant | | | 2,601 | | USA | Own | | | X | | | | | X | X | | |
| Khodyakov D, Mendoza-Graf A, Berry S, Nebeker C, Bromley E. Return of value in the new era of biomedical research—one size will not fit all. AJOB Empirical Bioethics. 2019;10(4):265–75. | Qual | | 44 | | | USA | | Adults | | X | | | | | | X | | |
| Kleiderman E, Avard D, Besso A, Ali-Khan S, Sauvageau G, Hébert J. Disclosure of incidental findings in cancer genomic research: investigators' perceptions on obligations and barriers. Clinical Genetics. 2015;88(4):320–6. | Qual | | 20 | | | CAN | | Adults | X | X | X | | X | | | X | | |

(Continued)

Table 2. (Continued)

| Full Citation | Data Type | Focusgroups | Interviews | Survey | Other | Country | Results (to be) Received | Results (to be) Given | Biobanks | Research | HCPs | Reivew Boards | Researchers | Research Participants | Publics | Hypothetical/Policy/Practice | Decided | Given/Received |
|---|---|---|---|---|---|---|---|---|---|---|---|---|---|---|---|---|---|---|
| Kleiderman E, Knoppers BM, Fernandez CV, Boycott KM, Ouellette G, Wong-Rieger D, et al. Returning incidental findings from genetic research to children: views of parents of children affected by rare diseases. Journal of Medical Ethics. 2014;40(10):691–6. | Qual | 6 | 9 | | | CAN | Child's | | | X | | | | X | | X | | |
| Klitzman R, Appelbaum PS, Fyer A, Martinez J, Buquez B, Wynn J, et al. Researchers' views on return of incidental genomic research results: qualitative and quantitative findings. Genetics in Medicine. 2013;15(11):888–95. | Mixed | | 28 | 241 | | USA | | Adults & Children | | X | X | | X | | | X | | |
| Kostick K, Pereira S, Brannan C, Torgerson L, Lazaro-Munoz G. Psychiatric genomics researchers' perspectives on best practices for returning results to individual participants. Genetics in Medicine. 2020;22(2):345–52. | Qual | | 39 | | | INT'L | | Adults | | X | X | | X | | | X | | |
| Kostick KM, Brannan C, Pereira S, Lazaro-Munoz G. Psychiatric genetics researchers' views on offering return of results to individual participants. American Journal of Medical Genetics—Part B. 2019;180(8):589–600. | Qual | | 39 | | | INT'L | | Adults | | X | X | | X | | | X | | |
| Kranendonk EJ, Ploem MC, Hennekam RC. Regulating biobanking with children's tissue: a legal analysis and the experts' view. European Journal of Human Genetics. 2016;24(1):30–6. | Qual | | 17 | | | NLD | | Children | X | | X | | | | | X | | |
| LaRusse S, Roberts JS, Marteau TM, Katzen H, Linnenbringer EL, Barber M, et al. Genetic susceptibility testing versus family history-based risk assessment: Impact on perceived risk of Alzheimer disease. Genetics in Medicine. 2005;7(1):48–53. | Mixed | | | 56 | | USA | Own | | | X | | | | X | | | | X |
| Lawal TA, Lewis KL, Johnston JJ, Heidlebaugh AR, Ng D, Gaston-Johansson FG, et al. Disclosure of cardiac variants of uncertain significance results in an exome cohort. Clinical Genetics. 2018;93(5):1022–9. | Mixed | | | 79 | | USA | Own | | | X | | | | X | | | | X |
| Lazaro-Munoz G, Torgerson L and Pereira S. Return of results in a global survey of psychiatric genetics researchers: practices, attitudes, and knowledge. Genetics in Medicine, 2021;23(2):298–305. | Quant | | | 407 | | INT'L | | Adults | | X | | | X | | | X | | X |
| Lazaro-Munoz G, Torgerson L, Smith HS and Pereira S. Perceptions of best practices for return of results in an international survey of psychiatric genetics researchers. European Journal of Human Genetics, 2021;29(2):231–240. | Quant | | | 407 | | INT'L | | Adults | | X | | | X | | | X | | X |

(Continued)

**Table 2.** (Continued)

| Full Citation | Data Type | Focusgroups | Interviews | Survey | Other | Country | Results (to be) Received | Results (to be) Given | Biobanks | Research | HCPs | Review Boards | Researchers | Research Participants | Publics | Hypothetical/Policy/Practice | Decided | Given/Received |
|---|---|---|---|---|---|---|---|---|---|---|---|---|---|---|---|---|---|---|
| Leitsalu L, Alavere H, Jacquemont S, Kolk A, Maillard AM, Reigo A, et al. Reporting incidental findings of genomic disorder-associated copy number variants to unselected biobank participants. Personalized Medicine, 2016;13(4):303–314. | Quant | | | 5 | | EST | Own | | X | | | | | X | | | | X |
| Leitsalu L, Palover M, Sikka TT, Reigo A, Kals M, Parn K, et al. Genotype-first approach to the detection of hereditary breast and ovarian cancer risk, and effects of risk disclosure to biobank participants. European Journal of Human Genetics, 2021;29(3):471–481. | Quant | | | 22 | | EST | Own | | X | | | | | X | | | | X |
| Lemke AA, Halverson C, Ross LF. Biobank participation and returning research results: perspectives from a deliberative engagement in South Side Chicago. American Journal of Medical Genetics—Part A. 2012;158(5):1029–37. | Mixed | | | 45 | 45 | USA | Own & Child's | | X | | | | | | X | X | | |
| Leof ER, Zhu X, Rabe KG, McCormick JB, Petersen GM, Breitkopf CR. Pancreatic cancer and melanoma related perceptions and behaviors following disclosure of CDKN2A variant status as a research result. Genetics in Medicine. 2019;21(11):2468–77. | Quant | | | 80 | | USA | Own | | | X | | | | X | | | | X |
| Lewis C, Hammond J, Hill M, Searle B, Hunter A, Patch C, et al. Young people's understanding, attitudes and involvement in decision-making about genome sequencing for rare diseases: A qualitative study with participants in the UK 100,000 Genomes Project. European Journal of Medical Genetics, 2020;63(11):1–7. | Qual | | 27 | | | GBR | Own | | | X | | | | X | | | X | |
| Lewis C, Sanderson S, Hill M, Patch C, Searle B, Hunter A, et al. Parents' motivations, concerns and understanding of genome sequencing: a qualitative interview study. European Journal of Human Genetics. 2020;28(7):874–884. | Quant | | | 37 | | GBR | Child's | | | X | | | | X | | | X | |
| Lewis KL, Heidlebaugh AR, Epps S, Han PK, Fishler KP, Klein WM, et al. Knowledge, motivations, expectations, and traits of an African, African-American, and Afro-Caribbean sequencing cohort and comparisons to the original ClinSeq® cohort. Genetics in Medicine. 2019;21(6):1355–62. | Mixed | | | 467 | | USA | Own | | | X | | | | X | | | X | |
| Lewis KL, Hooker GW, Connors PD, Hyams TC, Wright MF, Caldwell S, et al. Participant use and communication of findings from exome sequencing; a mixed-methods study. Genetics in Medicine. 2016;18(6):577–83. | Mixed | | 29 | 29 | | USA | Own | | | X | | | | X | | | | X |

*(Continued)*

**Table 2.** (Continued)

| Full Citation | Data Type | Focusgroups | Interviews | Survey | Other | Country | Results (to be) Received | Results (to be) Given | Biobanks | Research | HCPs | Review Boards | Researchers | Research Participants | Publics | Hypothetical/Policy/Practice | Decided | Given/Received |
|---|---|---|---|---|---|---|---|---|---|---|---|---|---|---|---|---|---|---|
| Lindor NM, Schahl KA, Johnson KJ, Hunt KS, Mensink KA, Wieben ED, et al. Whole-exome sequencing of 10 scientists: evaluation of the process and outcomes. Mayo Clinic Proceedings. 2015;90(10):1327–37. | Mixed | | | 10 | 3 | USA | Own | | | X | | | X | | | | | X |
| Linnenbringer E, Roberts JS, Hiraki S, Cupples LA, Green RC. "I know what you told me, but this is what I think": perceived risk of Alzheimer disease among individuals who accurately recall their genetics-based risk estimate. Genetics in Medicine. 2010;12(4):219–27. | Quant | | | 246 | | USA | Own | | | X | | | | X | | | | X |
| Loud JT, Bremer RC, Mai PL, Peters JA, Giri N, Stewart DR, et al. Research participant interest in primary, secondary, and incidental genomic findings. Genetics in Medicine. 2016;18(12):1218–25. | Quant | | | 507 | | USA | Own | | | X | | | | X | | | X | |
| Love-Nichols J, Uhlmann WR, Arscott P, Willer C, Hornsby W and Roberts JS. A survey of aortic disease biorepository participants' preferences for return of research genetic results. Journal of Genetic Counseling. 2020;30(3):645–655. | Quant | | | 225 | | USA | Own | | X | | | | | X | | X | | |
| Lynch J, Hines J, Theodore S, Mitchell M. Lay attitudes toward trust, uncertainty, and the return of pediatric research results in biobanking. AJOB Empirical Bioethics. 2016;7(3):160–6. | Mixed | 40 | | 40 | | USA | Child's | | X | | | | | X | X | X | | |
| Mackley MP, Blair E, Parker M, Taylor JC, Watkins H, Ormondroyd E. Views of rare disease participants in a UK whole-genome sequencing study towards secondary findings: a qualitative study. European Journal of Human Genetics. 2018;26(5):652–9. | Qual | | 16 | | | GBR | Own | | | X | | | | X | | | X | |
| Marsh V, Kombe F, Fitzpatrick R, Williams TN, Parker M and Molyneux S. Consulting communities on feedback of genetic findings in international health research: sharing sickle cell disease and carrier information in coastal Kenya. BMC Medical Ethics, 2013;14(1):1–13. | Qual | 63 | | | | KEN | Own & Child's & Relatives' | | | X | | | | | X | X | | |
| Marteau TM, Roberts S, LaRusse S, Green RC. Predictive genetic testing for Alzheimer's disease: impact upon risk perception. Risk Analysis. 2005;25(2):397–404. | Quant | | | 149 | | USA | Own | | | X | | | | X | | | | X |
| Master Z, Claudio JO, Rachul C, Wang JC, Minden MD, Caulfield T. Cancer patient perceptions on the ethical and legal issues related to biobanking. BMC medical genomics. 2013;6(8):1–10. | Quant | | | 98 | | CAN | Own | | X | | | | | X | | X | | |

(Continued)

**Table 2.** (Continued)

| Full Citation | Data Type | Focusgroups | Interviews | Survey | Other | Country | Results (to be) Received | Results (to be) Given | Biobanks | Research | HCPs | Review Boards | Researchers | Research Participants | Publics | Hypothetical/Policy/Practice | Decided | Given/Received |
|---|---|---|---|---|---|---|---|---|---|---|---|---|---|---|---|---|---|---|
| Matsen CB, Lyons S, Goodman MS, Biesecker BB, Kaphingst KA. Decision role preferences for return of results from genome sequencing amongst young breast cancer patients. Patient Education and Counseling. 2019;102(1):155–61. | Quant | | | 1,080 | | USA | Own | | | X | | | | X | | X | | |
| Matsui K, Lie RK, Kita Y, Ueshima H. Ethics of future disclosure of individual risk information in a genetic cohort study: A survey of donor preferences. Journal of Epidemiology. 2008;18(5):217–24. | Quant | | | 1,857 | | JPN | Own | | X | | | | | | | | X | |
| McGowan ML, Prows CA, DeJonckheere M, Brinkman WB, Vaughn L, Myers MF. Adolescent and parental attitudes about return of genomic research results: focus group findings regarding decisional preferences. Journal of Empirical Research on Human Research Ethics. 2018;13(4):371–82. | Qual | 33 | | | | USA | Own & Child's | | | | | | | X | X | X | | |
| McGuire AL, Robinson JO, Ramoni RB, Morley DS, Joffe S, Plon SE. Returning genetic research results: study type matters. Personalized Medicine. 2013;10(1):27–34. | Qual | | 35 | | | USA | | Unspecified | | X | | | X | | | | | X |
| McVeigh TP, Sweeney KJ, Kerin MJ, Gallagher DJ. A qualitative analysis of the attitudes of Irish patients towards participation in genetic-based research. Irish Journal of Medical Science. 2016;185(4):825–31. | Quant | | | 351 | | IRL | Own | | X | X | | | | X | X | X | | |
| Meacham MC, Starks H, Burke W, Edwards K. Researcher perspectives on disclosure of incidental findings in genetic research. Journal of Empirical Research on Human Research Ethics. 2010;5(3):31–41. | Qual | | 44 | | | USA | | Unspecified | | X | | | X | | | X | | |
| Meagher KM, Curtis SH, Borucki S, Beck A, Srinivasan T, Cheema A, et al. Communicating unexpected pharmacogenomic results to biobank contributors: a focus group study. Patient Education and Counseling. 2021;104(2):242–249. | Qual | 54 | | | | USA | Own | | X | X | | | X | X | | | | X |
| Meisel S, Wardle J. 'Battling my biology': psychological effects of genetic testing for risk of weight gain. Journal of Genetic Counseling. 2014;23(2):179–86. | Qual | | 18 | | | GBR | Own | | | X | | | | X | | | | X |
| Meulenkamp TM, Gevers SJ, Bovenberg JA, Smets EM. Researchers' opinions towards the communication of results of biobank research: a survey study. European Journal of Human Genetics. 2012;20(3):258–62. | Quant | | | 80 | | NLD | | Adults | X | | | | X | | | X | | |
| Meulenkamp TM, Gevers SK, Bovenberg JA, Koppelman GH, Vlieg AvH, Smets EM. Communication of biobanks' research results: what do (potential) participants want? American Journal of Medical Genetics—Part A. 2010;152(10):2482–92. | Quant | | | 1,163 | | NLD | Own | | X | | | | | X | X | X | | |

*(Continued)*

**Table 2.** (Continued)

| Full Citation | Data Type | Participants by Method | | | | Country | Results (to be) Received | Results (to be) Given | Context | | Stakeholders | | | | | Decision Type | | |
|---|---|---|---|---|---|---|---|---|---|---|---|---|---|---|---|---|---|---|
| | | Focusgroups | Interviews | Survey | Other | | | | Biobanks | Research | HCPs | Review Boards | Researchers | Research Participants | Publics | Hypothetical/Policy/Practice | Decided | Given/Received |
| Michie M, Cadigan RJ, Henderson G, Beskow LM. Am I a control?: Genotype-driven research recruitment and self-understandings of study participants. Genetics in Medicine. 2012;14(12):983–9. | Qual | | 24 | | | USA | Own | | X | X | | | X | | | | | X |
| Middleton A, Morley KI, Bragin E, Firth HV, Hurles ME, Wright CF, et al. Attitudes of nearly 7000 health professionals, genomic researchers and publics toward the return of incidental results from sequencing research. European Journal of Human Genetics. 2016;24(1):21–9. | Quant | | | 6,944 | | INT'L | Own & Child's | Adults & Children | | X | X | | X | | X | X | | |
| Mighton C, Carlsson L, Casalino S, Shickh S, McCuaig L, Joshi E, et al. Quality of life drives patients' preferences for secondary findings from genomic sequencing. European Journal of Human Genetics. 2020;28(9):1178–1186. | Qual | | 31 | | | CAN | Own | | | X | | | X | X | | X | | |
| Miller FA, Giacomini M, Ahern C, Robert JS, De Laat S. When research seems like clinical care: a qualitative study of the communication of individual cancer genetic research results. BMC Medical Ethics. 2008;9(4):1–12. | Qual | | 30 | | | CAN | Own | Adults | | X | X | | X | X | | | | X |
| Miller FA, Hayeems RZ, Bytautas JP. What is a meaningful result? Disclosing the results of genomic research in autism to research participants. European Journal of Human Genetics. 2010;18(8):867–71. | Qual | 34 | 48 | | | INT'L | Own & Child's | Adults & Children | | X | | | X | X | | | X | X |
| Miller FA, Hayeems RZ, Li L, Bytautas JP. One thing leads to another: the cascade of obligations when researchers report genetic research results to study participants. European Journal of Human Genetics. 2012;20(8):837–43. | Quant | | | 343 | | INT'L | | Adults & Children | | X | | | X | X | | X | | |
| Miller IM, Lewis KL, Lawal TA, Ng D, Johnston JJ, Biesecker BB, et al. Health behaviors among unaffected participants following receipt of variants of uncertain significance in cardiomyopathy-associated genes. Genetics in Medicine. 2019;21(3):748–52. | Quant | | | 68 | | USA | Own | | | X | | | | X | | | | X |
| Minion JT, Butcher F, Timpson N, Murtagh MJ. The ethics conundrum in Recall by Genotype (RbG) research: perspectives from birth cohort participants. PLoS ONE. 2018;13(8):e0202502. | Qual | | 53 | | | GBR | Own | | X | | | | | X | | X | | |
| Mitchell C, Ploem C, Retel V, Gevers S and Hennekam R. Experts reflecting on the duty to recontact patients and research participants: why professionals should take the lead in developing guidelines. European Journal of Medical Genetics. 2020;63(2):1–7. | Qual | | 14 | | | INT'L | | Adults | X | X | X | | X | X | | X | | |

(Continued)

**Table 2.** (Continued)

| Full Citation | Data Type | Focusgroups | Interviews | Survey | Other | Country | Results (to be) Received | Results (to be) Given | Biobanks | Research | HCPs | Review Boards | Researchers | Research Participants | Publics | Hypothetical/Policy/Practice | Decided | Given/Received |
|---|---|---|---|---|---|---|---|---|---|---|---|---|---|---|---|---|---|---|
| | | Participants by Method | | | | | | | Context | | Stakeholders | | | | | Decision Type | | |
| Mitchell PB, Ziniel SI, Savage SK, Christensen KD, Weitzman ER, Green RC, et al. Enhancing autonomy in biobank decisions: too much of a good thing? Journal of Empirical Research on Human Research Ethics. 2018;13(2):125–38. | Quant | | | 2,960 | | USA | Child's | | X | | | | | | X | | X | | |
| Moutel G, Duchange N, Raffi F, Sharara LI, Théodorou I, Noël V, et al. Communication of pharmacogenetic research results to HIV-infected treated patients: standpoints of professionals and patients. European Journal of Human Genetics, 2005;13(9):1055–1062. | Mixed | | | 140 | 121 | FRA | Own | Adults | | X | X | | X | X | | X | X | |
| Mozersky J, Hartz S, Linnenbringer E, Levin L, Streitz M, Stock K, et al. Communicating 5-year risk of Alzheimer's disease dementia: development and evaluation of materials that incorporate multiple genetic and biomarker research results. Journal of Alzheimer's Disease, 2021;79(2):559–572. | Qual | | 37 | | | USA | Own | | | X | | | | X | | X | | |
| Murphy J, Scott J, Kaufman D, Geller G, LeRoy L, Hudson K. Public expectations for return of results from large-cohort genetic research. The American Journal of Bioethics. 2008;8(11):36–43. | Qual | 141 | | | | USA | Own | | X | | | | | | X | X | | |
| Mweemba O, Musuku J, Mayosi BM, Parker M, Rutakumwa R, Seeley J, et al. Use of broad consent and related procedures in genomics research: perspectives from research participants in the Genetics of Rheumatic Heart Disease (RHDGen) study in a University Teaching Hospital in ZMB. Global Bioethics. 2019;31(1):184–199. | Qual | | 21 | | | ZMB | Own & Child's | | | X | | | X | X | | X | | |
| Myers MF, Martin LJ, Prows CA. Adolescents' and parents' genomic testing decisions: associations with age, race, and sex. Journal of Adolescent Health. 2020;66(3):288–95. | Quant | | | | 326 | USA | Own & Child's | | | X | | | | X | | | X | |
| Nilsson MP, Emmertz M, Kristoffersson U, Borg Å, Larsson C, Rehn M, et al. Germline mutations in BRCA1 and BRCA2 incidentally revealed in a biobank research study: experiences from re-contacting mutation carriers and relatives. Journal of Community Genetics. 2018;9(3):201–8. | Qual | | 3 | | | SWE | Own | | X | X | | | | X | | | | X |
| O'Daniel J, Haga S. Public perspectives on returning genetics and genomics research results. Public Health Genomics. 2011;14(6):346–55. | Mixed | 100 | | 100 | | USA | Own | | | X | | | | | X | X | | |
| Ormond KE, Cirino AL, Helenowski IB, Chisholm RL, Wolf WA. Assessing the understanding of biobank participants. American Journal of Medical Genetics—Part A. 2009;149(2):188–98. | Mixed | | 109 | 200 | | USA | Own | | X | | | | | X | X | X | | |

*(Continued)*

**Table 2.** (Continued)

| Full Citation | Data Type | Focusgroups | Interviews | Survey | Other | Country | Results (to be) Received | Results (to be) Given | Biobanks | Research | HCPs | Review Boards | Researchers | Research Participants | Publics | Hypothetical/Policy/Practice | Decided | Given/Received |
|---|---|---|---|---|---|---|---|---|---|---|---|---|---|---|---|---|---|---|
| Ormondroyd E, Harper AR, Thomson KL, Mackley MP, Martin J, Penkett CJ, et al. Secondary findings in inherited heart conditions: a genotype-first feasibility study to assess phenotype, behavioural and psychosocial outcomes. European Journal of Human Genetics, 2020;28(11):1486–1496. | Qual | | 10 | | | GBR | Own & Child's | | X | | | | | X | | | | X |
| Ormondroyd E, Mackley MP, Blair E, Craft J, Knight JC, Taylor JC, et al. "Not pathogenic until proven otherwise": perspectives of UK clinical genomics professionals toward secondary findings in context of a genomic medicine multidisciplinary team and the 100,000 Genomes Project. Genetics in Medicine. 2018;20(3):320–8. | Qual | | 19 | | | GBR | | Adults & Children | | X | X | | X | | | X | | |
| Ormondroyd E, Moynihan C, Watson M, Foster C, Davolls S, Ardern-Jones A, et al. Disclosure of genetics research results after the death of the patient participant: a qualitative study of the impact on relatives. Journal of Genetic Counseling. 2007;16(4):527–38. | Qual | | 13 | | | GBR | Relatives' | | | X | | | | X | | | | X |
| Pet DB, Holm IA, Williams JL, Myers MF, Novak LL, Brothers KB, et al. Physicians' perspectives on receiving unsolicited genomic results. Genetics in Medicine. 2019;21(2):311–8. | Qual | | 25 | | | USA | | Adults & Children | | X | X | | | | | X | | |
| Porteri C, Pasqualetti P, Togni E, Parker M. Public's attitudes on participation in a biobank for research: an Italian survey. BMC Medical Ethics. 2014;15(81):1–10. | Quant | | | 142 | | ITA | Own | | X | X | | X | | | X | X | | |
| Radecki Breitkopf C, Wolf SM, Chaffee KG, Robinson ME, Lindor NM, Gordon DR, et al. Attitudes toward return of genetic research results to relatives, including after death: Comparison of cancer probands, blood relatives, and spouse/partners. Journal of Empirical Research on Human Research Ethics. 2018;13(3):295–304. | Quant | | | 1,903 | | USA | Own | | X | X | | | | X | | X | | |
| Raghuram Pillai P, Prows CA, Martin LJ, Myers MF. Decisional conflict among adolescents and parents making decisions about genomic sequencing results. Clinical Genetics. 2020;97(2):312–20. | Quant | | | 326 | | USA | Own & Child's | | | X | | | | X | | X | | |
| Rahm AK, Bailey L, Fultz K, Fan A, Williams JL, Buchanan A, et al. Parental attitudes and expectations towards receiving genomic test results in healthy children. Translational Behavioral Medicine. 2018;8(1):44–53. | Qual | | | | 17 | USA | Child's | | X | | | | | X | | X | | |
| Ralefala D, Kasule M, Wonkam A, Matshaba M and de Vries J. Do solidarity and reciprocity obligations compel African researchers to feedback individual genetic results in genomics research? BMC Medical Ethics, 2020;21(1):1–11. | Qual | 93 | 12 | | | BWA | Own & Child's | | | X | | | | X | | X | | |

*(Continued)*

**Table 2.** (Continued)

| Full Citation | Data Type | Focusgroups | Interviews | Survey | Other | Country | Results (to be) Received | Results (to be) Given | Biobanks | Research | HCPs | Review Boards | Researchers | Research Participants | Publics | Hypothetical/Policy/Practice | Decided | Given/Received |
|---|---|---|---|---|---|---|---|---|---|---|---|---|---|---|---|---|---|---|
| Ramoni RB, McGuire AL, Robinson JO, Morley DS, Plon SE, Joffe S. Experiences and attitudes of genome investigators regarding return of individual genetic test results. Genetics in Medicine. 2013;15(11):882–7. | Quant | | | 200 | | INT'L | | Adults & Children | | X | X | | X | | | | | X |
| Rego S, Dagan-Rosenfeld O, Bivona SA, Snyder MP, Ormond KE. Much ado about nothing: a qualitative study of the experiences of an average-risk population receiving results of exome sequencing. Journal of Genetic Counseling. 2019;28(2):428–37. | Qual | | 12 | | | USA | Own | | | X | | | | X | | | | X |
| Reid AE, Taber JM, Ferrer RA, Biesecker BB, Lewis KL, Biesecker LG, et al. Associations of perceived norms with intentions to learn genomic sequencing results: roles for attitudes and ambivalence. Health Psychology. 2018;37(6):553–61. | Quant | | | 540 | | USA | Own | | | X | | | | X | | | X | |
| Rini C, Khan CM, Moore E, Roche MI, Evans JP, Berg JS, et al. The who, what, and why of research participants' intentions to request a broad range of secondary findings in a diagnostic genomic sequencing study. Genetics in Medicine. 2018;20(7):760–9. | Quant | | | 152 | | USA | Own | | | X | | | | X | | | X | |
| Roberts JS, Cupples LA, Relkin NR, Whitehouse PJ, Green RC, Group RS. Genetic risk assessment for adult children of people with Alzheimer's disease: the Risk Evaluation and Education for Alzheimer's Disease (REVEAL) study. Journal of Geriatric Psychiatry and Neurology. 2005;18(4):250–5. | Quant | | | 162 | | USA | Own | | | X | | | | X | | | | X |
| Roberts JS, Gornick MC, Le LQ, Bartnik NJ, Zikmund-Fisher BJ, Chinnaiyan AM, et al. Next-generation sequencing in precision oncology: patient understanding and expectations. Cancer Medicine. 2019;8(1):227–37. | Quant | | | 297 | | USA | Own | | | X | | | | X | | | | X |
| Roberts JS, Robinson JO, Diamond PM, Bharadwaj A, Christensen KD, Lee KB, et al. Patient understanding of, satisfaction with, and perceived utility of whole-genome sequencing: findings from the MedSeq Project. Genetics in Medicine. 2018;20(9):1069–76. | Quant | | | 202 | | USA | Own | | | X | | | | X | | | | X |
| Roche MI, Griesemer I, Khan CM, Moore E, Lin FC, O'Daniel JM, et al. Factors influencing NCGENES research participants' requests for non-medically actionable secondary findings. Genetics in Medicine. 2019;21(5):1092–9. | Mixed | | | 155 | | USA | Own | | | X | | | | X | | | X | X |
| Ruiz-Canela M, Valle-Mansilla J, Sulmasy D. Researchers' preferences and attitudes on ethical aspects of genomics research: a comparative study between the USA and Spain. Journal of Medical Ethics. 2009;35(4):251–7. | Quant | | | 204 | | INT'L | | Unspecified | | X | X | | X | | | X | | |

(Continued)

**Table 2.** (Continued)

| Full Citation | Data Type | Participants by Method — Focusgroups | Interviews | Survey | Other | Country | Results (to be) Received | Results (to be) Given | Context — Biobanks | Research | Stakeholders — HCPs | Review Boards | Researchers | Research Participants | Publics | Decision Type — Hypothetical/Policy/Practice | Decided | Given/Received |
|---|---|---|---|---|---|---|---|---|---|---|---|---|---|---|---|---|---|---|
| Ruiz-Canela M, Valle-Mansilla JI, Sulmasy DP. What research participants want to know about genetic research results: the impact of "genetic exceptionalism". Journal of Empirical Research on Human Research Ethics. 2011;6(3):39–46. | Quant | | | 279 | | INT'L | Own | | X | | | | | X | | X | X | |
| Sabatello M, Zhang Y, Chen Y and Appelbaum PS. In different voices: the views of people with disabilities about return of results from precision medicine research. Public Health Genomics, 2020;23(1–2):42–53. | Quant | | | 1,294 | | USA | Own | | | X | | | | X | X | X | X | |
| Salvaterra E, Giorda R, Bassi MT, Borgatti R, Knudsen LE, Martinuzzi A, et al. Pediatric biobanking: a pilot qualitative survey of practices, rules, and researcher opinions in ten European countries. Biopreservation and Biobanking. 2012;10(1):29–36. | Qual | | | 18 | | INT'L | | Children | X | | | | X | | | | | X |
| Sanderson SC, Diefenbach MA, Zinberg R, Horowitz CR, Smirnoff M, Zweig M, et al. Willingness to participate in genomics research and desire for personal results among underrepresented minority patients: a structured interview study. Journal of Community Genetics. 2013;4(4):469–82. | Quant | | | 205 | | USA | Own | | | X | | | | X | | X | | |
| Sanderson SC, Lewis C, Patch C, Hill M, Bitner-Glindzicz M, Chitty LS. Opening the "black box" of informed consent appointments for genome sequencing: a multisite observational study. Genetics in Medicine. 2019;21(5):1083–91. | Qual | | | | 45 | GBR | Own & Child's | Adults & Children | | X | X | | | | | X | | |
| Sanderson SC, Linderman MD, Suckiel SA, Diaz GA, Zinberg RE, Ferryman K, et al. Motivations, concerns and preferences of personal genome sequencing research participants: baseline findings from the HealthSeq project. European Journal of Human Genetics. 2016;24(1):14–20. | Mixed | | 35 | 35 | | USA | Own | | | X | | | | X | | | | X |
| Sanderson SC, Linderman MD, Suckiel SA, Zinberg R, Wasserstein M, Kasarskis A, et al. Psychological and behavioural impact of returning personal results from whole-genome sequencing: the HealthSeq project. European Journal of Human Genetics. 2017;25(3):280–92. | Mixed | | 35 | 35 | | USA | Own | | | X | | | | X | | X | | |
| Sapp JC, Dong D, Stark C, Ivey LE, Hooker G, Biesecker LG, et al. Parental attitudes, values, and beliefs toward the return of results from exome sequencing in children. Clinical Genetics. 2014;85(2):120–6. | Qual | | 25 | | | USA | Child's | | | X | | | | X | | X | | |

*(Continued)*

**Table 2.** (Continued)

| Full Citation | Data Type | Focusgroups | Interviews | Survey | Other | Country | Results (to be) Received | Results (to be) Given | Biobanks | Research | HCPs | Review Boards | Researchers | Research Participants | Publics | Hypothetical/Policy/Practice | Decided | Given/Received |
|---|---|---|---|---|---|---|---|---|---|---|---|---|---|---|---|---|---|---|
| Sapp JC, Johnston JJ, Driscoll K, Heidlebaugh AR, Miren Sagardia A, Dogbe DN, et al. Evaluation of recipients of positive and negative secondary findings evaluations in a hybrid CLIA-research sequencing pilot. American Journal of Human Genetics. 2018;103(3):358–66. | Mixed | | 13 | 107 | | USA | Own & Child's | | | X | | | | X | | | | X |
| Schmanski A, Roberts E, Coors M, Wicks SJ, Arbet J, Weber R, et al. Research participant understanding and engagement in an institutional, self-consent biobank model. Journal of Genetic Counseling. 2020;30(1):257–267. | Quant | | | 856 | | USA | Own | | X | | | | | X | | X | | |
| Similuk MN, Yan J, Setzer MR, Jamal L, Littel P, Lenardo M, et al. Exome sequencing study in a clinical research setting finds general acceptance of study returning secondary genomic findings with little decisional conflict. Journal of Genetic Counseling. 2020;30(3):766–773. | Quant | | | 76 | | USA | Own | | | X | | | | X | | | X | |
| Siminoff LA, Traino HM, Mosavel M, Barker L, Gudger G, Undale A. Family decision maker perspectives on the return of genetic results in biobanking research. Genetics in Medicine. 2016;18(1):82–8. | Mixed | | 55 | 22 | | USA | Relatives' | | X | | | | | X | | X | | |
| Simon CM, Williams JK, Shinkunas L, Brandt D, Daack-Hirsch S, Driessnack M. Informed consent and genomic incidental findings: IRB chair perspectives. Journal of Empirical Research on Human Research Ethics. 2011;6(4):53–67. | Qual | | 34 | | | USA | | Adults | | X | | X | | | | X | | |
| Smit AK, Newson AJ, Best M, Badcock CA, Butow PN, Kirk J, et al. Distress, uncertainty, and positive experiences associated with receiving information on personal genomic risk of melanoma. European Journal of Human Genetics. 2018;26(8):1094–100. | Quant | | | 103 | | AUS | Own | | | X | | | | X | | | | X |
| Sng WT, Yeo SN, Lin BX, Lee TS. Impacts of apolipoprotein E disclosure on healthy Asian older adults: a cohort study. International Psychogeriatrics. 2019;31(10):1499–507. | Quant | | | 280 | | SGP | Own | | | X | | | | X | | | | X |
| Spies G, Mokaya J, Steadman J, Schuitmaker N, Kidd M, Hemmings SMJ, et al. Attitudes among South African university staff and students towards disclosing secondary genetic findings. Journal of Community Genetics. 2021;12(1):171–184. | Quant | | | 674 | | ZAF | Own & Child's | | | X | | | | X | | X | | |
| Stein CM, Ponsaran R, Trapl ES, Goldenberg AJ. Experiences and perspectives on the return of secondary findings among genetic epidemiologists. Genetics in Medicine. 2019;21(7):1541–7. | Mixed | | | 216 | | INT'L | | Adults | | X | | | X | | | X | | X |

*(Continued)*

**Table 2.** (Continued)

| Full Citation | Data Type | Participants by Method — Focusgroups | Interviews | Survey | Other | Country | Results (to be) Received | Results (to be) Given | Context — Biobanks | Research | Stakeholders — HCPs | Review Boards | Researchers | Research Participants | Publics | Decision Type — Hypothetical/Policy/Practice | Decided | Given/Received |
|---|---|---|---|---|---|---|---|---|---|---|---|---|---|---|---|---|---|---|
| Sundby A, Boolsen MW, Burgdorf KS, Ullum H, Hansen TF, Middleton A, et al. Stakeholders in psychiatry and their attitudes toward receiving pertinent and incident findings in genomic research. American Journal of Medical Genetics—Part A. 2017;173(10):2649–58. | Quant | | | 2,637 | | DNK | Own & Relatives' | Adults & Relatives | | X | X | | X | X | X | X | | |
| Sundby A, Boolsen MW, Burgdorf KS, Ullum H, Hansen TF, Middleton A, et al. The preferences of potential stakeholders in psychiatric genomic research regarding consent procedures and information delivery. European Psychiatry. 2019;55:29–35. | Quant | | | 2,637 | | DNK | Own & Relatives' | Adults | | X | X | | X | X | X | X | | |
| Sundby A, Boolsen MW, Burgdorf KS, Ullum H, Hansen TF, Mors O. Attitudes of stakeholders in psychiatry towards the inclusion of children in genomic research. Human genomics. 2018;12(12):1–11. | Mixed | 22 | 7 | 2,637 | | DNK | Child's | Children | | X | | | | X | | X | | |
| Taber JM, Klein WM, Ferrer RA, Lewis KL, Biesecker LG, Biesecker BB. Dispositional optimism and perceived risk interact to predict intentions to learn genome sequencing results. Health Psychology. 2015;34(7):718–28. | Quant | | | 496 | | USA | Own | | | X | | | | X | | | X | |
| Taber JM, Klein WM, Lewis KL, Johnston JJ, Biesecker LG, Biesecker BB. Reactions to clinical reinterpretation of a gene variant by participants in a sequencing study. Genetics in Medicine. 2018;20(3):337–45. | Quant | | | 58 | | USA | Own | | | X | | | | X | | | | X |
| Tabor HK, Brazg T, Crouch J, Namey EE, Fullerton SM, Beskow LM, et al. Parent perspectives on pediatric genetic research and implications for genotype-driven research recruitment. Journal of Empirical Research on Human Research Ethics. 2011;6(4):41–52. | Qual | | 23 | | | USA | Child's | | | X | X | | X | X | X | X | | |
| Tamayo LI, Lin H, Ahmed A, Shahriar H, Hasan R, Sarwar G, et al. Research participants' attitudes towards receiving information on genetic susceptibility to arsenic toxicity in rural Bangladesh. Public Health Genomics. 2020;23(1–2):69–76. | Quant | | | 200 | | BGD | Own | | | X | | | | X | | X | | |
| Toccaceli V, Brescianini S, Fagnani C, Gigantesco A, D'Abramo F, Stazi MA. What potential donors in research biobanking want to know: a large population study of the Italian Twin Registry. Biopreservation and Biobanking. 2016;14(6):456–63. | Quant | | | 1,486 | | ITA | Own | | X | | | | | X | | X | | |
| Trinidad SB, Ludman EJ, Hopkins S, James RD, Hoeft TJ, Kinegak A, et al. Community dissemination and genetic research: moving beyond results reporting. American Journal of Medical Genetics—Part A. 2015;167(7):1542–50. | Qual | 121 | | | | USA | Own | | | X | | | | X | X | X | | |

(*Continued*)

**Table 2.** (Continued)

| Full Citation | Data Type | Focusgroups | Interviews | Survey | Other | Country | Results (to be) Received | Results (to be) Given | Biobanks | Research | HCPs | Reivew Boards | Researchers | Research Participants | Publics | Hypothetical/Policy/Practice | Decided | Given/Received |
|---|---|---|---|---|---|---|---|---|---|---|---|---|---|---|---|---|---|---|
| Turbitt E, Chrysostomou PP, Peay HL, Heidlebaugh AR, Nelson LM, Biesecker BB. A randomized controlled study of a consent intervention for participating in an NIH genome sequencing study. European Journal of Human Genetics. 2018;26(5):622–30. | Quant | | | 188 | | USA | Own | | | X | | | | X | | | X | |
| Turbitt E, Roberts MC, Hollister BM, Lewis KL, Biesecker LG, Klein WMP. Ethnic identity and engagement with genome sequencing research. Genetics in Medicine. 2019;21(8):1735–43. | Quant | | | 408 | | USA | Own | | | X | | | | X | | X | | |
| Vaz M and Vaz M. The views of ethics committee members and medical researchers on the return of individual research results and incidental findings, ownership issues and benefit sharing in biobanking research in a South Indian city. Developing World Bioethics, 2018;18(4):321–330. | Qual | | | 43 | | IND | | Adults | X | | | X | X | | | X | | |
| Verbrugge J, Cook L, Miller M, Rumbaugh M, Schulze J, Heathers L, et al. Outcomes of genetic test disclosure and genetic counseling in a large Parkinson's disease research study. Journal of Genetic Counseling, 2020;30(3):755–765. | Quant | | | 875 | | USA | Own | | | X | | | | X | | | | X |
| Vermeulen E, Rebers S, Aaronson NK, Brandenburg AP, van Leeuwen FE and Schmidt MK. Patients' attitudes towards the return of incidental findings after research with residual tissue: a mixed methods study. Genetic Testing and Molecular Biomarkers, 2018;22(3):178–186. | Mixed | | 146 | 673 | | NLD | Own | | X | X | | | | X | | X | | |
| Viberg Johansson J, Langenskiöld S, Segerdahl P, Hansson MG, Hösterey UU, Gummesson A, et al. Research participants' preferences for receiving genetic risk information: a discrete choice experiment. Genetics in Medicine. 2019;21(10):2381–9. | Quant | | | 351 | | SWE | Own | | | X | | | | X | | X | | |
| Waltz M, Meagher KM, Henderson GE, Goddard KAB, Muessig K, Berg JS, et al. Assessing the implications of positive genomic screening results. Personalized Medicine, 2020;17(2):101–109. | Qual | | 11 | | | USA | Own | | | X | | | | X | | | | X |
| Wendler D, Pentz R. How does the collection of genetic test results affect research participants? American Journal of Medical Genetics —Part A. 2007;143(15):1733–8. | Quant | | | 315 | | USA | Own | | | X | | | | X | | X | | |
| Wilkins CH, Mapes B, Jerome RN, Villalta-Gil V, Pulley JM, Harris PA. Understanding what information is valued by research participants and why. Health Affairs. 2019;38(3):399–407. | Quant | | | 2,549 | | USA | Own | | | X | | | | | X | X | | |

*(Continued)*

**Table 2.** (Continued)

| Full Citation | Data Type | Participants by Method | | | | Country | Results (to be) Received | Results (to be) Given | Context | | Stakeholders | | | | | Decision Type | | |
|---|---|---|---|---|---|---|---|---|---|---|---|---|---|---|---|---|---|---|
| | | Focusgroups | Interviews | Survey | Other | | | | Biobanks | Research | HCPs | Review Boards | Researchers | Research Participants | Publics | Hypothetical/Policy/Practice | Decided | Given/Received |
| Williams JK, Daack-Hirsch S, Driessnack M, Downing N, Shinkunas L, Brandt D, et al. Researcher and Institutional Review Board chair perspectives on incidental findings in genomic research. Genetic Testing and Molecular Biomarkers. 2012;16(6):508–13. | Qual | | 53 | | | USA | | Adults & Children | | X | X | X | X | | | X | | X |
| Wright MF, Lewis KL, Fisher TC, Hooker GW, Emanuel TE, Biesecker LG, et al. Preferences for results delivery from exome sequencing/genome sequencing. Genetics in Medicine. 2014;16(6):442–7. | Qual | 39 | | | | USA | Own | | | X | | | X | X | | | X | X |
| Wynn J, Lewis K, Amendola LM, Bernhardt BA, Biswas S, Joshi M, et al. Clinical providers' experiences with returning results from genomic sequencing: an interview study. BMC Medical Genomics. 2018;11:45. | Mixed | | 21 | 21 | | USA | | Adults & Children | | X | X | | | | | | | X |
| Wynn J, Martinez J, Bulafka J, Duong J, Zhang Y, Chiuzan C, et al. Impact of receiving secondary results from genomic research: a 12-month longitudinal study. Journal of Genetic Counseling. 2018;27(3):709–22. | Quant | | | 192 | | USA | Own | | | X | | | | X | | X | | X |
| Wynn J, Martinez J, Duong J, Chiuzan C, Phelan JC, Fyer A, et al. Research participants' preferences for hypothetical secondary results from genomic research. Journal of Genetic Counseling. 2017;26(4):841–51. | Quant | | | 219 | | USA | Own | | | X | | | | X | | X | | |
| Wynn J, Martinez J, Duong J, Zhang Y, Phelan J, Fyer A, et al. Association of researcher characteristics with views on return of incidental findings from genomic research. Journal of Genetic Counseling. 2015;24(5):833–41. | Quant | | | 241 | | USA | | Adults & Children | | X | X | | X | | | X | | X |
| Yamamoto K, Hachiya T, Fukushima A, Nakaya N, Okayama A, Tanno K, et al. Population-based biobank participants' preferences for receiving genetic test results. Journal of Human Genetics. 2017;62(12):1037–48. | Quant | | | 3,345 | | JPN | Own | | X | | | | | X | X | X | | |
| Yamamoto K, Shimizu A, Aizawa F, Kawame H, Tokutomi T, Fukushima A. A comparison of genome cohort participants' genetic knowledge and preferences to receive genetic results before and after a genetics workshop. Journal of Human Genetics. 2018;63(11):1139–47. | Quant | | | 112 | | JPN | Own | | X | | | | | X | | X | | |
| Yamamoto M, Sakurai K, Mori C and Hata A. Participant mothers' attitudes toward genetic analysis in a birth cohort study. Journal of Human Genetics. 2021;66(6):671–679. | Quant | | | 1,762 | | JPN | Own & Child's & Relatives' | | X | | | | | X | | X | | |

*(Continued)*

**Table 2.** (Continued)

| Full Citation | Data Type | Focusgroups | Interviews | Survey | Other | Country | Results (to be) Received | Results (to be) Given | Biobanks | Research | HCPs | Review Boards | Researchers | Research Participants | Publics | Hypothetical/Policy/Practice | Decided | Given/Received |
|---|---|---|---|---|---|---|---|---|---|---|---|---|---|---|---|---|---|---|
| Young M-A, Herlihy A, Mitchell G, Thomas DM, Ballinger M, Tucker K, et al. The attitudes of people with sarcoma and their family towards genomics and incidental information arising from genetic research. Clinical Sarcoma Research, 2013;3(1):1–9. | Quant | | | 1,200 | | AUS | Own & Relatives' | | | X | | | | X | | X | | |
| Yu JH, Crouch J, Jamal SM, Bamshad MJ, Tabor HK. Attitudes of non-African American focus group participants toward return of results from exome and whole genome sequencing. American Journal of Medical Genetics—Part A. 2014;164(9):2153–60. | Qual | 35 | | | | USA | Own & Child's | | | X | | | | | X | X | | |
| Yu JH, Crouch J, Jamal SM, Tabor HK, Bamshad MJ. Attitudes of African Americans toward return of results from exome and whole genome sequencing. American Journal of Medical Genetics—Part A. 2013;161(5):1064–72. | Qual | 41 | | | | USA | Own & Child's | | | X | | | | | X | X | | |
| Zhu X, Basappa SN, Ridgeway JL, Albertie ML, Pantoja E, Prescott D, et al. Perspectives regarding family disclosure of genetic research results in three racial and ethnic minority populations. Journal of Community Genetics, 2020;11(4):433–443. | Qual | 68 | | | | USA | Own & Relatives' | | | X | | | | | X | X | | |
| Zhu X, Leof ER, Rabe KG, McCormick JB, Petersen GM, Breitkopf CR. Psychological impact of learning CDKN2A variant status as a genetic research result. Public Health Genomics. 2018;21(3–4):154–63. | Quant | | | 63 | | USA | Own | | X | | | | | X | | | | X |
| Ziniel S, Savage SK, Huntington N, Amatruda J, Green RC, Weitzman ER, et al. Parents' preferences for return of results in pediatric genomic research. Public Health Genomics. 2014;17(2):105–14. | Quant | | | 1,060 | | USA | Own & Child's | | | X | | | | | X | X | | |
| Zoltick ES, Linderman MD, McGinniss MA, Ramos E, Ball MP, Church GM, et al. Predispositional genome sequencing in healthy adults: design, participant characteristics, and early outcomes of the PeopleSeq Consortium. Genome Medicine. 2019;11(1):10. | Quant | | | 543 | | USA | Own | | | X | | | | X | | | | X |

**Table 3. Demographics of included papers.**

| Category | Number of studies* | Percentage of studies* | Number of Participants* | Percentage of Participants* |
|---|---|---|---|---|
| **Total** | **221** | **100** | **118,874** | **100** |
| **Study type** | | | | |
| Quantitative | 118 | 53.4% | 103,883 | 87.4% |
| Qualitative | 69 | 31.2% | 2,591 | 2.2% |
| Mixed methods | 34 | 15.4% | 12,400 | 10.4% |
| **Country** | | | | |
| Australia | 5 | 2.3% | 2,157 | 1.8% |
| Bangladesh | 1 | 0.5% | 200 | 0.2% |
| Botswana | 1 | 0.5% | 105 | 0.1% |
| Canada | 12 | 5.4% | 1,063 | 0.9% |
| Denmark | 5 | 2.3% | 8,099 | 6.8% |
| Estonia | 2 | 0.9% | 27 | 0.0% |
| Finland | 2 | 0.9% | 272 | 0.2% |
| France | 1 | 0.5% | 261 | 0.2% |
| India | 1 | 0.5% | 43 | 0.0% |
| Ireland | 1 | 0.5% | 351 | 0.3% |
| Italy | 2 | 0.9% | 1,628 | 1.4% |
| Japan | 4 | 1.8% | 7,076 | 6.0% |
| Jordan | 1 | 0.5% | 3,196 | 2.7% |
| Kenya | 1 | 0.5% | 63 | 0.1% |
| Netherlands | 5 | 2.3% | 2,096 | 1.8% |
| Singapore | 1 | 0.5% | 280 | 0.2% |
| South Africa | 2 | 0.9% | 678 | 0.6% |
| Sweden | 2 | 0.9% | 354 | 0.3% |
| Switzerland | 1 | 0.5% | 25 | 0.0% |
| UK | 11 | 5.0% | 1,619 | 1.4% |
| USA | 144 | 65.2% | 79,698 | 67.0% |
| Zambia | 1 | 0.5% | 21 | 0.0% |
| Multi-country | 15 | 6.8% | 9,562 | 8.0% |
| **Context** | | | | |
| Biobanks | 61 | 27.6% | 47,487 | 39.9% |
| Research | 168 | 76.0% | 72,302 | 60.8% |
| **Stakeholders** | | | | |
| Healthcare Professionals | 23 | 10.4% | 14,944 | 12.6% |
| Review Boards | 13 | 5.9% | 2,822 | 2.4% |
| Researchers | 40 | 18.1% | 15,042 | 12.7% |
| Research Participants | 153 | 69.2% | 85,270 | 71.7% |
| Public | 36 | 16.3% | 40,967 | 34.5% |
| **Situation type** | | | | |
| Hypothetical/Policy | 129 | 58.4% | 96,416 | 81.1% |
| Decision made | 38 | 17.2% | 23,736 | 20.0% |
| Results returned | 71 | 32.1% | 17,164 | 14.4% |

* The total number of papers or participants in the various groupings below is sometimes greater than 221 and 118,874 respectively. This is because: (1) papers were frequently assigned to multiple categories (e.g. a biobank and research context; decision made and results returned); and (2) study participants were similarly assigned more than once (e.g. interview and survey). Some percentage totals are thus also greater than 100%.

**Table 4. Summaries of articles that address stakeholder interest in receiving study-specific results.**

| Stakeholder | Setting | Interest in IRR | Publications |
|---|---|---|---|
| Participants | Clinical Research | 48% to 97% | [1–13, 30–32] |
|  | Biobank | 57% to 96% | [33–41]. |
| Patients/parents | Clinical Research | 61% to 98% | [13, 34, 42–50] |
|  | Biobank | 53.5% to 88% | [50–52] |
| Public | Clinical Research | 73% to 95% | [13, 44, 49, 53–59] |
|  | Biobank | 91% to 98% | [60–63] |

to change their doctor's approach to their care [3, 40]. Yet desires for information persist for some participants, both with and without existing medical conditions, even when options for return of information include diseases without known prevention, treatment, or other action-ability, and genes with uncertain significance [3, 5, 66, 67]. A study of 219 cognitively normal adults enrolled in longitudinal aging studies indicated that 51.9% wanted Apolipoprotein E (*APOE*) genotype, despite the fact that Alzheimer's disease is non-actionable [7].

With hypothetical scenarios, there was a strong preference for results in parents of children participating in research [68]. Interviews with 25 parents of children in an exome sequencing study at the National Institute of Health (NIH) showed that all participants wanted to receive their children's results [11]. Similarly, 97% of 362 parents of children with rare inherited child-hood diseases or pediatric cancer who were participating in one of three large-scale genome research consortia stated a positive right to receive SSR [6].

High levels of interest in learning about results were also expressed by participants making actual decisions about whether they wanted to receive results [1, 2, 12]. Several studies observed that return of results was a key reason for research participation [2, 4, 69, 70] and a survey exploring participation across several Clinical Sequencing Exploratory Research (CSER) consortium sites showed that only 5% who declined did so because they did not want research results returned [71]. In a study of 263 veterans and 1,159 non-veteran adult Mexi-can-Americans enrolled in genetic family studies aimed at identifying increased susceptibility to diabetes and diabetic nephropathy, 95.7% and 93.1% respectively expressed interest in receiving their SSR [1]. Similarly, in a study of adolescents aged 14–18 years who were either undergoing treatment for substance and conduct problems (SCPs; n = 320) or were non-SCP controls (n = 109), most participants (77.8% of SCPs and 72.5% of non-SCPs) wanted to know results if there were health or behavioral implications [4].

In general, very low proportions of participants– 8.2% of 790 undiagnosed adults and children with conditions suspected to have a primarily monogenic cause and 3% of 506 adult members of families at high genetic risk for cancer–refused all results [72, 73]. However, some studies highlighted that some participants only want results when investigators have assessed the risks and benefits of sharing results with them (68/271; 25.1%), or if the results would be useful for their doctor's decision-making (74/271; 27.3%) [10]. Surveys and interviews with 35 participants from New York showed that 94% wanted to receive all categories of results [12]. Yet, when participants were asked to nominate which categories of results they wished to receive, 100% wanted pharmacogenetics information, but only 74% wished to receive results regarding conditions which were unpreventable [12]. Studies have found that adults [74] and parents [75, 76] may be uncertain about which results they want and what to do with them. Although 85% of 154 parents of children recruited to studies investigating genetic causes and novel therapeutics for rare diseases, indicated they had a strong/very strong right to receive results, even in situations when the possibility of an ameliorative therapy was uncertain [77],

others did not wish to receive results that could not be interpreted or were non-actionable/incurable for their children [78].

Two studies that returned results at study completion–one with 107 men and women from families with a known mismatch repair (MMR) gene mutation and the other with 31 adult participants from the ClinSeq cohort who had received one sequencing result with personal health implications–showed that 79% and 80.6% respectively elected to receive results at the start of the study [8, 9]. A third study of 31 healthy individuals, conducted by Stanford University, showed that receiving exome results to learn more about their health risks was the most common reason for participation [79]. A further study of 162 adults who had a living or deceased parent with Alzheimer's disease and were randomized to receive or not receive their own *APOE* genotype showed that some participants randomized to the nondisclosure group were dissatisfied at not receiving their genotyping results [80].

Participants from all these studies present a range of reasons for wanting to receive SSR, which are summarised in Table 5. These reasons differ slightly depending on the condition under investigation and whether the research testing was being conducted in adults or their children. For example, parents of children in an exome sequencing study at the NIH wanted results to identify an explanation for their child's condition, provide information about their child's health, coordinate better management for their child's condition, and prepare for their child's future healthcare needs [11]. Some wanted answers from genomic sequencing to lead to treatments [75]. Parents also described feelings of responsibility toward their children and desires for control associated with receiving the results [11]. In contrast, adult participants want SSR in order to have greater certainty about personal risk [8, 30], to determine whether they required screening [8], because it was recommended by their healthcare professional or desired by a relative [8], or for insurance or planning purposes [8, 30, 88]. Participants also listed a desire to know health information [5, 12, 81], provide diagnostic certainty [82], prevent disease/improve health [5, 12, 81, 86, 88], adopt better health habits [88] and alter medical management as reasons for wanting their results [5, 82].

In a hospital-based study of genetic susceptibility to melanoma, 68% of 19 participants wanted to know their personal risk so they could discuss it with their doctor, so they could use

**Table 5. Summary of reasons for wanting and not wanting to receive SSR and UF/SF.**

| | Participants (SSR) | Participants (UF/SF) | Patients (SSR) | Patients (UF/SF) | Public (SSR) | Public (UF/SF) |
|---|---|---|---|---|---|---|
| **Reasons for wanting to receive results** | | | | | | |
| For health information, clinical utility, disease prevention, or to improve health or medical management | [5, 12, 45, 81–86] | [69, 87, 88] | [42, 48, 52, 89] | [90] | [56, 91–93] | [94] |
| Identify cause of child's current/future condition, informed about child's health and care management, avoid harm to child | [11, 36, 45, 75] | | | [95] | | [61, 96] |
| Curiosity, information seeking, empowerment, ownership over results | [12, 30, 83, 85] | [69, 87, 88] | [48] | [95] | [91, 92, 97] | [96] |
| Responsibility or moral obligation to children or family | [11, 36] | [75, 87] | | | | |
| To inform their children and other family members | [8, 12, 30, 83, 85] | | [89] | | [98] | |

*(Continued)*

**Table 5.** (Continued)

| | Participants (SSR) | Participants (UF/SF) | Patients (SSR) | Patients (UF/SF) | Public (SSR) | Public (UF/SF) |
|---|---|---|---|---|---|---|
| Planning and insurance purposes | [8, 30, 36] | [88] | [42] | [95] | | |
| Motivation for change or healthier lifestyle | [36] | [69, 88, 99] | [42] | | | |
| Diagnostic certainty or better understanding of primary condition | [82] | [100] | | | | |
| Understanding of personal risk, to know if need screening, so can discuss with doctor | [8, 30, 83] | | | | | |
| Family or personal history of a disease | [5] | [87, 99] | | | | |
| Family planning | [69, 85, 101] | | | | [96] | |
| Personal utility | [36] | | | | | [102] |
| Reassurance, peace of mind, contribute to overall wellbeing | [83] | [88] | [89] | | | |
| Recommended by healthcare professional | [8] | | | | | |
| Desire for control | [11] | [99] | | | | |
| Right to know own or child's results | | | | [95] | | [94] |
| To participate in a clinical trial | [103] | | | | | |
| Promotion of autonomy | | | | | [92] | |
| To improve population health or public health knowledge | | | | | [92] | |
| Advocate for clinical services | | | | | | [96] |
| Reduce stigma | | | | | | [96] |
| Shows participation is valued | [86] | | | | | |
| **Reasons for not wanting to receive results** | | | | | | |
| Potential for adverse psychological impact | [2, 12, 85] | [69, 75, 88, 104, 105] | | | [56, 59, 91, 93] | [62, 94, 96] |
| Implications for children (e.g. insurance, legal, privacy) | [12, 101] | [69, 75] | | | | |
| Balancing benefits and risks of knowing | | [69, 75] | | | | |
| Participants may not wish to know | [106] | | | | | [94] |
| May become overly vigilant/lead to unnecessary appointments | | [88] | | | | |
| Inability to make health changes | [85] | | | | | |
| Concerns about availability of health or life insurance | [106] | | | | [56, 59] | |
| Concerns about discrimination, stigma | [85] | | | | | [94] |
| Concerns about privacy | [84, 85, 106] | | | | | |
| Concerns about potential for inaccurate, uncertain, changeable results | | | | | [56, 93] | [94] |
| Do not consider themselves to be at risk | | | | | [97] | |
| Lack of resources and clinical expertise | | | | | | [94] |

it for preventative purposes (58%), or wanted reassurance that they were not at increased risk (37%). Yet almost half (47%) also wanted to receive results out of curiosity [83]. This interest in receiving results out of curiosity was supported in the study of healthy participants by Sanderson et al [12]. In addition, 74% wanted results to be able to inform their children [83], a finding which has also been recognised by others, along with a desire to tell other family members [12] and for family planning purposes [69, 101]. The ClinSeq study indicated that a proportion were motivated to receive results by their family history of a particular disease [5]. In addition, adults enrolled in the Dominantly Inherited Alzheimer's Network want SSR in order to participate in a clinical trial [103]. Interestingly, in a study of 246 cancer patients and 315 participants who had a family history of Alzheimer's disease (but no disease themselves), participants said they would be more likely to want to know their results if they knew the researchers already had access to that information [107]. Yet, participants from three other studies also raised concerns about receiving IRR due to the potential for adverse psychological impact of receiving results that have implications for future health [2, 12], and implications of results for their children, including the impact on insurance, legal issues and privacy [12, 101].

## 1.1 Participants' preferences for receiving study-specific results (continued)

**Biobank setting.** Participants in the context of biobanks also express high interest in SSR, with between 57% and 98% wanting results to be returned [33–41, 84, 85, 106, 108–111]. Participants generally expressed higher interest in results that conveyed some sort of actionability [33, 34, 39–41, 112]. For example, in a study of 555 biobank participants recruited in the context of clinical care appointments, 90% wanted results for conditions that are treatable, compared to 64% who wanted results for non-treatable conditions [33]. These participants also showed a preference for receiving results that conveyed high disease risk over low disease risk (79% vs 66%), and risk of serious disease over less serious disease, (83% vs 68%). Yet, 57% expressed interest in receiving uncertain results. This is in contrast to a study which conducted interviews and focus groups with 26 female breast cancer patients who had previously given consent to donate blood or tissue samples to a tissue bank for breast cancer research, in which most participants did not want to receive results of uncertain clinical significance [113]. Another study, which used a discrete choice experiment to survey 351 participants from a Swedish research program, identified a preference to receive life threatening disease risk over other diseases (such as physical disability, mental disease, and physical disease) [40]. They found greater willingness to learn disease risks when the estimate of penetrance was higher, and also when the recipient is able to implement lifestyle changes rather than medical interventions; this proportion increased as did the effectiveness of the interventions [40].

A study of 55 family decision makers who had authorized the donation of deceased loved ones' tissue, and 22 requesters recruited through an organ procurement organization, showed that 94.3% favored the return of results suggestive of treatable diseases and 84.9% for diseases that could affect their children, compared to 71.7% for non-treatable diseases [39]. Likewise, a study of biobank participants from two regions in Japan also highlighted that the majority of respondents (88.2% from region 1 and 82.3% from region 2) preferred to receive their own genetic information [41]. Interest was highest for diseases that could be modified by lifestyle, as well as adult-onset and actionable conditions, with much less interest expressed in receiving pharmacogenetics and adult-onset non-actionable results [41]. Yet, a second Japanese study conducted an educational workshop with 112 participants and showed that scores for interest in receiving five categories of results–lifestyle diseases, pharmacogenetics, adult-onset non-clinically actionable diseases, non-clinically actionable multifactorial diseases, and all genetic

information–significantly decreased after the workshop [114]. However, even in the face of this decrease, over 95% still wanted to know results for the diseases that could be influenced by lifestyle, pharmacogenetics, and adult-onset clinically actionable diseases [114]. In addition, 17 parents of healthy children participating in a biobank wanted to know both childhood- and adult-onset medically actionable conditions for their child and felt it was more important to protect their child's health than preserve their future autonomy [115]. However, one study of 53 young adult participants from the Avon Longitudinal Study of Parents and Children (ALSPAC) showed only mild interest in receiving SSR and only if they were of clinical relevance [116].

Despite the overall high interest in receiving SSR, studies suggest that expectations for their return differ. Interviews with 109 NUgene biobank participants showed that while 1/3 hoped to be recontacted with results, 1/3 expected results to be returned only if something severe was found, and 1/3 had no expectation that they would be recontacted [117]. This is important as participants were told during the consent process that it was extremely unlikely that they would be contacted with research results that could have a significant impact on their health. A study of 3630 adults, which included 464 biobank participants with a diagnosis of pancreatic cancer, indicated that 62.1% expect to receive SSR; high proportions of participants held expectations that they would be told about 'bad' stuff (e.g., health risks for conditions), rather than 'good' stuff (e.g., things that do not have associated health risks) [34]. Yet, interviews with 17 cancer patients, 6 first degree relatives and 7 cancer-free controls showed that while many felt researchers should return research reports to patients, over half said there was no moral obligation to do so [37].

Four studies explicitly explored participants reasons for wanting to receive SSR in the biobank setting [36, 84]. They identified benefits to participants' and their families' health [36, 85], motivation to adopt healthier lifestyles [36], earlier diagnosis or prevention [36, 84], and improvements in chances for better treatment as key reasons for wanting results [36, 84]. Participants also mentioned personal utility, finding meaning in knowing their own or their family members' genetic information, the possibility for future planning [85], including taking out long term insurance [36] and family planning, a sense of obligation or responsibility to their family, having ownership over their results [85], and wanting to be a good parent to their children [36].

## 1.2 Patients' (and parents of patients') preferences for receiving study-specific results

As with research participants, patients (and also parents of patients in the case of children) generally express high interest (between 53.5% and 98%) in return of SSR [13, 34, 42–50, 118, 119]. This applied in both the clinical research and biobanks settings.

**Clinical research setting.** In a study of 904 participants from the US-based Northwest Cancer Genetics Registry, which included 340 patients with cancer, participants strongly endorsed that researchers have an ethical obligation to return results that would affect their health or health care [44]. A study of 205 patients attending an outpatient clinic in the USA showed a significant increase (p<0.001) in interest to participate in a hypothetical genomic research study when they were told that results would be offered following study completion [48]. Sixty-eight percent of these respondents said they would want results returned for heart disease, 67% for diabetes type 2, 70% for cancer and 61% for obesity [48]. Another study of 25 adult parents of children who were inpatients at a pediatric hospital, showed that 64% of parents wanted results for both preventable and non-preventable conditions returned, whereas 35% just wanted to receive results for preventable conditions for their children. Similarly, 76%

of parents wanted both severe and non-severe conditions, 16% wanted results suggesting severe conditions only, and 8% only wanted results for non-severe conditions returned [42]. Yet, a study of patients with mental disorders showed that patients and family members were willing not to receive findings if doing so would compromise the research [13].

Patients discuss wanting to know their results out of personal curiosity [48] or peace of mind [89], because it may provide personal health benefit [48], clinical utility [89] or action-ability [42, 52], or to inform their children of their risks [89]. A qualitative study of 25 parents highlighted that participants considered 'actionability' as a broad concept, encompassing aspects such as medical interventions, lifestyle modifications, education, mental preparation, and planning, including insurance, housing, and finances [42].

**Biobank setting.**   In contrast to the findings of Sundby et al., in a study of 1903 pancreatic cancer patients, their spouses, and other blood relatives, 76.3% of patients said that regardless of the cost, researchers should offer results to research participants [49]. Surveys with 2,960 parents of children from Boston Children's Hospital felt that results would reduce stress over a search for the child's diagnosis and help them look out for symptoms for early screening [52]. One study has suggested that parents are interested in receiving more information about their children than about themselves; 84.6% of 1060 parents/guardians of children who had received care at Boston Children's Hospital indicating they probably or definitely wanted to receive SSR about themselves and 88% wanting them for their children [50]. Interest reduced consid-erably when asked to specify the categories of results they wanted.

## 1.3 Publics' preferences for receiving study-specific results

As identified in studies on both participants and patients, several studies identified return of SSR as a reason for participation by members of the public [45, 56, 58, 60, 62, 91, 120]. This applied to both clinical research and biobank contexts. For example, a hypothetical study of 4659 adult Americans showed that offering SSR was associated with the largest increase in will-ingness to participate in clinical research–greater than factors such as the study being low-bur-den and offering higher compensation [62]. In fact, ¾ of respondents said they would be less willing to participate if SSR were not returned [62]. However, focus groups conducted with 89 members of the general public showed that not all respondents saw returning SSRs as a condi-tion of research participation, with a small number believing the purpose of the research was to study health within the population rather than benefiting individuals [91].

**Clinical research setting.**   A large number of studies have assessed public views on return of SSR in the context of clinical research. Overall, interest in SSR is high [44, 91, 121–124], ranging from 73–100% [13, 49, 53, 54, 57–59]. Although some studies showed high support for receiving all types of results [121], the most support has been shown for results relating to an increased risk of an actionable, treatable, or preventable condition [59, 91, 125, 126] but also for conditions that are do not currently have treatment options [53, 91]. While some stud-ies have suggested that respondents favour receiving results for serious conditions [91, 124], responses to results concerning life-threatening or fatal disorders have been mixed [124]. However, respondents in a study of 100 adults, who were recruited based on the fact they had never worked in genetics, disagreed with the idea that that definition of benefit for returning SSR should be limited to clinical benefit [122].

Studies also indicate members of the public wish to receive information about non-medical traits and information that could change over time [91]. In fact, 59% of respondents in the Dutch study stated that researchers have a duty to inform participants about mutations, even when the consequences for their health are unclear [125]. Interestingly, the magnitude of risk did not appear to be relevant for most respondents in one study, with many suggesting they

wanted to receive information about both very highly elevated and also only slightly elevated risks of conditions [91].

Studies show that members of the public express a range of reasons for wanting to receive their SSR. Some of these are based in desires to improve health, such as that having results can directly help treat or avoid disease or can motivate them to plan or take action relating to their health, now or in the future [56, 91–93]. Some held expectations that results would impact on the management of their existing conditions, or those of their family members [59]. Yet, for others, results were wanted more out of curiosity, or because they felt a sense of ownership or empowerment from knowing the information in their genes [91, 97]. One study utilized a deliberative strategy with 19 Alaska Native and American Indian community members [92]. These stakeholders desired results for individual purposes, such as promotion of autonomy, privacy, empowerment to make informed decisions, early detection of disease, and improved preventative care, but also to empower and improve health at the population level and increase public health knowledge [92].

However, interest in SSR is not universal. One study of 1418 members of the general public in Missouri indicated that only 12.5% were extremely interested in receiving results regardless of whether there was an available treatment [7]. In another US-based study of 1515 respondents, 12% stated that they would not want or need any SSR to be returned and 56% agreed it would be fair to only receive results that were treatable or preventable [53]. Respondents also showed less interest in receiving results relating to common diseases or that showed major changes in disease risk [53]. Assessment of attitudes of 41 African American parents in Washington, showed that only 15% wanted SSR that could provide answers about an illness or health condition they were currently affected by [59]. In the same study, 26% of non-African American parents did not want to receive any results for themselves or their children [93].

A large proportion of public sentiment behind not wanting to receive SSR appears to be based in concern for the implications of the information, both for personal mental health but also more broadly. Studies have shown that some members of the public fear what they may learn from the results, and that this could lead to distress, depression, or an inability to cope with the information [56, 59, 91, 93]. This was particularly the case with untreatable conditions due to the inevitability that the condition would develop and lack of control over its course [91]. Others expressed concerns about availability of health, life and long-term care insurance and that an SSR might not be actionable if a person were to lose their health insurance [56, 59]. Some respondents were concerned about the potential for inaccurate or uncertain results [56, 93], whereas others either 'felt healthy' or did not have a family history and therefore did not consider themselves to be at risk [97].

## 1.3 Publics' preferences for receiving study-specific results (continued)

**Biobank setting.**   Members of the public generally express high hypothetical preferences (91–98%) for receiving SSR from biobanks [60–63]. This is the case both for themselves and also for their children, as evidenced by a study of 141 individuals across 15 focus groups where, of the 7 focus groups that discussed return of children's results, members of 6 of the groups said that some or all of results of children <18 years should be returned to their parents [61]. Desires to receive SSR related to all types of findings; of 4659 adult Americans surveyed, 91% wanted to receive results about health risks, regardless of their actionability, 95% wanted to know if they were at increased genetic risk for something treatable (e.g., asthma), 96% wanted to know if they were at increased genetic risk for something untreatable (e.g., Alzheimer's disease), and 96% wanted to know if they were at increased genetic risk for "a bad reaction to certain types of medicine" [62]. A study of 45 adult African American caregivers of

children recruited through a health centre showed that 89% were interested in receiving IRR for asthma, 93% for Alzheimer's disease, 80% for a gene change shown to be more common in a racial group, and 82% for SSR with uncertain significance [56]. Another article citing the same research cohort also indicated that SSR for non-actionable results were met with more opposition than those that were actionable and that, although respondents generally desired information about Alzheimer's and dementia, they did not wish receive information about other mental disorders [55].

Yet, some respondents were "ambivalent" about return of SSR from biobanks, suggesting that they donate altruistically and therefore would not feel entitled to receive them [127]. Some respondents suggested they would not want SSR because it would be "too much information" (8%), because IRR predicting future illness would worry them (17%) or were just "not that interested" (7%) [62]. Respondents have also been quite divided in their interest to receive results concerning life-threatening or fatal disorders [55].

## 1.4 Mixed professionals' views on returning study-specific results

**Clinical research setting.**    Three studies assessed views of a range of professionals in the clinical research context [18, 128, 129]. For two of these, a large proportion of their cohort was comprised of researchers. As such, they have been discussed within the researcher stakeholder section below. The third interviewed 14 senior professionals from the Netherlands and the UK with expertise a range of areas, such as clinical care, genetics research, molecular genetics, and health care management [129]. The professionals held varying views about whether there was a duty to recontact participants in response to SSR. While some felt there was no duty because research and clinical care have different aims, others felt that the preferences of the participants should guide whether results are returned [129]. Issues such as workload burdens and difficulties identifying to whom results should be returned were also raised.

**Biobank setting.**    At an interactive workshop, a group of 9 human research ethics committees (HREC) members, 10 researchers, 3 health consumers, and 7 "others" including genetic counselors and members of genetic support groups, were asked about their views on returning biobank results to participants [130]. Professionals were generally not supportive of returning results because of the additional time and resources it would require and the difficulties obtaining consent for result return [130]. In contrast, a study of 17 Dutch key figures in the field with a mixture of disciplinary backgrounds suggested more positive views towards returning results to biobank participants [131]. Participants suggested that individual findings for which treatment options are available should be reported regardless of parental wishes but that late onset non-treatable conditions in children should not be returned [131]. Views varied considerably depending on the background of the participant. Biobank experts felt it was important to distinguish between validated, health-related, and actionable findings compared to those without health-related significance. Medical practitioners in the study felt an obligation to return these findings, but not others [131]. Patient representatives were more liberal with which information should be disclosed to parents than other participants [131].

## 1.5 Health professionals' views on returning study-specific results

Three studies deliberately sought to explore health professionals' views on returning SSR [13, 127, 132].

**Clinical research setting.**    Two studies explored views relating to the clinical research setting. One study which included 74 psychiatrists and 28 clinical geneticists showed that clinical geneticists and psychiatrists were less positive about receiving any kinds of findings for themselves compared to people with mental disorders or relatives [13]. Psychiatrists and clinical

geneticists were also less positive about receiving genomic findings compared with blood donors. The second study, which interviewed 25 pediatric and adult physicians and non-genetic specialists (in oncology and cardiology), suggested that these practitioners only wanted to receive results for their patients where clear actions could be taken, such as the identification of earlier interventions or new care pathways [132]. However, the clinicians also had a number of concerns about returning results to patients. Some of these related to potential for harm to participants, such as anxiety or distress, false negatives, turning healthy individuals into 'patients-in-waiting', subjecting people to unnecessary investigations, and genetic discrimination. Yet other concerns related to their own lack of preparedness to return the results and provide support and the extra time required [132].

**Biobank setting.** One study, which assessed views of healthcare providers working in both public and private practice on return of results from biobanks, suggested that they felt biobank participants had a strong right to know results [127]. However, they also reported being conflicted because the results also have implications for family members, flagging that result recipients require proper support and guidance with how to manage them.

## 1.6 Researchers' views on returning study-specific results

**Clinical research setting.** Seventeen studies have investigated views of researchers on returning IRR [44, 76, 84, 119, 18, 128, 133–143]. A study of 39 psychiatric genetic researchers from 17 countries indicated that the majority of participants were either not returning results at all or had returned results but were not doing so in a systematic way [137]. In a study of 74 genomics researchers (which included medical geneticists, genetics researchers, and clinicians), only 8.1% said results should not be returned [134], and interviews with 23 researchers (both clinical and non-clinical) who investigate Autism Spectrum Disorder generally felt that results that explain the cause of autism should be disclosed [76]. High support for return of results was seen in other studies [44, 137, 142].

Researchers were more supportive of returning results if they explain the cause of the condition under study [76], relate to treatable or preventable conditions [137, 142], and have clinical utility [134], clinical relevance [135], or if the result is available as a clinical test [134]. However, one study also showed support for returning results that provided non-medical actionability (54%) because it could lead to behavior change or because patients had a right to know [136]. Another study showed some support for results that are medically relevant but not actionable (45%) and higher support for returning risk for Huntington disease (71%) and Alzheimer's disease risk (64%) for planning purposes [142]. There was very little support for returning VUS. Researchers also discussed other benefits of returning results, such as the ability to allow for early interventions [136], plan their futures [137], to improve quality of life [136], and the potential to lower their environmental risks for psychiatric disease and reduce stigma associated with mental illness [136]. Researchers also suggested returning results shows respect for patient-participant autonomy and recognizes participants' ownership of their data [142].

However, despite these favourable views, others raised concerns about the uncertainty of the information being returned to participants [137, 140], the potential impact on scientific progress [140], burden on researchers who cannot provide the necessary support [136] or lack of infrastructure and resources [143], and blurring lines between research and clinical care [136, 140]. Others held concerns for participants' privacy and confidentiality [128, 137], discrimination from insurance companies and banks [143], their ability to retain control of their data [137] and ensuring informed consent [18, 137]. In addition, researchers were worried about the potential for adverse psychological reactions from returning results, including worry

[136], confusion [136, 137], anxiety, guilt [137], and stigmatization [137]. Respondents in one study raised concerns that returning IRR would mean that participants use the information to make clinical decisions, which would then place clinical demands on researchers [139]. Another study stressed the need for results to be disclosed by a physician who can explain the significance of the findings and the importance of results being communicated to participants in a personalized manner [84]. In addition, two studies suggested that relatively high proportions of researchers were not giving consideration to returning results to participants [133, 135]. One study of 105 genetics/genomics researchers indicated that, at the time, only 54% had considered the issue of returning research results, 28% had offered to return IRR, and only two of these incorporated this into the study planning [135].

## 1.6 Researchers' views on returning study-specific results (continued)

**Biobank setting.**   Five studies assessed views of researchers on returning SSR to biobank participants [144–148]. The first, a study of 80 researchers involved in biobanks, showed that 74% reported that biobank participants only need to be informed of results if they have treatment or prevention implications and 95% reported that it was fine for participants not to find out about SSR where health implications are unclear [145]. This could be related to the fact that 81% were concerned that returning results might frighten patient-participants and 66% believed that information on genetic variation could influence insurance premiums. In addition, 91% of respondents stated that patients are not more entitled than healthy participants to receive SSR (defined in the study as 'genetic variations that in some form or other may be relevant for a specific individual participant') [145].

Another study, which assessed views of 10 directors of Cooperative Group cancer clinical trial biobanks, showed that all participants completely disagree that the Cooperative Group Bank should be responsible for disclosure of results to patients, a view which was heavily influenced by lack of adequate funding to accommodate this process [144]. In contrast, of 18 researchers involved in pediatric biobanking from 10 different countries in the third study, most researchers wanted to keep connections with patients that had biospecimens and data stored to be able to provide them (or their relatives) with relevant information from ongoing research projects [146]. Interviews with 19 researchers from Saudi Arabia showed differing views: some thought returning SSR was a moral and professional duty whereas others felt returning results was not consistent with the goals of medical research [148].

## 1.7 Institutional Review Boards' views on returning study-specific results

**Clinical research setting.**   Five studies, all from North America, explored views of IRBs on the return of SSR [44, 149–152]. Generally, members of IRBs thought that research findings should be returned, provided participants want to know their results [44, 150]. However, while in one study 15/22 (68%) agreed that participants should 'probably' or 'definitely' be offered the choice to receive results that may be useful for participants' or family's health [151], another study indicated that a large majority of the 65 Chairs of US IRBs studied favoured offering the result to participants, even if there is a lack of clear utility associated with the findings [149]. One study explored views of a cohort of Canadian IRB members, coordinators, and chairs on returning IRR [152]. They found 50% (30/60) supported returning results when they indicated a probable medical condition or explanation for a response to a medication [152]. Interestingly, a study of 208 IRB professionals and 351 human genetic researchers showed that in two separate scenarios, IRB professionals were more likely to agree that individual research results should be returned to participants than researchers [44].

**Biobank setting.** One study addressed views of IRBs on the return of results related to study-specific results in the biobank setting. An Indian interview-based study of 21 ethics committee members and 22 researchers, showed that IRB members were unsure about their duty to review SSR and make decisions about their return [147]. They also held concerns about the need for a counselor or physician to convey research findings and participants suggested biobanks could have a coordinating centre that could assist interactions between sample donors and researchers, including playing a role in return of findings [147].

# 2. Views on return of unsolicited and secondary findings

## 2.1 Participants' preferences for receiving UF or SF

**Clinical research context.** Research participants generally expressed high interest in receiving UF, with values in most studies ranging from 61–100% depending on the nature of the UF [6, 11, 13, 66, 69, 71, 72, 87, 153–158]. Although participants were keen to receive all types of UF [77, 154, 158, 159], studies that asked respondents to specify which types of UF they wished to receive showed participants were most interested in receiving those that could have implications for their health [66, 156]. For example, a study of 58 adult volunteers in an ongoing family study of bipolar disorder showed that 97% wanted to receive UF that could have health implications [66]. However, 83% of the cohort still desired UF regardless of whether they were actionable. Similarly, responses from 219 adult parents of affected children who were receiving whole exome sequencing indicated that while 73% wanted to know all UF, this increased to between 93% and 97% for UF for which there was effective treatment and intervention, such as breast and ovarian cancer, hemochromatosis, arrhythmia, and cardiomyopathy [156]. However, these parents also expressed high interest in results that conveyed no personal disease risk information, such as pharmacogenetics (98%), carrier status (95%), and ancestry (96%). Positive views towards receiving carrier status have been shown by others [104, 155, 160].

A study of 362 parents of children with rare inherited childhood diseases or pediatric cancer also indicated elevated interest in UF for non-treatable fatal conditions relating to themselves (83%), which was only slightly lower than their interest in UF for fatal treatable/preventable conditions (87%) [6]. Yet, while 92% of this group wanted to know about UF for treatable conditions in their child, this dropped to 65% for non-treatable conditions and only 70% wanted to know uncertain results. This lower interest to receive SF for non-treatable conditions also seems to be reflected in adults. A study of 152 of 'cognitively intact adults' offered SF showed that while 76% of participants intended to learn some or all categories of SF and values ranged from 61–63% for receiving pharmacogenetic, carrier status, SNPs, and *APOE* variants, this dropped to 49% for rare, highly penetrant, unpreventable/untreatable, progressive conditions [155]. In contrast, of 149 parents of children recruited to studies investigating genetic causes and novel therapeutics for rare diseases, only 15% decided they did not want to receive UF that indicated risk of death in their children [77].

Studies indicate that participants (or their parents in the case of minors) want to receive UF for health-related planning and to be prepared [69, 87], to make lifestyle changes [69], out of curiosity [87] or because they are information seekers [69], because they have personal experience with health conditions, or because feel obligations to family members [87]. Some parents may also feel a moral obligation toward their children [75] or believe UF will provide them with a better understanding of the primary condition [100]. Interestingly, a study of 241 persons with mental disorders showed that although 91% thought that participants should receive UF, they believed that researchers should not actively search for SF [13].

However, three studies showed results that were not in line with the general trend. These included a study of 25 parents of children who had exome sequencing study of which only 50% wanted to learn UF relating to carrier status for children [11] and another which found that 16/23 (70%) parents were ambivalent about genomic sequencing, predominantly due to the possibility of UF [75]. The third exception was the study of 16 patient-participants with rare disease (or parents) who were offered the option to receive SF which identified mixed responses; while some wanted to receive all findings, regardless of actionability, others did not, particularly when they were for non-treatable conditions or of uncertain significance [161]. However, the interviews also indicated that these participants had an expectation that if they declined receiving UF now they would be offered the test again at a later stage. Parents have expressed not wanting to receive UF for their children because they are balancing the benefits and risks of knowing [69, 75], and small numbers of parents who were worried have raised concerns regarding insurance discrimination [69, 75], potential emotional/psychological impacts [69, 75, 104, 105], and anxiety brought on by secondary results [69].

Overall, there did not appear to be a difference between views of participants who were asked about hypothetical desires for UF and those who were actually deciding to receive these types of findings. However, a study of 155 individuals offered non-medically actionable SF as a second-tier test indicated that a lower proportion of participants actually requested to receive one or more categories (32%) compared to their decision immediately after their initial diagnostic disclosure consultation (76%) [105]. Another study of 223 participants of research studies using exome sequencing showed that preference for UF dropped from 76% at baseline to 65% after pre-test counseling, suggesting that receiving information and deliberation during the counseling process may change preferences for UF [87].

**Biobank setting.**    No studies explored participants' preferences for returning UF or SF in the biobank setting.

## 2.2 Patients' (and parents of patients') preferences for receiving UF or SF

**Clinical research setting.**    Only two studies explored parents' preferences for receiving UF for their children [95]. In interviews with parents of children affected by rare diseases, all 15 participants felt that knowing and receiving UF was empowering [95]. Parents also discuss wanting to be more aware of clinical risks for their child, to take responsibility for their child's health, and to allow them to plan (e.g., through proactive financial and health measures) and support their child [95]. However, parents also mention that they want their child's results because they have a right to know them and in order to exercise control over information that is relevant to their child with the goal of improving care [95]. Parents also want to know their child's carrier status so they can prepare their children for the future [95]. A Dutch study conducted 673 surveys and 146 interviews with patients [90]. They found that participants wanted to receive UF for curable (92%) and incurable conditions (76%), and also where the risk of developing the condition was high (84%) and low (79%) [90].

**Biobank setting.**    No studies explored patients' or parents' preferences for returning UF or SF in the biobank setting.

## 2.3 Publics' preferences for receiving UF or SF

Generally, members of the public expressed either substantial interest in receiving UF or strong beliefs that UF should be made available to research participants (or their parents) if they want to receive them [13, 62, 102, 121, 124, 162]. Interviews and focus groups with African Americans suggested they felt that professionals are obligated to disclose UF, even if may cause anxiety for the patient [102]. However, a study which included 1,623 blood donors

indicated that although 91% said UF should be made available to participants, they were willing to forego receiving UF if returning these findings might compromise the research [13]. Respondents highlighted personal utility [102] and determining how to help their child as reasons for wanting to receive UF [61].

However, some members of the public showed less interest in receiving UF. In a study of 4659 adult Americans, some questioned whether variants predisposing children to late onset conditions should be returned [62]. This was based on the idea that it may "do more harm than good" to return inconclusive or non-actionable findings and publics felt that parents should have the option to opt out of this information. Similarly, a study of 800 Australians showed that although most participants wanted to receive some UF, less wanted to receive all types of UF [121]. Two studies suggested that researchers should not actively search for these findings, as was also expressed by participants [13, 124].

## 2.4 Health professionals' views on returning UF or SF

Only three studies have assessed health professionals' views and experiences returning UF or SF to participants [163, 164]. In one study, which interviewed 21 genetic and non-genetic clinicians returning results of GS as part of the NIH funded Clinical Sequencing Exploratory Research (CSER), the clinicians stated that in many cases patient-participants were disappointed when no SF were identified and excited when they were. However, this was not always the case and some were surprised by the identification of SF and unclear about what they meant for their health risks [164]. In contrast, interviews with 19 professionals, which included medical doctors, and genetic counselors presented a more cautious approach to both screening for and returning SF [163]. This was based on a view that the current evidence was lacking and also a concern about limited availability of resources. Similarly, a study which included 533 genetic health professionals and 843 non-genetic health professionals showed that genetic health professionals were five times more likely than the public to think that UF should not be returned and three times more likely than the public to think that genomic researchers should not actively search for SF irrelevant to their research [124].

## 2.5 Researchers' views on returning UF or SF

To date, most of the studies assessing researchers views of returning UF are hypothetical with few researchers having experience of actually returning UF [165, 166]. Some articles included perspectives of those who had and had not returned UF/SF. For example, in a study of 198 investigators whose research focus was human disease gene identification, only 16.7% had returned UF, although 28.8% said they planned to disclose UF in future research studies and 20.2% said they planned to disclose to previous participants [167]. Likewise, in another study, 12% of 234 surveyed researchers had returned UF with another 28% intending to do so [168]. It was not always possible to establish how many of the respondents had experience returning UF/SF. This was the case in a study of 44 researchers presented with a vignette about identifying a UF which has an increased risk for colorectal cancer, 38 of whom (86%) said that they would disclose UF [169]. Interestingly, genomics researchers (6/34, 18%) were significantly less likely than medical geneticists to report a feeling of responsibility to examine the data for incidental clinically relevant UF (15/26, 57%) [134]. However, if UF are identified 68% (50/74) felt participants have a strong right to receive them [134]. In a study of 166 stakeholders, which included 19 genetic researchers and 33 clinical/laboratory geneticists, professional stakeholders seemed more cautious about the extent that UF should be disclosed to parents than members of the public, especially if there is uncertainty related to the findings [162].

Overall, there were very high levels of agreement that participants should be given the option of deciding whether they wish to receive UF [134, 167, 168] with some studies showing consensus [170], or near consensus on this point [171]. In a study of 20 basic researchers and clinical oncology researchers in the Quebec/Ontario adult cancer research community, participants stated that they felt a moral duty to identify and communicate UF, even if they were not discussed when participants provided consent [170]. Several studies explored researchers' reasons for returning UF [148, 165, 168–170]. Researchers raised concepts including UF providing benefit to participants (e.g. for their health or more broadly) [165, 169], minimizing harm [169], such as by avoiding participants' potential anger at developing a condition they were not informed about [170], and respecting participants' desires for information [165]. Researchers also discussed participants' rights to receive UF and having a moral obligation to return information that could be lifesaving [168].

However, some researchers have stated that they were unsure and conflicted on the issue, attempting to balance the pros and cons of returning UF [168]. Researchers have expressed concerns that the research infrastructure and study design to confirm or return UF are generally lacking [147, 170]. More specifically, they hold concerns about the quality of the test, the risk for false positive results, and the predictability, reliability and validity of the findings [148, 170]. Of 198 researchers surveyed, many rated return of UF as a moderate (66%) or significant/heavy (38.8%) burden [167].

Regarding views on criteria for returning UF, actionability of the findings was highlighted as a prerequisite and researchers flagged a need for clinical validation and presence of treatment options associated with the finding [170]. Similarly, 95% of 234 surveyed researchers said that highly penetrant and clinically actionable UF should be returned, although 15% said that researchers should return all UF [168], with high quality of information and clinical utility also being listed as main factors by others [169]. Another study showed that researchers favoured a case-by-case determination of whether or not to return UF [166].

Researchers reasons for not returning UF include the uncertain clinical utility of the findings [165], a lack of expertise in identifying UF [168], concerns that that participants will misunderstand results [165] and that UF might have negative emotional impacts on participants if not returned in a supportive manner [165, 168, 170]. Practical considerations were discussed, such as the costs and time constraints associated with returning UF, a lack of guidelines and resources on the issue, being ill-equipped to handle the data analysis and delivery of UF [170], and a need to ensure access to trained clinicians [165]. They also raised concerns relating the potential for loss of confidentiality [165] and issues relating to difficulties with insurance [170]. Researchers also suggested distinct goals between the clinical setting and research, which aims to generate new knowledge [166].

### 2.6 Institutional Review Boards' views on returning UF or SF

Five studies, four of which were based in North America, have shown that IRBs have varying degrees of experience with assessing protocols that discuss return of unsolicited findings [147, 150, 152, 166, 172]. Two studies indicated that, at the time of study, very few of their respondents had actual experience with unsolicited findings [150, 166], and another study of Canadian IRB members, IRB coordinators, and IRB chairs showed that 40% (24/60) had experience in evaluating protocols that involved the communication of UF [152]. An Indian interview-based study of 21 ethics committee members highlighted that most had encountered the possibility of UF in their reviews of research protocols and were unsure how to deal with these situations [147]. However, a large study of 796 IRB members and other IRB professionals reported that 74% had experience dealing with genetic UF [172]. Of these, the majority (65%) felt that

there was "sometimes" an obligation to disclose UF with only a small proportion (13%) saying there was "always" an obligation to disclose UF [172]. Their rationale behind disclosure of UF was because they felt there was a duty to warn participants if they are in significant, imminent danger, because it respects the autonomy of participants, and because of the potential benefit [172]. However, 96% indicated that it is either definitely or probably acceptable for a participant to elect not to receive any UF [172].

A study of 34 Chairs of IRBs in institutions in which Genome Wide Association Studies (GWAS) had been conducted suggested that IRB chairs view genetic UF as different to other types of findings due to the uncertainty relating to both the significance and potential of the findings, as well as the broader implications for family members and potential for social implications, such as discrimination and stigma [173].

When determining whether UF should or should not be returned from research studies, consideration of risks to participants was highlighted as an important factor, as were current regulations [152, 166]. IRB chairs preferred procedures for disclosure to be decided upon prior to researchers seeking ethics approval, rather than on a case-by-case basis [166]. Factors such as the additional time and effort required for the researcher to disclose UF [172], a lack of resources [172], and disclosure based on the concept of reciprocity [152] were not viewed as important considerations in determining whether UF should be returned. Although a study suggested that the financial cost of communicating UF was not a strong reason for not communicating these findings, they identified that IRB members were more likely to consider the financial costs if they were more experienced in reviewing genetic/genomic protocols, particularly if they involved returning UF [152].

## 3. Experiences with receiving or returning results

A summary of the experiences of participants, patients and public with receiving results is provided in Table 6. A summary of participants', patients' and publics' perceptions of utility and behavioural change is provided in Table 7.

### 3.1 Participants' reactions to receiving results

**Clinical research setting.**   Many of the studies assessing participants' responses and outcomes of receiving IRR were quite positive. Studies have shown high [87] or moderate to high satisfaction with decisions to receive results [8, 174]. Ten studies that asked participants whether they regretted participating in the research showed that either most [175, 176] or all of the participant reported little [155] or no regret at taking part and receiving IRR [79, 87, 174, 177–181]. In fact, a study of 202 participants–comprised of roughly half healthy adults and half patients with either hypertrophic cardiomyopathy or dilated cardiomyopathy–who received health information based on either family history alone or family history and genomic sequencing information, found that those who received genomic results reported lower average levels of decisional regret [182]. In addition, a study of 152 adults randomized to be offered SF found that participants who chose not to receive SF were more likely to feel regret than those who chose to learn SF [155]. However, one study of 117 adult participants of a ClinSeq Study in whom a variant associated with coronary artery disease risk was identified and returned showed that decision regret was significantly different between participant groups; those with VUS-low results (meaning that the evidence relating to the pathogenicity of these variants is approaching 'likely benign') reported greater regret than those with VUS-high results (where the evidence was approaching 'likely pathogenic') [183]. This suggests that uncertainty resulting from lack of information about the variants identified increased

**Table 6. Summary of experiences of participants, patients and public with receiving results.**

| Experiences of receiving IRR | Participants | Patients | Publics |
|---|---|---|---|
| Moderate to high satisfaction | [8, 87, 174, 184] | | |
| Relief | [9, 79, 87, 153, 175, 176, 185, 186] | | |
| Gratitude | [153, 176] | | |
| Increased knowledge | | [43] | |
| Decreased anxiety | | [43] | |
| High rates of wellbeing | [8] | | |
| Positive emotions | [187] | | |
| Low levels of uncertainty | [188] | [189] | |
| Low levels of negative emotions | [159, 188] | | |
| Low levels of concern | [190, 191] | | |
| No or low levels of distress | [83, 87, 188, 192] | [189] | |
| No impact on anxiety or depression | [31, 180, 193] | | |
| No impact of self-rated health | [176] | | |
| No adverse impact on quality of life | [175, 176] | | |
| No psychological harm | [83] | | [194] |
| Low perceived harm | [181] | | |
| No adverse impact on wellbeing | [193] | | |
| No adverse effects on emotions | [176] | | |
| No or little regret | [79, 87, 155, 174, 177–181, 188] | [189] | |
| Lower regret if chose to receive IRR | [155, 182] | | |
| No fatalistic reactions | [195] | | |
| Some distress or worry | [80, 176, 180, 186, 196, 197] | | |
| Some anxiety and depression | [175, 179, 198–201] | | |
| Lower positive feelings | [176] | | |
| Disappointment | [175] | | |
| Concern | [83, 175, 199] | | |
| Guilt | [180, 202] | | |
| Indifference | [175] | | |
| Uncertainty and confusion | [175, 176, 180, 182, 203] | | |
| Increased perception of risk | [80, 83] | [204] | |
| Desire for more results | [174, 175] | | |

**Table 7. Summary of participants', patients' and publics' perceptions of utility and behavioral change.**

| Perceptions of utility | Participants | Patients | Publics |
|---|---|---|---|
| Value in having an answer or a name | [174, 177] | | |
| Potential for surveillance, early disease detection, access to treatment | [9, 87, 153, 177, 182, 198] | | |
| Knowledge for children and ability to share information with family | [87, 153] | | |
| Reproductive planning | [177] | | |
| Empowerment and greater sense of control | [87, 174] | | |
| Benefit to science | [177] | | |
| **Behavior change** | | | |
| Some behavior change | [9, 31, 100, 186, 200, 205] | | |
| No or minimal behavior change | [8, 79, 87, 183, 198, 206, 207] | | [194] |

decisional regret [183]. Returning results to parents led to greater satisfaction as did offering a choice about which results to receive [184].

Most participants reported either positive or neutral impacts of receiving results, both when they were related to the study and also for UF/SF [9, 65, 79, 153, 157, 159, 174–176, 178, 179, 181, 182, 185, 187, 190, 191, 195]. A study of 17 research participants and family members who received UF showed that most (16/17) found the process mainly positive or useful and were thankful for being told they have the disease, both for their own wellbeing and also because it provides valuable knowledge for their children [153]. Another study of 31 adult participants in the ClinSeq cohort found that participants expressed relief that the result did not suggest a more serious condition, reassurance about their current healthcare, and satisfaction that they were able to access surveillance [9]. Most participants in the study (27/31) reported that their feelings about their result became more positive over time. This finding was supported by another study of 133 individuals who received testing for the *CDKN2A* gene for melanoma, where both carriers and noncarriers reported greater levels of hopefulness 6 months after disclosure than beforehand [176].

Similarly, although reports from 223 participants suggest modest impact from receiving IRR, they too expressed relief, either that an increased risk was not identified, or that the results were not indicative of more severe health consequences [87]. Interviews with 12 individuals from the Integrated Personal Omics Profiling project showed that while the majority of participants were underwhelmed by their results, several expressed feelings of validation or closure when their results could play a role in an existing health condition [79]. Other studies have also identified relief [79, 176, 185], gratitude [176], feelings of greater control [174], and that receiving results was valuable [174] and would influence medical treatment [182]. Additionally, assessment of 10 scientists and researchers who received results from genomic sequencing showed no apparent adverse events or reactions from disclosure of SF; variants were found in 9/10 participants [208]. There was also no evidence of adverse effects on self-rated health, quality of life, or emotional experience among either carriers or noncarriers tested for *CDKN2A* melanoma risk 6 months after disclosure [176].

However, some studies identified negative or mixed emotions from receiving IRR [174–178, 182, 196, 202]. A study of 10 women who had participated in the Australian Ovarian Cancer Study to determine prevalence of *BRCA1* and *BRCA2* mutations and 15 next of kin showed that interviewees had mixed responses to receiving feedback; many of the relatives were initially distressed, particularly if they had not realised their mother had participated in the study [196]. Another study of 3 adults diagnosed with invasive breast cancer in whom *BRCA1* or *BRCA2* mutations were identified and then returned also showed mixed reactions, including some guilt, although one year after confirmatory testing all of the interviewees considered that there were more advantages than disadvantages to receiving the information [202]. Individuals identified to be carriers of the at-risk *CDKN2A* variant for melanoma (n = 15) reported higher distress, higher uncertainty, and lower positive feelings immediately after receiving their result than noncarriers [176].

A few participants in the study of 35 individuals who underwent whole genome sequencing expressed negative reactions, such as concern, disappointment, indifference, confusion, and a desire for more results [175]. Similarly, 54% of 543 healthy participants from within four projects of the PeopleSeq Consortium were disappointed that their results did not tell them more information [174]. Participants from the study of 202 healthy adults and patients with either hypertrophic cardiomyopathy or dilated cardiomyopathy who had received genetic results were more likely to feel they had received a lot, or even too much information compared to those who received family history information alone [182]. They were also generally less satisfied with their understanding and felt lower levels of confidence in their ability to explain results to family

members [182]. Interestingly, a study of 24 individuals who participated in a genotype driven study on cystic fibrosis (9 participants with CF and 15 from a biobank cohort) showed that biobank participants were usually less sure than CF patients about why they had been selected and how they should conceive of themselves and their health, suggesting that being recruited based in genotype may create uncertainty for participants [203].

Some studies have suggested aspects across a range of different domains of utility that participants feel they gained through return of IRR. Participants appreciate the potential for surveillance and the ability to seek targeted medical care, which can lead to early detection and/or disease prevention, both for themselves and also their family members [9, 87, 198]. They also express empowerment from the knowledge of the genetic cause of their condition or, in the case of other information, such as carrier status or pharmacogenetic information, highlighted the importance of sharing this with family [87]. A study of 18 volunteers who agreed to have a genetic test for the *FTO* gene related to obesity believed that knowing their result would motivate them to try to control their weight in the future [195]. Another study showed parents valued having an answer and being able to put a name to their child's condition, as well as the ability to predict and manage their child's future health [177]. They also discussed the benefits of reproductive planning for any family member and being able to help science [177]. Focus groups with 24 members of the Hmong community in the USA who received pharmacogenetic results showed that as well providing benefit through allowing for changes in their medication, participants also identified that returning results benefited the broader community [187].

In relation to psychological outcomes, several studies have shown low levels of distress [87, 192], low levels of concern [190, 191], low levels of negative emotions [159], and high rates of wellbeing from receiving IRR [8]. In Sanderson et al's 35 healthy adult participants, no changes were detected in anxiety, depression or quality of life in response to receiving results between baseline and later measures [175]. Similarly, a project that tested 19 adult participants with a personal history of melanoma for genetic susceptibility to the condition showed no significant psychological harm from disclosure [83]. Event-specific distress was also low in these participants. Despite this, mutation carriers reported greater subjective concern about test results and also perceived their risks for another melanoma to be higher than non-carriers. In addition, a study of 13 adult family members of deceased men with early-onset prostate cancer who had participated in a study and who had been found to have a *BRCA2* mutation showed that some participants experienced distress and anxiety, although this was resolved through genetic counseling [198].

A number of studies have explored psychological outcomes in response to receiving *APOE* results in various iterations of the REVEAL study [80, 185, 192, 193, 199, 201, 209, 210]. These studies have explored responses to receiving versus not receiving genetic risk results, receiving deterministic versus susceptibility information, and also to different ways of disclosing results. The initial publication from this study, reporting on 162 adults who had a parent with Alzheimer's Disease (AD) suggested genotype disclosure did not adversely affect participants' psychological wellbeing, or lead to anxiety or depression with group means of these measures rating well below clinical cut off scores [193]. A subsequent study comparing disclosure of results of susceptibility testing within the REVEAL study (n = 101) versus deterministic testing in another study by University of Washington (n = 22) showed similar low levels of distress in both cohorts [192]. Another study where 162 participants were randomized into disclosure (n = 111) and non-disclosure (n = 51) arms showed no significant differences in distress between the two arms, yet did report differences between *APOE* e4+ and e4- at 6 weeks, 6 months and 12 months [80]. Although participants reported higher likelihood of perceived risk and an overall negative effect of disclosure in the *APOE* e4+ group, they were no less likely

to say they would retest than the *APOE* e4- group and attributed their psychological state to factors other than the results [80].

A study comparing extended, in-person *APOE* result disclosure with a condensed disclosure process suggested that 45% of their 269 participants reported an increase in depressive symptoms although only 9% were above the clinical cut off at 6 months [199]. Reports of increased depression at 12 months were associated with lower causal attribution to genetics and higher perceived risk of developing AD and level of concern decreased significantly more in those who received e4- results than e4+ [199]. In a study that compared outcomes from disclosing risks for AD alone with AD plus Coronary Artery Disease (CAD), 24% of participants reported moderate anxiety, depression, or test-related distress at one or more follow-up time points with no difference between the AD-only and AD+CAD groups [210]. In this study, mean distress scores and anxiety were greater in participants that received e4+ results but only for those in the AD only group. Another study that included 111 participants with mild cognitive impairment who were randomized into disclosure (n = 75) and non-disclosure arms (n = 39), showed that mean anxiety and depression scores in both arms were below clinical cut-offs at all time points [209]. Those in the disclosure arm who were e4- had lower test-related distress and greater positive impact than those who were e4+ [209]. Interestingly, individual scores for anxiety, depression, and hopelessness were more likely to be above clinical cut-offs, indicating cause for concern, if they were in the non-disclosure than the disclosure group [209].

Several studies have also assessed behavioral change in response to IRR. Most of these studies have shown little to no behavioral change [8, 79, 83, 87, 198, 206, 207] nor intentions to change behavior [183]. For example, disclosure of IRR in participants of the melanoma study had very little impact on motivating participants to adopt more prevention behaviors, such as wearing long sleeve shirts or pants when exposed to the sun or performing self-examinations, even in those who were identified to carry pathogenic variants in *CDKN2A* [83]. Family members of men with early-onset prostate cancer were less likely to engage with information and have screening if they were sceptical about the relative that informed them of their risk, if they were younger, and if they were afraid of cancer [198]. Although a study of 107 men and women from families with a known MMR gene mutation (which increases cancer risk) showed a slight increase in screening measures among participants, irrespective of test result, this was not significant [8].

Yet, participants do appear to take some steps based on receiving results. More participants in the MMR study who received an IRR indicating a pathogenic mutation took further steps to discuss screening than those who were negative [8]. In a study of 68 individuals without cardiomyopathy who had received VUS for a cardiomyopathy gene, 15 went on to engage in health-related behavior change: nine had cardiac testing (for some of whom there were clinical indications) and 12 made lifestyle changes [205]. The study of healthy individuals who were randomized to receive testing for either AD+CAD or AD alone showed that 57% reported changing at least one health behavior at 12 months in response to genetic risk disclosure [210]. Although this was more likely in the AD+CAD group than AD alone, this finding was independent of genotype [210]. In the ClinSeq cohort, 25/31 participants sought information about the variant and its associated health condition online after receiving their result [9]. Yet, of the seven healthy participants that received likely pathogenic or pathogenic rare disease associations, only 2 had acted on their results [175].

### 3.1 Participants' reactions to receiving results (continued)

**Biobank setting.** Four studies explored the outcomes from returning IRR to biobank participants [204]. Surveys were conducted with 55 participants who were part of a cohort of

people with pancreatic cancer who were tested for variants in *CDKN2A* for melanoma risk, explored the outcomes from returning SSR to biobank participants [204]. They found that *CDKN2A* carriers worried more about developing pancreatic cancer than non-carriers at pre-disclosure, immediately post disclosure, and also 6 months later [204]. In addition, more carriers thought they had a higher than average chance of developing melanoma at predisclosure and 6 months post-disclosure than non-carriers, which remained significant after excluding those with history of melanoma [204]. An Estonian study that returned pathogenic variants in BRCA1/2 genes to 22 biobanks participants found they generally felt calm and relaxed following genetic counselling, although a small number of participants reported feeling tense or worried [197]. A second Estonian study, in which 16p11.2 CNVs were reported back to 5 carriers, showed most were relieved and were coping with the information, although two said they were slightly worried [186]. In addition, surveys with ten thoracic aortic dissection biobank participants in the USA who received pathogenic variants showed low levels of psychological distress, negative feelings, uncertainty and privacy concerns [188].

In the two studies that explored behavior change in response to results was generally poor [186, 204]. Most participants in the *CDKN2A* did not have their results confirmed by a clinical laboratory, less than half had enacted pancreatic checks, and less than a third had had a skin test by 6 months follow up [204]. In the Estonian study that reported UF, two respondents visited a genetic specialist and this led to a treatment modification for one participant [186].

## 3.2 Patients' (and parents of patients') reactions to receiving results

Only two studies have examined the outcomes from returning results to patients [43, 189]. A study of 107 *BRCA1/2*-negative women with early-onset breast cancer, multiple primary cancers, or a family history of breast cancer showed that knowledge increased significantly after predisclosure counseling and receipt of results, including knowledge in those who received negative results or VUS [43]. Psychological assessments indicated that general anxiety and intrusive cancer-specific distress declined significantly for participants who received positive and negative results, as did depression [43]. However, cancer worry only declined significantly for those with a negative result [43]. An Australian study that recruited 133 participants from the Cancer Council registry, which included cancer patients, their family and friends, and members of the general public, showed that 95% of participants reported 'never' feeling regret about receiving their risk information [189]. Although the mean total scores of distress and uncertainty differed significantly depending on the risk category–the group at high-risk reported the highest mean Multidimensional Impact of Cancer Risk Assessment (MICRA) scores–scores were relatively low overall [189].

## 3.3 Publics' reactions to receiving results

Only two studies have investigated outcomes from returning results to members of the public. The first is the Australian study of 133 participants from the Cancer Council registry described above, which also included cancer patients and their family and friends [189]. The second reported on 280 cognitive healthy, Chinese English-speaking participants in Singapore that had *APOE* genotype testing [194]. Assessment using the Beck Anxiety Inventory (BAI) and Centre for Epidemiologic Studies Depression scale (CESD) showed no evidence of *APOE* genotype disclosure resulting in adverse psychological outcomes [194]. In addition, they did not identify any significant associations between *APOE* e4 genotype and behavior changes, such as diet, supplement consumption, and physical or cognitive activity [194].

### 3.4 Health professionals' experiences returning results

Only two studies assessed health professionals' experiences with returning individual research results to participants. One study conducted surveys (n = 21) and interviews (n = 22) with health professionals who had been involved with the eMERGE study [178]. While most participants thought sequencing results were important to participant's health, nearly half held concerns about inflicting harm through unnecessary investigations [178]. Many lacked confidence in their ability to explain the results to patients (72%) and to answer any questions they might have (78%). Interviews with 21 genetic and non-genetic clinicians suggested variable reactions by patients to receiving individual research results [164]. While some patients showed enthusiasm and relief at receiving IRR, others expressed confusion and disappointment; this difference depended on the results they received, their health status, and also their existing healthcare experiences. While some patient-participants felt it was good to have an answer, it could also be challenging to receive a result when the genetic basis indicates that the condition is progressive or worse than anticipated. This may be exacerbated when the condition is rare, meaning that access to information and support groups is limited. Responses to negative results (i.e., when no answer is identified) also varied from disappointment to relief, depending on the expectations that patients held for a result being identified. Finally, in relation to patients' reactions to receiving VUS, health professionals described difficulties getting patient-participants to understand the results and overinterpretation of the significance of the variant in some cases. Similar to receiving a result that identified the genetic basis of their condition, patients could have either positive or neutral reactions, depending on their diagnostic odyssey.

## Discussion

To our knowledge, this is the first systematic review to focus exclusively on stakeholder perspectives on return of results from genomic research. Overall, interest in receiving SRR was high across all stakeholder groups, particularly in the biobank setting. Although interest in receiving UF/SF was also quite high across all stakeholders, some members of the public did indicate less interest in receiving such results, questioning whether returning inconclusive or non-actionable findings, such as variants predisposing children to late onset conditions, might do more harm than good [62].

Interest in results was highest both when they were health related and when they were what could be considered to be 'actionable', i.e., the information could lead to some form of prevention or treatment [3, 4, 10, 33, 34, 39, 41, 42, 112, 155]. The concept of 'actionability' is often used to suggest that a result has clinical utility, i.e., that the result has the potential to influence patient management [211]. However, the exact meaning of this remains highly contested, despite considerable attention in the literature [212–214], making it difficult for researchers to determine what types of results might be appropriate to return, and when. We suggest that examining the reasons that patients and participants wish to receive results may provide us with another lens for assessing which results to return. Studies showed a wide range of reasons for wanting study-specific results and UF. Many of the reasons for desiring IRR related to the potential for individual health benefits, such as having greater perceived certainty about their personal level of risk [8, 83], or that of their child [95], determining whether they require screening [8] or medical interventions [42], health-related planning [87], and being able to prepare for their child's future heath needs [11]. However, some express desires to be mentally prepared and to be proactive in planning their finances, including insurance and housing [42, 95].

Others go beyond clinical utility and into the realms of what could be considered 'personal utility'. Participants describe feelings of parental responsibility or moral obligation, a perceived right to know their child's results, and desires to inform their children of their results [11, 75,

83, 95]. Finally, some just want to know results out of curiosity [48, 83, 87], or a need for control [11]. A study by Kohler et al used a Delphi method with participants of the ClinSeq cohort to delineate 14 types of personal utility [215]. These include 'mental preparation', 'ability for future planning', 'feelings of responsibility', 'to enhance coping', and 'curiosity', which succinctly encompass the reasons identified by our analysis.

Yet, it is important to distinguish personal utility from 'perceived utility', where those undergoing testing may believe that there might be some benefit from receiving the result when, in fact, there is not because the information is not valid or reliable [216]. As such, whether a result has personal utility or merely perceived utility can only be determined on a result-by-result basis [216]. While the question remains as to whether results that have bona fide personal utility should be returned in the context of genomic research (with the resourcing issues this entails), these results suggest that grouping results based on actionability (depending on how it is defined) may be less helpful than considering a broader concept of utility. Whether the IRR are study-specific or beyond the research question may also need to be considered.

It was striking that the position from which IRR are judged (i.e., positionality) and how IRR impact one's particular circumstances (i.e., situatedness) [217], both strongly shaped the perspectives of stakeholders. While members of the public also desire results to improve both current and future health [56, 91, 93], there is some evidence that they are more interested in receiving information about non-medical traits and variants of uncertain significance than patients or participants [91]. Their rationale that these types of results might be meaningful later on or may reveal something about them or their family may reflect the fact that they are representing a 'lay' perspective, which is in contrast to those who are participating in genetic research because of their disease status. In addition, although general willingness to receive IRR was high, several studies showed that some members of the general public were concerned about receiving results because of its implications for insurance discrimination and the potential impact of the findings on their mental health [56, 59, 91, 93]. It is interesting that these concerns did not arise in studies of patients or participants, which may again be due to the fact that they (or their child) are already affected with a genetic condition and therefore may be more focused on receiving information that may promote their current health than concerns for their future health [218].

It is also important to highlight the ways in which both the framing of questions and the opportunity for deliberation may influence interest in receiving IRR. Our review identified instances where the findings of qualitative studies with participants or members of the public showed lower support for receiving IRR than those that were quantitative in nature [91]. In addition, several studies showed that while interest in receiving IRR remains relatively high, there was a decrease in interest after either counselling or some form of educational intervention [87, 105, 114]. Furthermore, a recent study has shown that proportions of participants who were informed that a new genomic finding had been identified chose not to proceed with variant confirmation and detailed disclosure [219, 220]. These findings support the need for education and support for participants during the consent process to ensure their decision is informed. More research into the ways in which this can be achieved is required.

Importantly, our analysis shows that many of the studies assessing participants' actual experiences of receiving individuals research results were quite positive. A proportion of these studies spoke to different domains of utility that participants felt they gained through return of results, including the potential for surveillance and the ability to seek targeted medical care, which in turn could lead to early detection and/or disease prevention both for themselves and family members [9, 87, 198]. They also expressed empowerment from the knowledge of the genetic cause of their condition [87]. Several studies assessing psychological outcomes of actual experiences around returning results (both qualitative and quantitative in nature) showed low

levels of negative emotions [159], low levels of distress [87], no significant psychological harm [83], no fatalistic reactions [195], and high rates of wellbeing from the receipt of individual results [8]. One study also found that disclosure of results may reduce anxiety, depression and feelings of hopelessness, suggesting that uncertainty may be more harmless than disclosure [209]. Although some studies identified negative or mixed emotions from returning results [174–178, 182, 196, 202], importantly, one study in which some participants experienced distress and anxiety, suggested that this resolved with genetic counseling [198]. We should note that we did not assess the processes by which IRR were returned within this review and the level of support provided in the return process are likely to impact on how well IRR are received, as raised as a concern in several studies [165, 168, 170].

In contrast to views of participants, patients and members of the publics, healthcare professionals, researchers, and members of IRBs seemed overall to be less positive and more cautious about returning results [44, 124], though there were fewer studies that assessed the perspectives of these stakeholders. They were generally more supportive of returning results that are reliable and have clinical relevance and utility, rather than results where the significance or utility of the findings is uncertain [76, 134, 135, 138]. In contrast to potential recipients of genomic results, the key concerns expressed by professionals mainly related to the uncertainty of the results being returned and the blurring of lines between research and clinical care [139, 140]. Concerns about results instilling fear in participants were also raised [145]. The fact that professionals raise different concerns to potential recipients of results of genomic research is perhaps unsurprising given the more cautious approach that genetic health professionals take to providing testing in other contexts [221, 222]. It may also be reflective of the role they play in determining which results should be returned, particularly if they are involved in the analysis/interpretation process. It should also be noted that ability and even willingness to return results by researchers and health professionals in certain studies may be dictated by country-specific regulations. While it may be unsurprising that parents care about the impact on their child's health or that researchers are concerned about the resource-based of returning findings, the importance of these contexts for shaping return of IRR cannot be understated. In particular, the finding that IRB members who are not experienced in reviewing genomic protocols were less likely to consider the financial costs of returning IRR, highlights how critical it is to involve those who have experience in this area in protocol development [152].

Only two studies focused on actual experiences of health professionals returning IRR [164, 178], and no studies focused on experiences of researchers or IRBs, although several studies included proportions of stakeholders that had done so. The study by Wynn et al. reported similarly mixed responses by participants to receiving results from genomic research, which were often dependent on the type of result they received, their health status, and their previous experiences in the healthcare system, including the duration of their diagnostic odyssey [164]. Further research could be undertaken to explore professionals' experiences with returning IRR in order to help develop the most appropriate infrastructure and resources required to return IRR.

This review has certain limitations. The majority of relevant studies were conducted in the USA, United Kingdom or Canada and were comprised predominantly of White participants. Participant views are likely to be heavily influenced by what is standard practice in the clinical setting and therefore dependent on the healthcare system, and availability of health insurance, in their region. As such, the perspectives identified cannot necessarily be generalized to other cultural groups or national healthcare contexts and should be used with caution in the development of regulation for unrepresented groups. The vast majority of studies are from the USA which is important given they are currently the only nation to have guidelines that endorse actively searching for SF in the clinical setting [223]. Several studies with different types of stakeholders did not present separate findings for each stakeholder group. Where this was the

case, the results of the study were reported within the most prominent stakeholder group or included in the 'mixed professional' category where there was no prominent stakeholder group. At times this made it difficult to determine whether studies were discussing just study-specific results or all IRR, as some papers appeared to conflate the two. In such circumstances, categorisation was led by the context in which the stakeholders were being asked the question. We also acknowledge that our quality assessment approach did not prioritize inclusion of studies based on study quality. Finally, some of the studies included are older and as such information about 'current' practices regarding return of results–particularly in relation to researchers, health professionals and IRBs–are likely to have evolved considerably.

## Conclusion

We have provided overwhelming evidence that, at least for the United States, Canada and some other countries, there is a high interest in return of results from the stakeholders who either are, or would be, receiving them. There is also a general willingness to provide such results by those who would be doing so, although the latter tend to adopt a more cautious stance. While all results are desired to some degree, those that have the potential to change clinical management appear to be more valued. In addition, professional stakeholders appear more willing to return results that are reliable and clinically relevant. Furthermore, the lack of evidence of enduring psychological harm from returning results suggests that doing so is appropriate.

As such, we feel it is time to move away from questions about whether individual research results should be returned and to focus on *which types* of IRR to return in each context and *how* to return them in an ethically appropriate and supportive manner. Gaps remain in our knowledge of how to achieve this. First, a large proportion of the studies identified gauged hypothetical views while very few used deliberative methodologies to help participants understand the implications of their decisions regarding return of results. Second, gaining a more comprehensive understanding of the motivations of different stakeholders will allow for feedback of results to be better tailored to the research study and context. To address these two points, if more research in this field is to be conducted it should involve individuals who are actually making decisions about receiving IRR and investigating ways in which to best support this process, both within the scope of existing large-scale projects and biobanks, and also in the planning of future research endeavours. Third, as three quarters of the papers were based on studies conducted in the United States and Canada, along with the fact that the limited studies conducted in other parts of the world suggest some cultural variation, more work will also be required to accommodate different cultural settings.

To ensure that return of IRR is done well across the board, best practices for the return of IRR are needed [22]. Clear pathways for return of results are required, as is training for those returning IRR and those obtaining consent to do so. There also needs to be adequate resources available to return IRR, including access to genetic counselling and other specialist expertise when required. We must think as well about the point at which the obligations of researchers to return IRR cease, particularly as reanalysis and reinterpretation processes become more automated. Finally, the development of practical guidelines and informative frameworks adaptable to all settings are needed to support researchers in putting appropriate return of IRR protocols in place.

## Supporting information

**S1 Table. Systematic search terms.**
(PDF)

**S2 Table. Data extraction form.**
(PDF)

**S1 Checklist. PRISMA 2009 checklist.**
(DOC)

# Acknowledgments

We thank the following members of the Return of Results Task Team of the Regulatory and Ethics Work Stream of the Global Alliance for Genomics and Health for their assistance in the data extraction process: Prof Toby Bloom, Dr Clement Chen, Dr Lea Harty, Dr Jessica Reusch, Dr Laura Lyman Rodriguez, Rosalyn Ryan, Dr Anne-Marie Tasse, Adrian Thorogood and Dr Susan Wallace. Finally, we thank Dr Anna Lewis for her helpful suggestions for the manuscript.

# Author Contributions

**Conceptualization:** Danya F. Vears, Joel T. Minion, Stephanie J. Roberts, Bartha M. Knoppers, Madeleine J. Murtagh.

**Formal analysis:** Danya F. Vears, Joel T. Minion, Stephanie J. Roberts, James Cummings, Mavis Machirori, Mwenza Blell, Isabelle Budin-Ljøsne, Lorraine Cowley, Stephanie O. M. Dyke, Clara Gaff, Robert Green, Alison Hall, Amber L. Johns, Stephanie Mulrine, Christine Patch, Eva Winkler, Madeleine J. Murtagh.

**Methodology:** Danya F. Vears, Madeleine J. Murtagh.

**Project administration:** Joel T. Minion, Stephanie J. Roberts.

**Writing – original draft:** Danya F. Vears.

**Writing – review & editing:** Danya F. Vears, Joel T. Minion, Stephanie J. Roberts, James Cummings, Mavis Machirori, Mwenza Blell, Isabelle Budin-Ljøsne, Lorraine Cowley, Stephanie O. M. Dyke, Clara Gaff, Robert Green, Alison Hall, Amber L. Johns, Bartha M. Knoppers, Stephanie Mulrine, Christine Patch, Eva Winkler, Madeleine J. Murtagh.

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
