## [Decision Letter · Decision Letter 0]

19 Apr 2021

PONE-D-21-02012

Return of individual research results from genomic research: A systematic review of stakeholder perspectives

PLOS ONE

Dear Dr. Vears,

Thank you for submitting your manuscript to PLOS ONE. After careful consideration, we feel that it has merit but does not fully meet PLOS ONE’s publication criteria as it currently stands. Therefore, we invite you to submit a revised version of the manuscript that addresses the points raised during the review process.

We look forward to receiving your revised manuscript.

Kind regards,

Prof. Ritesh G. Menezes, M.B.B.S., M.D., Diplomate N.B.

Academic Editor

PLOS ONE

Journal Requirements:

2. Please include your tables as part of your main manuscript and remove the individual files. Please note that supplementary tables should be uploaded as separate "supporting information" files.

Reviewers' comments:

Reviewer's Responses to Questions

**Comments to the Author**

1. Is the manuscript technically sound, and do the data support the conclusions?

Reviewer #1: Yes

Reviewer #2: Yes

Reviewer #3: Yes

Reviewer #4: Yes

2. Has the statistical analysis been performed appropriately and rigorously? 

Reviewer #1: N/A

Reviewer #2: Yes

Reviewer #3: N/A

Reviewer #4: Yes

3. Have the authors made all data underlying the findings in their manuscript fully available?

Reviewer #1: Yes

Reviewer #2: Yes

Reviewer #3: Yes

Reviewer #4: Yes

4. Is the manuscript presented in an intelligible fashion and written in standard English?

Reviewer #1: Yes

Reviewer #2: Yes

Reviewer #3: Yes

Reviewer #4: Yes

5. Review Comments to the Author

Reviewer #1: Although exceptionally long, this is a very solid manuscript that systematically reviews the available evidence about the return of individual genetic research results. It reads well and is very comprehensive. I had very minor comments, namely:

- The final search for articles was conducted in April 2020, which is now almost a year ago. The authors should be asked to run the search again and make sure that no articles were published in the meantime that challenge their main findings.

- The main (or initial) data extraction was conducted by a dispersed set of persons who were part of a working group. The extraction data sheet is provided and is thorough, providing some confidence in the approach followed. Furthermore, the lead investigators verified and clarified data extraction. The only piece of missing information is whether and how the data extractors in the taskforce were instructed. Was there an online meeting describing the form and explaining how to use it? This information could be added in a sentence or two;

- The analytical approach described in lines 224-234 needs to be justified better. Note that the one reference you use to support your analytical approach (reference number 21) is specific for systematic reviews in normative literature; yet you included in your systematic review also studies that presented qualitative and quantitative data. As far as I understand from your methods description, you didn’t only assess normative arguments made in those papers but you also looked at the data itself. And yet you did not adopt a systematic approach to analysing that data. This needs to be explained better and you need to cite a broader number of references that have used the approach described in reference 21.

- The sentences in lines 235-240 are not at all clear to me. One way of reading it is saying that ‘the way we present our findings in the paper will allow the reader to assess whether our argument is logical’ – is that what you mean to say?

- Did you use any software for data analysis? If not, what did you use? Would be good to add this in a few words.

- The reference to an article ‘in press’ (lines 1303-1305) is difficult to understand. Also, it would help if this reference was included in the numbered references. If this information cannot be provided for some reason, then this sentence should be removed.

- In the Conclusion, whilst I agree that you have found ‘overwhelming evidence’ that people desire to receive results, I think it is imperative that you add a clause indicating that this is so ‘at least for the United States, Canada and some other countries’. It seems a bit far-fetched to suggest that this is therefore the case for entire other continents where hardly any research has been done. You mention this in the Discussion before as a limitation but I think it deserves slightly greater prominence when you then make the kind of broad claim in the first paragraph of the Conclusion. Similarly, you indicate that the majority of people whose views are represented are white. This also seems to somewhat challenge the observation of ‘overwhelming evidence’ that people want result – there is overwhelming evidence that some kinds of people in some jurisdictions want results; what others want really hasn’t been investigated.

- I spotted a few typos, omissions etc (e.g. line 1337; 1349; 1354 with should be in; 1355 comma missing) etc.

Reviewer #2: Vears and colleagues present a thorough and detailed assessment of the topic of returning NGS dataset results to participants. The resources they use are carefully assembled with attention to detail and results are presented in an understandable and concise manner.

A few minor questions I would like to ask the authors to address:

1. Have the authors carefully considered the effects of country-specific regulations on the returning of results? Is the willingness to return results to participant a result of regulation mandating said return or does it come from other stakeholders? For example differences in privacy laws in the EU, US and other countries? Can and should this be evaluated on an individual country basis?

2. Are there any protection in the informed consents regarding potential false positive and/or negative results coming from the research study and protection of the researchers conducting the study to be immune from malpractice claims potentially arising from these false positive and negative results? Does this question have potential country-specific regulations aspects? Does it influence ‘willingness’ to return the results?

3. Are there differences in the results of the study presented in the manuscript whether NGS studies for Oncology vs. other indications are considered? Along similar lines, are there differences in Healthy donors vs. diseased individuals in requesting/being interested in obtaining results of NGS studies?

Reviewer #3: In general, the topic is interesting, but hard to compile all the results as there are different methodologies in each independent studies. Thus, the approach was clever and enable to draw conclusions about the high interest of stakeholders to be informed of their genomic results.

• However, I believe the information should be compacted. In some parts it seems the information is repeated. It maybe worth mentioning few but concise examples. Try not to repeat in the discussion the results but instead discuss the reason of obtaining these results.

• Mentioning many numbers in the text may not have a great impact as there are not any statistical analysis.

• In articles that have both clinical and research settings, was data separated or not? Does this have a relationship with the statement “Where only some of an article’s content was eligible for inclusion, only this data was extracted and included for synthesis”

• Can you explain why you are not considering “carrier screening or neonatal genomic screening, even when part of a research protocol?

• In Figure 1 in the screening section, the reason why 6,864 studies were excluded is not explicitly written as done in eligibility section. Numbers should be referred in the text.

• Number of articles removed or eligible mentioned for the two time periods should be reflect clearly in Figure 1.

• In the limitations: “Where this was the case, the results of the study were reported within the most prominent stakeholder group or included in the 'mixed professional' category”. You mean several studies, how many? If you assigned them randomly to the prominent stakeholder or mixed professional, does it make any difference for the conclusions?

• In some studies, the techniques used are stated whereas not in others. Did all studies had NGS data to inform, or some have only gene polymorphism data? Is it relevant to mention them?

• Check reference style some are only with number others with the author number.

Reviewer #4: The manuscript is interesting and very well written. The limitation of the study regarding cultural variability and research ethical norms out of North America, may be expanded a little bit. IRB standards variations of various countries might have influenced the research outcomes and discussion generated from selected countries.

6. PLOS authors have the option to publish the peer review history of their article (what does this mean?). If published, this will include your full peer review and any attached files.

Reviewer #1: No

Reviewer #2: No

Reviewer #3: No

Reviewer #4: No

---

## [Author Response · Author response to Decision Letter 0]

2 Jul 2021

Journal Requirements:

We have formatted the manuscript to meet PLOS ONE's style requirements, including those for file naming.

2. Please include your tables as part of your main manuscript and remove the individual files. Please note that supplementary tables should be uploaded as separate "supporting information" files.

The tables have been included in the manuscript and supporting documents are uploaded as supporting information.

We hope we have now resolved this issue.

Comments to the Author

Reviewer #1: 

1. The final search for articles was conducted in April 2020, which is now almost a year ago. The authors should be asked to run the search again and make sure that no articles were published in the meantime that challenge their main findings.

In line with the reviewers’ request, we re-ran the search. However, in order to be consistent with our systematic approach, that meant we also had to go through the process of assessing inclusion criteria with multiple authors, data extraction for each eligible article, which also had to be checked for accuracy by a second author. Data was then coded as per the methodology and incorporated into the relevant sections and tables within the review. 

Our updated search identified an additional 38 articles for inclusion. Additions are marked throughout using track changes, except for where additional references were just inserted to support points already mentioned. The methods section, figures and tables have been updated accordingly.

2. The main (or initial) data extraction was conducted by a dispersed set of persons who were part of a working group. The extraction data sheet is provided and is thorough, providing some confidence in the approach followed. Furthermore, the lead investigators verified and clarified data extraction. The only piece of missing information is whether and how the data extractors in the taskforce were instructed. Was there an online meeting describing the form and explaining how to use it? This information could be added in a sentence or two.

We have included the following in the methods section: “Data extractors were provided with a written overview of the study aims and criteria for article inclusion. They were instructed to complete all fields of the data extraction form for each of their allocated articles and, when entering the ‘Key findings’ section, to only insert data that was relevant to the research question.”

3. The analytical approach described in lines 224-234 needs to be justified better. Note that the one reference you use to support your analytical approach (reference number 21) is specific for systematic reviews in normative literature; yet you included in your systematic review also studies that presented qualitative and quantitative data. As far as I understand from your methods description, you didn’t only assess normative arguments made in those papers but you also looked at the data itself. And yet you did not adopt a systematic approach to analysing that data. This needs to be explained better and you need to cite a broader number of references that have used the approach described in reference 21.

The reviewer’s comments have highlighted an error in the structure of our methods section which has unfortunately conflated our analysis method (systematic content analysis) with our approach to quality appraisal (assessment of transparency, methodological appropriateness and coherence). We have remedied this by restructuring the section, indicating the influence of the reason-based approach (without the reference to the analysis of papers) and giving more detail about the quality assessment approach. Unfortunately, the previous paragraph on quality could be read to imply that we only assessed normative arguments or that we did not take a systematic approach to analysis. This was not the case. As our paragraph on analysis explains, we undertook a systematic content analysis of all data. As this was unclear, we have strengthened the analysis description by reference to a number of systematic reviews which also use the content analysis approach in complex systematic multi-method reviews (lines 270 – 296). 

4. The sentences in lines 235-240 are not at all clear to me. One way of reading it is saying that ‘the way we present our findings in the paper will allow the reader to assess whether our argument is logical’ – is that what you mean to say?

As above, we note the confusion produced by our conflation of quality and analysis. We have amended the paragraph for clarity and have removed the reference to the reader’s judgement as that is self-evident and its articulation is unnecessary.

5. Did you use any software for data analysis? If not, what did you use? Would be good to add this in a few words.

We just used Word documents. This has been added to the methods section, which now reads “Data were coded and interpreted by DV using Word documents; MM analysed subsets of the data to confirm the coding scheme.” 

6. The reference to an article ‘in press’ (lines 1303-1305) is difficult to understand. Also, it would help if this reference was included in the numbered references. If this information cannot be provided for some reason, then this sentence should be removed.

We have changed the sentence to read “Furthermore, a recent study has shown that proportions of participants who were informed that a new genomic finding had been identified chose not to proceed with variant confirmation and detailed disclosure.” As this paper is still not published, we have also added references to publicly available conference papers that presented this data.

7. In the Conclusion, whilst I agree that you have found ‘overwhelming evidence’ that people desire to receive results, I think it is imperative that you add a clause indicating that this is so ‘at least for the United States, Canada and some other countries’. It seems a bit far-fetched to suggest that this is therefore the case for entire other continents where hardly any research has been done. You mention this in the Discussion before as a limitation but I think it deserves slightly greater prominence when you then make the kind of broad claim in the first paragraph of the Conclusion. Similarly, you indicate that the majority of people whose views are represented are white. This also seems to somewhat challenge the observation of ‘overwhelming evidence’ that people want result – there is overwhelming evidence that some kinds of people in some jurisdictions want results; what others want really hasn’t been investigated.

We thank the reviewer for raising this important point. We have amended the sentence in the conclusion, which now reads “We have provided overwhelming evidence that, at least for the United States, Canada and some other countries, there is a high interest in return of results from the stakeholders who either are, or would be, receiving them.”

8. I spotted a few typos, omissions etc (e.g. line 1337; 1349; 1354 with should be in; 1355 comma missing) etc.

We thank the reviewer for picking these up. We have amended them and also proofread the manuscript again.

Reviewer #2: 

1. Have the authors carefully considered the effects of country-specific regulations on the returning of results? Is the willingness to return results to participant a result of regulation mandating said return or does it come from other stakeholders? For example differences in privacy laws in the EU, US and other countries? Can and should this be evaluated on an individual country basis?

We agree that the question of legal and regulatory context is key. While such review is beyond the scope of this systematic review, we note the recent and thorough review of international law on return of results by Thorogood et al which addresses these questions, finding that there is great unevenness in the international legal landscape. We have added a few sentences to our introduction to cover these issues and, in particular, note Thorogood et al’s conclusion that clarity is required in the ethical and policy approach to return of results (see below). This review contributes to that clarity by assessing the wealth of existing empirical research with stakeholders on these matters.

“The legal and regulatory landscape regarding return of results currently comprises a patchwork of often contradictory rules for researchers, especially where research collaborations stretch across countries and continents as many now do in the field of genomics. In their recent review, Thorogood et al identified sufficient discrepancies between policies to prevent reconciliation of rules about which results should or should not be returned in research projects (21). Moreover, they found that policies, including thresholds for data quality and clinical significance, were evolving in uneven ways, further complicating policy development for return of results. Thorogood et al call for greater clarity in the ethical and policy approach to return of results.” 

2. Are there any protection in the informed consents regarding potential false positive and/or negative results coming from the research study and protection of the researchers conducting the study to be immune from malpractice claims potentially arising from these false positive and negative results? Does this question have potential country-specific regulations aspects? Does it influence ‘willingness’ to return the results?

We thank the reviewer for raising this interesting question. However, we can only comment on the information that is provided within the papers included in the study and believe it is beyond the scope of the manuscript to try to access the informed consent forms for each of the studies on which the articles are based, some of which are 15 years old. We have, however, added a sentence to the discussion as follows: “It should also be noted that ability and even willingness to return results by researchers and health professionals in certain studies may be dictated by country-specific regulations.”

3. Are there differences in the results of the study presented in the manuscript whether NGS studies for Oncology vs. other indications are considered? Along similar lines, are there differences in Healthy donors vs. diseased individuals in requesting/being interested in obtaining results of NGS studies?

The question of whether here are differences between views on studies in oncology versus other indications is an interesting. However, there were relatively few articles assessing views on return of results in cancer patients so we do not feel that we can compare this in a meaningful way. Regarding differences between ‘healthy’ donors (in. which I assume the reviewer is also including members of the public) and individuals with medical conditions, as we state in the discussion on line 1528 “there is some evidence that they are more interested in receiving information about non-medical traits and variants of uncertain significance than patients or participants”.

Reviewer #3: 

1. However, I believe the information should be compacted. In some parts it seems the information is repeated. It maybe worth mentioning few but concise examples. Try not to repeat in the discussion the results but instead discuss the reason of obtaining these results. 

We have removed a lot of the repetition of the results in the discussion and only kept in information about the results to aid discussion of important points. We have tightened up the results section by cutting out some the examples but have tried to retain the nuances of all the studies, which we believe are important for the reader to appreciate.

2. Mentioning many numbers in the text may not have a great impact as there are not any statistical analysis.

We thank the reviewer for their suggestion and we agree that mentioning numbers in the text may have less impact than if were conducting statistical analysis. However, we believe including the percentages of participants and study numbers allows the reader to gain some perspective on the weight that the findings of each study should be afforded. For example, a quantitative survey of 5,000 participants may be viewed as having more impact than a survey of 50 participants. For this reason, we have elected to keep the numbers in the manuscript.

3. In articles that have both clinical and research settings, was data separated or not? Does this have a relationship with the statement “Where only some of an article’s content was eligible for inclusion, only this data was extracted and included for synthesis”

The reviewer is correct. In articles that included data relating to both research and clinical settings (of which there were very few) where it was differentiated, we only included data relating to the research setting.

4. Can you explain why you are not considering “carrier screening or neonatal genomic screening, even when part of a research protocol?

We elected to exclude both carrier screening and neonatal screening because ‘screening’, which is often population-based, and ‘testing’, which often relates more to the individual and their family are different contexts that would involve different result return mechanisms. As such, it seemed appropriate not to conflate the two.

5. In Figure 1 in the screening section, the reason why 6,864 studies were excluded is not explicitly written as done in eligibility section. Numbers should be referred in the text.

This has been added.

6. Number of articles removed or eligible mentioned for the two time periods should be reflect clearly in Figure 1.

This has been added.

7. In the limitations: “Where this was the case, the results of the study were reported within the most prominent stakeholder group or included in the 'mixed professional' category”. You mean several studies, how many? If you assigned them randomly to the prominent stakeholder or mixed professional, does it make any difference for the conclusions?

The assignment of articles to either the most prominent stakeholder group or the mixed professional category was not random and we apologise if we have that impression. We have changed the text to add clarity, which now reads “Where this was the case, the results of the study were reported within the most prominent stakeholder group or included in the ‘mixed professional’ category where there was no prominent stakeholder group.”

8. In some studies, the techniques used are stated whereas not in others. Did all studies had NGS data to inform, or some have only gene polymorphism data? Is it relevant to mention them?

Not all studies used genomic data. Generally, when the results related to gene polymorphism data this has been mentioned.

9. Check reference style some are only with number others with the author number.

We thank the reviewer for picking up on this discrepancy. Hopefully we have now changed any references listed by author to a number format.

Reviewer #4: 

1. The limitation of the study regarding cultural variability and research ethical norms out of North America, may be expanded a little bit. 

Rather than expand on this in the limitations section (because the manuscript is already very long) we have amended a line in the conclusion, which now reads: “We have provided overwhelming evidence that, at least for the United States, Canada and some other countries, there is a high interest in return of results from the stakeholders who either are, or would be, receiving them.”

2. IRB standards variations of various countries might have influenced the research outcomes and discussion generated from selected countries.

We have included a statement in the discussion that now reads “It should also be noted that ability and even willingness to return results by researchers and health professionals in certain studies may be dictated by country-specific regulations.”

Other

All figures were checked using PACE.

---

## [Decision Letter · Decision Letter 1]

24 Aug 2021

PONE-D-21-02012R1

Return of individual research results from genomic research: A systematic review of stakeholder perspectives

PLOS ONE

Dear Dr. Vears,

Thank you for submitting your manuscript to PLOS ONE. After careful consideration, we feel that it has merit but does not fully meet PLOS ONE’s publication criteria as it currently stands. Therefore, we invite you to submit a revised version of the manuscript that addresses the points raised during the review process.

Please submit your revised manuscript by 02-September-2021. Please include the following items when submitting your revised manuscript:

A 'Response to Reviewers' letter that responds to each point raised by the academic editor and reviewer(s). You should upload this letter as a separate file labeled 'Response to Reviewers'.A marked-up copy of your manuscript that highlights changes made to the original version. You should upload this as a separate file labeled 'Revised Manuscript with Track Changes'.An unmarked version of your revised paper without tracked changes. You should upload this as a separate file labeled 'Manuscript'.

We look forward to receiving your revised manuscript.

Kind regards,

Prof. Ritesh G. Menezes, M.B.B.S., M.D., Diplomate N.B.

Academic Editor

PLOS ONE

Journal Requirements:

Reviewers' comments:

Reviewer's Responses to Questions

**Comments to the Author**

1. If the authors have adequately addressed your comments raised in a previous round of review and you feel that this manuscript is now acceptable for publication, you may indicate that here to bypass the “Comments to the Author” section, enter your conflict of interest statement in the “Confidential to Editor” section, and submit your "Accept" recommendation.

Reviewer #2: All comments have been addressed

Reviewer #3: All comments have been addressed

2. Is the manuscript technically sound, and do the data support the conclusions?

Reviewer #2: Yes

Reviewer #3: Yes

3. Has the statistical analysis been performed appropriately and rigorously? 

Reviewer #2: Yes

Reviewer #3: N/A

4. Have the authors made all data underlying the findings in their manuscript fully available?

Reviewer #2: Yes

Reviewer #3: Yes

5. Is the manuscript presented in an intelligible fashion and written in standard English?

Reviewer #2: Yes

Reviewer #3: Yes

6. Review Comments to the Author

Reviewer #2: (No Response)

Reviewer #3: 1. Table 2 is very well summarised, but I considered the full citation is not needed, the abbreviated citation will be enough. In that way the table can be shortened.

2. The idea in line 256-257 is not clear

3. Check the verb in line 710

4. Label heading and subheadings with numbers to make a easy reading pattern

5.In lines 886-887 number are written and number as used as symbols better to be consistent through the text

6.In line 1117 the word “well-being” is separated by a high fen but not in Table 6 and other parts in the text

7. Some sentences must be split or they need to have more punctuation marks

7. PLOS authors have the option to publish the peer review history of their article (what does this mean?). If published, this will include your full peer review and any attached files.

Reviewer #2: No

Reviewer #3: No

---

## [Author Response · Author response to Decision Letter 1]

1 Sep 2021

Reviewers' comments:

Reviewer #3

1. Table 2 is very well summarised, but I considered the full citation is not needed, the abbreviated citation will be enough. In that way the table can be shortened.

While we thank the reviewer for their suggestion and agree that a shortened table would, in some ways, be desirable, in our opinion the context that is provided to the table by including the full reference (particularly the title) provides the reader with a valuable resource. As such, we believe it is in the readers’ interests that we leave it as it currently stands. 

2. The idea in line 256-257 is not clear

We thank the reviewer for picking up this error. The sentence now reads “We used content analysis to enable systematic analysis of the methodologically diverse articles in this review.”

3. Check the verb in line 710

We thank the reviewer for picking up this error. The sentence now reads “Researchers also suggested returning results shows respect for patient-participant autonomy and recognizes participants’ ownership of their data.”

4. Label heading and subheadings with numbers to make a easy reading pattern

Headings and subheadings in the results section have been numbered.

5. In lines 886-887 number are written and number as used as symbols better to be consistent through the text

We thank the reviewer for their suggestion. However, throughout the text we have consistently used text for numbers 1-9 and symbols for numbers 10 and above. We understand this to be an accepted way to report numbers.

6. In line 1117 the word “well-being” is separated by a high fen but not in Table 6 and other parts in the text

We thank the reviewer for picking up this error. We have removed the hyphen in this instance and double-checked the remaining text. 

7. Some sentences must be split or they need to have more punctuation marks

We have tried to split long sentences or add more punctuation throughout.

---

## [Decision Letter · Decision Letter 2]

4 Oct 2021

Return of individual research results from genomic research: A systematic review of stakeholder perspectives

PONE-D-21-02012R2

Dear Dr. Vears,

We’re pleased to inform you that your manuscript has been judged scientifically suitable for publication and will be formally accepted for publication once it meets all outstanding technical requirements.

Kind regards,

Prof. Ritesh G. Menezes, M.B.B.S., M.D., Diplomate N.B.

Academic Editor

PLOS ONE

Reviewers' comments:

Reviewer's Responses to Questions

**Comments to the Author**

1. If the authors have adequately addressed your comments raised in a previous round of review and you feel that this manuscript is now acceptable for publication, you may indicate that here to bypass the “Comments to the Author” section, enter your conflict of interest statement in the “Confidential to Editor” section, and submit your "Accept" recommendation.

Reviewer #1: All comments have been addressed

Reviewer #3: All comments have been addressed

2. Is the manuscript technically sound, and do the data support the conclusions?

Reviewer #1: Yes

Reviewer #3: Yes

3. Has the statistical analysis been performed appropriately and rigorously? 

Reviewer #1: Yes

Reviewer #3: N/A

4. Have the authors made all data underlying the findings in their manuscript fully available?

Reviewer #1: Yes

Reviewer #3: Yes

5. Is the manuscript presented in an intelligible fashion and written in standard English?

Reviewer #1: Yes

Reviewer #3: Yes

6. Review Comments to the Author

Reviewer #1: This is an excellent overview of all the evidence around the issue of feedback of individual genetic research findings and I thoroughly enjoyed reading it. Congratulations on this comprehensive work and on getting it published!

Reviewer #3: The article has improved through out the revision process. The main aim has been fulfilled and the information is better organised. I agree the article can be accepted for publication

7. PLOS authors have the option to publish the peer review history of their article (what does this mean?). If published, this will include your full peer review and any attached files.

Reviewer #1: No

Reviewer #3: No

---

## [Editor Report · Acceptance letter]

27 Oct 2021

PONE-D-21-02012R2 

Return of individual research results from genomic research: A systematic review of stakeholder perspectives 

Dear Dr. Vears:

I'm pleased to inform you that your manuscript has been deemed suitable for publication in PLOS ONE. Congratulations! Your manuscript is now with our production department. 

Kind regards, 

on behalf of

Prof. Dr. Ritesh G. Menezes 

Academic Editor

PLOS ONE